# Contrastive Moments: Unsupervised Halfspace Learning in Polynomial Time

**Xinyuan Cao**
Georgia Tech
xcao78@gatech.edu

**Santosh S. Vempala**
Georgia Tech
vempala@gatech.edu

## Abstract

We give a polynomial-time algorithm for learning high-dimensional halfspaces with margins in $d$-dimensional space to within desired Total Variation (TV) distance when the ambient distribution is an unknown affine transformation of the $d$-fold product of an (unknown) symmetric one-dimensional logconcave distribution, and the halfspace is introduced by deleting at least an $\epsilon$ fraction of the data in one of the component distributions. Notably, our algorithm does not need labels and establishes the unique (and efficient) identifiability of the hidden halfspace under this distributional assumption. The sample and time complexity of the algorithm are polynomial in the dimension and $1/\epsilon$. The algorithm uses only the first two moments of *suitable re-weightings* of the empirical distribution, which we call *contrastive moments*; its analysis uses classical facts about generalized Dirichlet polynomials and relies crucially on a new monotonicity property of the moment ratio of truncations of logconcave distributions. Such algorithms, based only on first and second moments were suggested in earlier work, but hitherto eluded rigorous guarantees.

Prior work addressed the special case when the underlying distribution is Gaussian via Non-Gaussian Component Analysis. We improve on this by providing polytime guarantees based on TV distance, in place of existing moment-bound guarantees that can be super-polynomial. Our work is also the first to go beyond Gaussians in this setting.

## 1 Introduction

Suppose points in $\mathbb{R}^d$ are labeled according to a linear threshold function (a halfspace). Learning a threshold function from labeled examples is the archetypal well-solved problem in learning theory, in both the PAC and mistake-bound models; its study has led to efficient algorithms, a range of powerful techniques and many interesting learning paradigms. While the sample complexity in general grows with the dimension, when the halfspace has a *margin*, the complexity can instead be bounded in terms of the reciprocal of the squared margin width [PCST99, SBS$^+$00, AV06, LS11]. The problem is also very interesting for special classes of distributions, e.g., when the underlying distribution is logconcave, agnostic learning is possible [KKMS08], and active learning needs fewer samples compared to the general case [BBL06].

The main motivation for our work is learning a halfspace with a margin *with no labels*, i.e., unsupervised learning of halfspaces. This is, of course, impossible in general — there could be multiple halfspaces with margins consistent with the data — raising the question: *Can there be natural distributional assumptions that allow the unsupervised learning of halfspaces?* For example, suppose data is drawn from a Gaussian in $\mathbb{R}^d$ with points in an unknown band removed, i.e., we assume there exists a unit vector $u \in \mathbb{R}^d$ and an interval $[a, b]$ so that the input distribution is the Gaussian restricted to the set $\{x \in \mathbb{R}^d | \langle u, x \rangle \leq a \text{ or } \langle u, x \rangle \geq b\}$. Can the vector $u$ be efficiently learned? Such a distributional assumption ensures that the band normal to $u$ is essentially unique, leaving open the question of whether it can be efficiently learned.

37th Conference on Neural Information Processing Systems (NeurIPS 2023).

Such models have been considered in the literature, notably for Non-Gaussian Component Analysis (NGCA) [BKS+06, TV18], learning relevant subspaces [Blu94, VX11] and low-dimensional convex concepts [Vem10] where data comes from a product distribution with all components being Gaussian except for one (or a small number). It is assumed that the non-Gaussian component differs from Gaussian in some low moment and the goal is to identify this component. Another related model is Independent Component Analysis (ICA) where the input consists of samples from an affine transformation of a product distribution and the goal is to identify the transformation itself [Com94, Car98, GVX14, JKV23]. For this problem to be well-defined, it is important that at most one component of the product distribution is Gaussian. No such assumption is needed for NGCA or the more general problem we consider here.

Formally, we consider the following model and problem, illustrated in Figure 1.1.

**Definition 1** (Affine Product Distribution with $\epsilon$-Margin). Let $q$ be a symmetric one-dimensional isotropic logconcave density function. Let $Q$ be the $d$-fold product distribution obtained from $q$. Let $\hat{q}$ be the isotropized density obtained after restricting $q$ to $\mathbb{R}\backslash[a, b]$ where $q((-\infty, a]) \geq \epsilon, q([a, b]) \geq \epsilon$ and $q([b, \infty)) \geq \epsilon$. Let $P$ be the product of one copy of $\hat{q}$ and $d-1$ copies of $q$. Let $\widehat{P}$ be obtained by a full-rank affine transformation of $P$; we refer to $\widehat{P}$ as an *Affine Product Distribution with $\epsilon$-Margin*. Let $u$ be the unit vector normal to the margin before transformation.

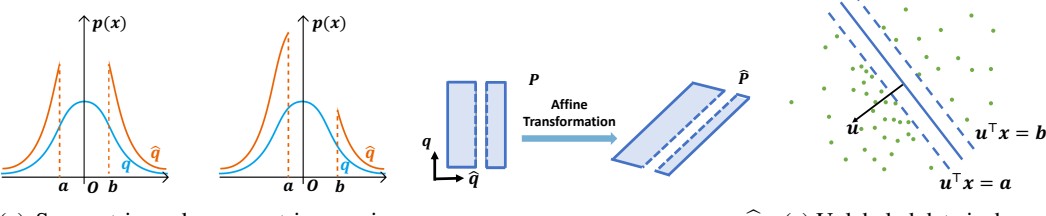

(a) Symmetric and asymmetric margin $[a, b]$. We use contrastive covariance and mean respectively.

(b) Definition 1 with uniform $q$. $\widehat{P}$ is a full-rank affine transformation of the product distribution $P$.

(c) Unlabeled data is drawn from $\widehat{P}$. The goal is to learn the normal vector $u$.

Figure 1.1: Affine Product Distribution with Margin.

With this model in hand, we have the following algorithmic problem.

**Problem.** Given input parameters $\epsilon, \delta > 0$ and access to iid samples from $\widehat{P}$, an affine product distribution with $\epsilon$-margin, the learning problem is to compute a unit vector $\tilde{u}$ that approximates $u$ to within $TV$ distance $\delta$. That is, the TV distance between the corresponding $\tilde{P}$ and $P$ is at most $\delta$, where $\tilde{P}$ is the distribution with margin normal to $\tilde{u}$.

In this formulation of the problem with a TV distance guarantee, if each side of the halfspace receives a different label, then the probability that the output halfspace of the data disagrees with the true label (up to swapping the labels) is at most $\delta$.

A natural approach to identifying the halfspace is maximum margin clustering [XNLS04]: find a partition of the data into two subsets s.t. the distance between the two subsets along some direction is maximized. Unfortunately, this optimization problem is NP-hard, even to approximate.

There are at least two major difficulties we have to address. The first is the unknown affine transformation, which we cannot hope to completely identify in general. The second is that, even if we reversed the transformation, the halfspace normal is in an arbitrary direction in $\mathbb{R}^d$ and would be undetectable in almost all low-dimensional projections, i.e., we have a needle in a haystack problem.

## 1.1 Results and Techniques

We give an efficient algorithm for the unsupervised halfspace learning problem under any symmetric product logconcave distribution. It consists of the following three high-level steps.

(1) Make the data isotropic (mean zero and covariance matrix identity).

(2) Re-weight data and compute the re-weighted mean $\tilde{\mu}_i$ and the top eigenvector $v$ of the re-weighted covariance.

(3) Project data along the vectors $\tilde{\mu}_i, v$, and output the vector with the largest margin.

Although the algorithm is simple and intuitive, its analysis has to overcome substantial challenges. Our main result is the following. The formal version (Theorem 7) is in the Appendix.

**Theorem 1** (Main). *There is an algorithm that can learn any affine product distribution with $\epsilon$-margin to within TV distance $\delta$ with time and sample complexity that are polynomial in $d, 1/\epsilon$ and $1/\delta$ with high probability.*

To see the idea of the algorithm, we first consider the case when no affine transformation is applied. In this case, we can detect the direction $u$ by calculating the empirical mean and top eigenvector of the empirical uncentered covariance matrix. If the margin $[a, b]$ lies on one side of the origin, the mean along $u$ is nonzero while the mean in any other direction that is orthogonal to $u$ is zero. Thus the mean itself reveals the vector $u$. Otherwise, we can show that the second moment along $u$ is higher than along any other orthogonal direction. Thus, there is a positive gap between the top two eigenvalues of the uncentered covariance matrix and the top eigenvector is $u$. In fact, the algorithm applies more generally, to the product distribution created from one-dimensional bounded isoperimetric distributions. A one-dimensional distribution $p$ is isoperimetric if there exists a constant $\psi > 0$ such that for any $x \in \mathbb{R}$, $p(x) \geq \psi \min\{p([x, \infty)), p((-\infty, x])\}$. The theorem is stated as follows. The formal version (Theorem 6), algorithm and proofs are included in the Appendix.

**Theorem 2** (Isotropic Isoperimetric Distribution). *There is an algorithm that can learn any isotropic isoperimetric bounded product distribution with $\epsilon$-margin to within TV distance $\delta$ with time and sample complexity that are polynomial in $d, 1/\epsilon, 1/\delta$ with high probability.*

In the general case, when an unknown affine transformation is applied, the algorithm first computes the empirical mean and covariance of the sample and makes the empirical distribution isotropic. Then we will consider two cases as illustrated in Figure 1.1a. If the unknown band is *not centered* around the mean along $u$, we can expect the empirical mean to differ from the mean of the underlying product distribution without the margin. Consequently, if we knew the latter, we can use the difference to estimate $u$. However, in general, we do not have this information. Instead, we demonstrate that there exists a re-weighting of the sample so that re-weighted empirical mean compared to the unweighted empirical mean is a good estimate of $u$. In other words, with appropriate re-weighting, the mean shifts along the normal direction to the unknown band. On the other hand, if the band is centered along $u$, the mean shift will be zero. In this scenario, we will show that the maximum eigenvector of a re-weighted covariance matrix is nearly parallel to $u$!

Our algorithm only uses first and second order moments, can be implemented efficiently, and is in fact practical (see Section 5). The main challenges are (1) proving the existence of band-revealing re-weightings and (2) showing that a polynomial-sized sample (and polynomial time) suffice.

To prove the main theorem, we will show that either the re-weighted mean induces a contrastive gap (Lemma 1), or the eigenvalues of the re-weighted covariance matrix induce a contrastive gap (Lemma 2). In the subsequent two lemmas, we adopt the notation from Definition 1. Here, $P$ represents a product distribution with $\epsilon$-margin defined by the interval $[a, b]$ (before transformation).

**Lemma 1** (Contrastive Mean). *If $|a + b| > 0$, then for any two distinct nonzero $\alpha_1, \alpha_2 \in \mathbb{R}$, at least one of the corresponding re-weighted means is nonzero, i.e.,*

$$\max\left(\left|\mathop{\mathbb{E}}_{x \sim P} e^{\alpha_1 \|x\|^2} u^\top x\right|, \left|\mathop{\mathbb{E}}_{x \sim P} e^{\alpha_2 \|x\|^2} u^\top x\right|\right) > 0.$$

**Lemma 2** (Contrastive Covariance). *If $a + b = 0$, then there exists an $\alpha < 0$, such that (1) there is a positive gap between the top two eigenvalues of the re-weighted uncentered covariance matrix $\tilde{\Sigma} = \mathbb{E}_{x \sim P} e^{\alpha \|x\|^2} (xx^\top)$. That is, $\lambda_1(\tilde{\Sigma}) > \lambda_2(\tilde{\Sigma})$. (2) The top eigenvector of $\tilde{\Sigma}$ is $u$.*

The proof of Lemma 1 uses Descartes' rule of signs applied to a suitable potential function. To prove Lemma 2, we develop a new monotonicity property of the moment ratio (defined as the ratio of the variance of $X^2$ and the squared mean of $X^2$) for truncations of logconcave distributions. The moment ratio is essentially the square of the coefficient of variation of $X^2$. An insight from the monotonicity of the moment ratio is that for logconcave distributions with positive support, when the distribution is restricted to an interval away from the origin, it needs a smaller sample size to estimate its second moment accurately. We state the lemma as follows.

**Lemma 3** (Monotonicity of Moment Ratio). *Let $q$ be a logconcave distribution in one dimension with nonnegative support. For any $t \geq 0$, let $q_t$ be the distribution obtained by restricting $q$ to $[t, \infty)$. Then the moment ratio of $q$, defined as $mr_q(t) := \frac{\mathrm{var}_{q_t}(X^2)}{(\mathbb{E}_{q_t} X^2)^2}$, is strictly decreasing with $t$.*

To obtain polynomial guarantees, we will need quantitative estimates of the inequalities in the above two lemmas. Establishing such quantitative bounds is the bulk of the technical contribution of this paper. While our focus is on proving *polynomial* bounds, whose existence a priori is far from clear, we did not optimize the polynomial bounds themselves; our experimental results suggest that in fact the dependence on both $d$ and $1/\epsilon$ might be linear!

## 1.2 Related Work

Efficient algorithms for supervised halfspace learning [Ros58, MP69], combined with the kernel trick [CST⁺00, HDO⁺98], serve as the foundation of much of learning theory. Halfspaces *with margin* are also well-studied, due to their motivation from the brain, attribute-efficient learning [Val98, Blu90], random projection based learning [AV06], and turn out to have sample complexity that grows inverse polynomially with the margin, independent of the ambient dimension. When examples are drawn from a unit Euclidean ball in $\mathbb{R}^d$, and the halfspace has margin $\gamma$, then the sample complexity grows as $O(1/\gamma^2)$ regardless of the dimension. This leads to the question of whether labels are even necessary, or the halfspace can be identified from unlabeled samples efficiently — the focus of the present paper.

The model of unsupervised learning we study is similar to other classical models in the literature, notably Independent Component Analysis where input data consists of iid samples from an unknown affine transformation of a product distribution. There, the goal is to recover the affine transformation under minimal assumptions. Known polynomial-time algorithms rely on directional moments, and the assumption that component distributions differ from a Gaussian in some small moment. A related relevant problem, Non-Gaussian Component Analysis (NGCA), aims to extract a hidden non-Gaussian direction in a high-dimensional distribution. Here too, the main idea is the fact that non-Gaussian component must have some finite moment different from that of a Gaussian. While finite moment difference implies a TV distance lower bound, to get $\epsilon$-TV distance, one might need to use $k$'th moments for $k = \Omega(\log(1/\epsilon))$ even for logconcave densities. As the dependence on the moment number is exponential (even for the sample complexity), this approach does not yield polytime algorithms in terms of TV distance, the natural notion for classification.

The idea of applying Principal component analysis (PCA) to re-weighted samples was used in [BV08] to unravel a mixture of well-separated Gaussians. For a mixture of two general Gaussians that are mean separated, after making the mixture isotropic, it was shown that either the mean or top eigenvector of the covariance of a re-weighted sample reveals the vector of the mean differences. This high-level approach was used for solving general ICA by estimating re-weighted higher moments (tensors) [GVX14]. Higher moment re-weightings were also used by [VX11] to give an algorithm for factoring a distribution and learning "subspace juntas", functions of an unknown low-dimensional subspace, and by [TV18] to give a more efficient algorithm for the special case of NGCA. The question of whether expensive higher moment algorithms could be replaced by re-weighted second moments is natural and one variant was specifically suggested by [TV18] for NGCA. Our work validates this intuition with rigorous polynomial-time algorithms.

## 2 Algorithm

Our algorithm first makes the data to be isotropic using the sample mean and sample covariance. Then we apply the weight $w(y, \alpha) = e^{\alpha \|y\|^2}$ to each isotropized sample $y$, and compute the re-weighted mean and the top eigenvector of the re-weighted covariance matrix. Then for each candidate normal vector, we project the data to it, and scan to find the maximum gap. The algorithm outputs the vector with the maximal gap among all candidate vectors. We give the formal description in Algorithm 1.

## 3 Analysis

We demonstrate that Algorithm 1 operates within polynomial time and sample complexity. The details regarding sample complexity are presented in Theorem 1. The time complexity is justified by

---

**Algorithm 1** Unsupervised Halfspace Learning with Contrastive Moments

---

**Input:** Unlabeled data $S = \{x^{(1)}, \cdots, x^{(N)}\} \subset \mathbb{R}^d$. $\epsilon, \delta > 0$.

- (Isotropize) Compute the sample mean and covariance:

$$\hat{\mu} = \frac{1}{N} \sum_{j=1}^{N} x^{(j)}, \qquad \hat{\Sigma} = \frac{1}{N} \sum_{j=1}^{N} (x^{(j)} - \hat{\mu})(x^{(j)} - \hat{\mu})^{\top}.$$

   Make the data isotropic: $y^{(j)} = \hat{\Sigma}^{-1/2}(x^{(j)} - \hat{\mu})$.

- (Re-weighted Moments) Set $\alpha_1 = -c_1\epsilon^{82}/d, \alpha_2 = -c_2\epsilon^{42}/d$ and $\alpha_3 = -c_3\epsilon^2$. Let $w(y, \alpha) = e^{\alpha\|y\|^2}$. Compute the re-weighted sample means $\mu_{\alpha_1}, \mu_{\alpha_2}$ using $\alpha_1, \alpha_2$ and the re-weighted sample covariance using $\alpha_3$ as follows:

$$\tilde{\mu}_{\alpha_i} = \frac{1}{N} \sum_{j=1}^{N} w(y^{(j)}, \alpha_i) y^{(j)}, \text{ for } i \in \{1, 2\} \text{ and } \tilde{\Sigma} = \frac{1}{N} \sum_{j=1}^{N} w(y^{(j)}, \alpha_3) y^{(j)} y^{(j)\top}$$

   Compute the top eigenvector $v$ of $\tilde{\Sigma}$.

- (Max Margin) Calculate the max margin (i.e., maximum gap) of the one-dimensional projections of the data along the vectors $\hat{\mu}_{\alpha_1}, \hat{\mu}_{\alpha_2}, v$, and let $\hat{u}$ be the vector among these with the largest margin.

**return** $\hat{u}$.

---

the algorithm's process: it calculates the re-weighted sample mean and the top eigenvector of the re-weighted sample covariance matrix, both of which require polynomial time.

In our algorithm, we consider two cases depending on whether the removed band $[a, b]$ is origin-symmetric. If it is asymmetric, we will show that one of the re-weighted means with two $\alpha$s gives us the correct direction by showing that the re-weighted mean along $u$ has a gap from zero while the re-weighted mean along all other orthogonal directions is zero. We state the positive gap quantitatively in Lemma 4. Otherwise, if the band is symmetric, we will show a positive gap between the top two eigenvalues of the re-weighted covariance matrix, and the top eigenvector corresponds to $u$. We quantify the gap between the top two eigenvalues in Lemma 5. In the algorithm, since we know neither the underlying distribution mean nor the location of the removed band, we have to compute both re-weighted means and re-weighted covariance, and then get the correct direction among all three candidate vectors by calculating the margin and finding the one with the largest margin.

**Lemma 4** (Quantitative Gap of Contrastive Mean). *Suppose that $|a + b| \geq \epsilon^5$. Then, for $\alpha_1 = -c_1\epsilon^{82}/d, \alpha_2 = -c_2\epsilon^{42}/d$, the re-weighted mean of $P$, denoted as $\mu_{\alpha_1}$ and $\mu_{\alpha_2}$, satisfies*

$$\max\left(\left|u^{\top}\mu_{\alpha_1}\right|, \left|u^{\top}\mu_{\alpha_2}\right|\right) > \frac{C\epsilon^{159}}{d^2} \text{ for some constant } C > 0,$$

$$\forall v \perp u, \quad v^{\top}\mu_{\alpha_1} = v^{\top}\mu_{\alpha_2} = 0.$$

**Lemma 5** (Quantitative Spectral Gap of Contrastive Covariance). *Suppose that $|a + b| < \epsilon^5$. Choose $\alpha_3 = -c_3\epsilon^2$ for some constant $c_3 > 0$. Then, for an absolute constant $C$, the top two eigenvalues $\lambda_1 \geq \lambda_2$ of the corresponding re-weighted covariance of $P$ satisfy*

$$\lambda_1 - \lambda_2 \geq C\epsilon^3\lambda_1.$$

## 4   Proofs

Recall that we are given data $x^{(1)}, \cdots, x^{(N)}$ drawn from the affine product distribution with $\epsilon$-margin $\widehat{P}$. Algorithm 1 first makes the data isotropic. Denote $y^{(1)}, \cdots, y^{(N)}$ as the corresponding isotropicized data. Then each $y^{(j)}$ is an independent and identically distributed variable drawn from $P = \hat{q} \otimes q \otimes \cdots q$. Since we compute the re-weighted moments on $y^{(j)}$ in the algorithm, we analyze the moments of $P$ directly.

Recall in Definition 1 that $q$ is the symmetric one-dimensional isotropic logconcave density function, and $\tilde{q}$ is the density obtained by restricting $q$ to $\mathbb{R}\backslash[a,b]$ for some unknown $a < b$. Denote $\mu_1, \sigma_1^2$ as the mean and variance of $\tilde{q}$. $\hat{q}$ is the density obtained after making $\tilde{q}$ isotropic, with support $\mathbb{R}\backslash[a',b']$, where $a' = \frac{a-\mu_1}{\sigma_1}, b' = \frac{b-\mu_1}{\sigma_1}$. We denote the standard basis of $\mathbb{R}^d$ by $\{e_1, \cdots, e_d\}$, and assume wlog that $e_1 = u$ is the (unknown) normal vector to the band. We write $x_i := \langle x, e_i \rangle$ as $x$'s $i$-th coordinate. We assume in our proof that $|b| > |a|$. If this condition is not met, we can redefine our interval by setting $a' = -b$ and $b' = -a$. The proof can then be applied considering the distribution is restricted to $\{x \in \mathbb{R}^d : u^\top x \leq a' \text{ or } u^\top x \geq b'\}$. For a vector $x \in \mathbb{R}^d$, we use $\|x\|$ to denote its $l_2$ norm. For a matrix $A \in \mathbb{R}^{m \times n}$, we denote its operator norm as $\|A\|_{\mathrm{op}}$.

**Contrastive Mean.** We can write the contrastive mean as a linear combination of exponential functions of $\alpha$. By Descartes' rule of signs, the number of zeros of this function is at most two. Since $\alpha = 0$ is one root and corresponds to mean zero, there is at most one nonzero root. And thus we have that for any two distinct nonzero $\alpha$'s, at least one of them achieves nonzero contrastive mean.

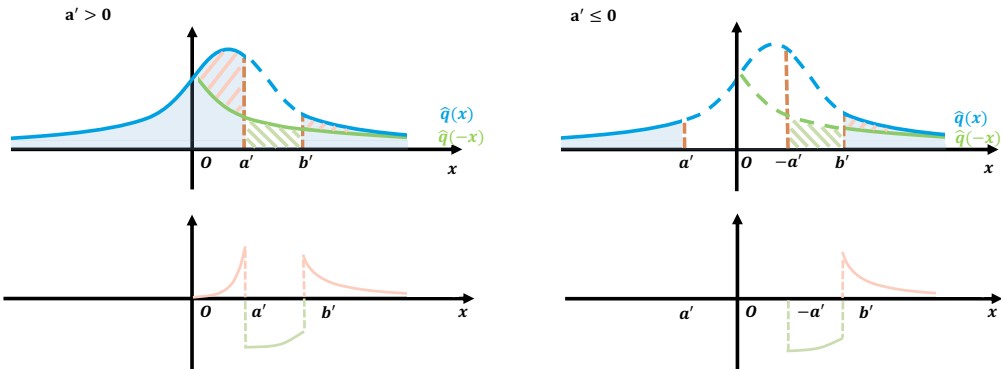

Figure 4.1: Coefficient of $e^{\alpha x^2}$ in $F(\alpha)$. In the proof of Lemma 1, by combining $e^{\alpha x^2}$ terms, we flip $\hat{q}(x)$ horizontally. For $a' > 0$, the coefficient is negative when $x \in (a', b')$ and non-negative outside the interval. For $a' \leq 0$, it is negative when $x \in (-a', b')$ and positive when $x > b'$.

**Theorem 3** (Descartes' Rule of Signs in the Integral Form). *Let $F(\alpha) = \int_0^\infty a(x)e^{\alpha x^2}\,dx$. Then the number of roots of $F(\alpha) = 0$ is at most the number of sign changes in $a(x)$ for $x \geq 0$.*

*Proof of Lemma 1 (Contrastive Mean).* $|b| > |a|$ implies that $\mu_1 < 0$. For any $x \geq 0$, we have

$$\hat{q}(x) = \frac{\sigma_1 q(x\sigma_1 + \mu_1)}{1 - \int_a^b q(x)\,dx} \geq \frac{\sigma_1 q(-x\sigma_1 + \mu_1)}{1 - \int_a^b q(x)\,dx} = \hat{q}(-x).$$

Since $P$ is a product distribution, we have

$$\mathbb{E}_{x \sim P} e^{\alpha\|x\|^2} x_1 = \mathbb{E}_{x_1 \sim \hat{q}} e^{\alpha x_1^2} x_1 \cdot \prod_{i=2}^d \mathbb{E}_{x_i \sim q} e^{\alpha x_i^2}$$

We denote

$$F(\alpha) = \mathbb{E}_{x \sim \hat{q}} e^{\alpha x_1^2} x_1 = \int_{\mathbb{R}\backslash[a',b']} e^{\alpha x^2} x\hat{q}(x)\,dx \tag{4.1}$$

Then we rearrange $F(\alpha)$ by combining $e^{\alpha x^2}$ as in Figure 4.1.

If $a' \leq 0$, we rewrite $F(\alpha)$ as

$$F(\alpha) = -\int_{-a'}^{b'} x\hat{q}(-x)e^{\alpha x^2}\,dx + \int_{b'}^\infty x(\hat{q}(x) - \hat{q}(-x))e^{\alpha x^2}\,dx$$

We treat $F(\alpha)$ as the integral of $a(x)e^{\alpha x^2}$ for $x \geq -a'$. Since $\hat{q}(x) - \hat{q}(-x) > 0$ for $x > b'$, we have $a(x) > 0$ for $x \in [-a', b')$ and $a(x) < 0$ for $x > b'$. In other words, for increasing $x$, the sign of $a(x)$ only changes once. By Theorem 3, $F(\alpha) = 0$ has at most one root.

If $a' > 0$, we arrange $F(\alpha)$ in the same way and get

$$F(\alpha) = \int_0^{a'} x(\hat{q}(x) - \hat{q}(-x))e^{\alpha x^2}\,dx - \int_{a'}^{b'} x\hat{q}(-x)e^{\alpha x^2}\,dx + \int_{b'}^{\infty} x(\hat{q}(x) - \hat{q}(-x))e^{\alpha x^2}\,dx$$

Similarly, we treat $F(\alpha)$ as the integral of $a(x)e^{\alpha x^2}$ for $x \geq 0$. For increasing $x$, the sign of $a(x)$ changes twice. By Descartes' rule of signs, $F(\alpha) = 0$ has at most two roots. In addition, we know $F(0) = \mathbb{E}_P x_1 = 0$ by definition of $P$, which implies that $\alpha = 0$ is one root of $F(\alpha)$. So there is at most one nonzero root of $F(\alpha)$. In other words, for any two distinct nonzero $\alpha_1, \alpha_2$, at least one of $F(\alpha_1), F(\alpha_2)$ is nonzero. Consequently,

$$\max\left( \left| \mathbb{E}_{x \sim P} e^{\alpha_1 \|x\|^2} x_1 \right|, \left| \mathbb{E}_{x \sim P} e^{\alpha_2 \|x\|^2} x_1 \right| \right) > 0.$$

$\square$

*Proof Idea of Lemma 4 (Quantitative Gap of Contrastive Mean).* To get a quantitative bound on the contrastive mean, we follow the same idea of bounding the number of roots of $F(\alpha)$ as in Lemma 1. By taking the derivative of $F(\alpha)$, we can show that either $F'(0) \neq 0$ or $F''(0) \neq 0$. Then by Taylor expansion, we can choose two distinct $\alpha$'s (near zero) so that one of the corresponding contrastive means is bounded away from zero.

**Moment Ratio and Contrastive Covariance.** To prove Lemma 2, we develop a new monotonicity property of the moment ratio of logconcave distributions. Moment ratio is specifically defined as the ratio of the fourth moment to the square of the second moment of truncated versions of the distribution. This measurement essentially reflects the uncentered kurtosis of the distribution. We will prove the monotonicity of the moment ratio by reducing the case of general logconcave distributions to exponential distributions.

*Proof of Lemma 3 (Moment Ratio).* We show the monotonicity of moment ratio by showing its derivative with respect to $t$ is negative. Define $M_k(t) = \int_t^{\infty} x^k q(x)\,dx$. By calculation, the derivative of moment ratio is proportional to $-H(t)$, where $H(t)$ is defined as follows:

$$H(t) = t^4 M_0(t) M_2(t) + M_2(t) M_4(t) - 2t^2 M_0(t) M_4(t).$$

Let $h(x) = \beta e^{-\gamma x}$ be an exponential function ($\beta, \gamma > 0$) such that

$$M_0(t) = N_0(t),\ M_2(t) = N_2(t),\ \text{where } N_k(t) = \int_t^{\infty} x^k h(x)\,dx,\ k \in \mathbb{N}.$$

Then we have

$$\int_t^{\infty} (h(x) - q(x))\,dx = 0,\ \int_t^{\infty} x^2(h(x) - q(x)) = 0$$

By the logconcavity of $q$, the graph of $h$ intersects with the graph of $q$ at exactly two points $u' < v$, where $v > 0$. Also we have $h(x) \leq q(x)$ at the interval $[u', v]$ and $h(x) > q(x)$ outside the interval. Let $u = \max\{0, u'\}$. So for $x \geq 0$, $(x - u)(x - v)$ has the same sign as $h(x) - q(x)$. Since $t \geq 0$, we have

$$\int_t^{\infty} (x^2 - u^2)(x^2 - v^2)(h(x) - q(x)) \geq 0$$

Expanding the terms and we get

$$\int_t^{\infty} x^4(h(x) - q(x)) \geq (u^2 + v^2)\int_t^{\infty} x^2(h(x) - q(x))\,dx - u^2 v^2 \int_t^{\infty} (h(x) - q(x))\,dx = 0$$

This shows that $N_4(t) \geq M_4(t)$. To show that $H(t) > 0$, we consider two cases.

Firstly if $M_2(t) - 2t^2 \geq 0$, we have

$$H(t) = t^4 M_0(t) M_2(t) + M_4(t)(M_2(t) - 2t^2) > 0.$$

Secondly if $M_2(t) - 2t^2 < 0$, by calculation of the exponential function's moments, we have
$$t^4 N_0(t) N_2(t) > -N_4(t)(N_2(t) - 2t^2)$$
This implies that
$$H(t) = t^4 M_0(t) M_2(t) + M_4(t)(M_2(t) - 2t^2) \geq (M_2(t) - 2t^2)(M_4(t) - N_4(t)) \geq 0$$
The equality holds if and only if $M_4(t) = N_4(t) > 0$. Then we have
$$H(t) = t^4 N_0(t) N_2(t) + N_4(t)(N_2(t) - 2t^2) > 0.$$
Combining both cases, $\mathrm{mr}'_q(t) < 0, \forall t \geq 0$, which implies that the moment ratio of $q$ is strictly decreasing with respect to $t$. $\square$

*Proof Idea of Lemma 2 (Constrastive Covariance).* View the spectral gap of the re-weighted covariance, denoted as $\lambda_1(\tilde{\Sigma}) - \lambda_2(\tilde{\Sigma})$, as a function $S(\alpha)$. By calculation, $S(0) = 0$ and $S'(0)$ is proportional to $\mathrm{mr}_q(b) - \mathrm{mr}_q(0)$, which is negative by the monotonicity property of moment ratio. Then we can prove the spectral gap for the chosen $\alpha$ using Taylor expansion.

*Proof Idea of Lemma 5 (Quantitative Gap of Contrastive Covariance).* We first prove the case when $a = -b$, and then extend the result to the near-symmetric case when $|a + b| < \epsilon^5$ by comparing the re-weighted second moments of two distributions created by restricting $q$ to $\mathbb{R} \setminus [a, b]$ and $\mathbb{R} \setminus [-b, b]$ respectively.

The following theorem enables us to bound the sample complexity to estimate the covariance matrix.
**Lemma 6** (Covariance Estimation [SV13]). *Consider independent isotropic random vectors $X_i$ in $\mathbb{R}^d$ s.t. for some $C, \eta > 0$, for every orthogonal projection $P$ in $\mathbb{R}^d$,*
$$\mathbb{P}(\|PX_i\| > t) \leq Ct^{-1-\eta} \text{ for } t > C\mathrm{rank}(P).$$
*Let $\epsilon \in (0, 1)$. Then with the sample size $N = O(d\epsilon^{-2-2/\eta})$, we have*
$$\mathbb{E} \|\Sigma - \hat{\Sigma}\|_{op} \leq \epsilon \|\Sigma\|_{op}.$$

The following classical theorem will allow us to use the eigenvalue gap to identify the relevant vector.
**Lemma 7** (Davis-Kahan [DK70]). *Let $S$ and $T$ be symmetric matrices with the same dimensions. For a fixed $i$, assume that the largest eigenvalue of $S$ is well separated from the second largest eigenvalue of $S$, i.e.$\exists \delta > 0$ s.t. $\lambda_1(S) - \lambda_2(S) > \delta$. Then for the top eigenvectors of $S$ and $T$, we have*
$$\sin \theta(v_1(S), v_1(T)) \leq \frac{2\|S - T\|_{op}}{\delta}.$$
**Proof Sketch of Main Theorem.** We prove the theorem by considering whether the removed band $[a, b]$ is symmetric or not. If $|a + b| \geq \epsilon^5$, by Lemma 4, for the chosen two $\alpha$s, at least one of the contrastive means $\tilde{\mu}_i$ along $u$ direction is bounded away from zero (by $\mathrm{poly}(1/d, \epsilon)$) and the projection of the contrastive mean along any direction orthogonal to $u$ is zero by symmetry. So by Chebyshev's Inequality, we will ensure that the angle between $u$ and $\tilde{\mu}_i$ is less than $\delta$ with high probability using $O(\mathrm{poly}(d, 1/\epsilon, 1/\delta))$ samples. On the other hand, if $|a + b| < \epsilon^5$, we rely on the contrastive covariance. By Lemma 5, the top eigenvector $v$ aligns with $u$ while the top two eigenvalues $\lambda_1 > \lambda_2$ satisfy $\lambda_1 - \lambda_2 > \mathrm{poly}(\epsilon)$. By Lemma 7, we can upper bound the angle between $u$ and $v$ by the quotient of the operator norm of the difference of the contrastive covariance and sample contrastive covariance and the spectral gap $\lambda_1 - \lambda_2$. The covariance matrix itself can be estimated to desired accuracy efficiently with $O(\mathrm{poly}(d, 1/\epsilon, 1/\delta))$ samples by Lemma 6. This ensures the closeness of $u$ and top eigenvector $v$ and hence a TV-distance guarantee. The formal statement of the theorem and the proofs are included in the Appendix.

## 5 Experiments

While our primary goal is to establish polynomial bounds on the sample and time complexity, our algorithms are natural and easy to implement. We study the efficiency and performance of Algorithm 1 on data drawn from affine product distributions with margin. Here we consider three special cases of logconcave distribution: Gaussian, uniform in an interval and exponential. We include four experiments. In all results, we measure the performance of the algorithm using the sin of the angle between the true normal vector $u$ and the predicted vector $\hat{u}$, i.e., $\sin \theta(u, \hat{u})$, which bounds the $TV$ distance between the underlying distribution and the predicted one after isotropic transformation. Experimental results strongly suggest that the sample complexity is a small polynomial, perhaps even just nearly linear in both the dimension and the separation parameter $\epsilon$.

**Overall Performance.** Here we conduct the experiments based on a grid search of $(a, b)$ pairs on three special cases of logconcave distribution: Gaussian, uniform in an interval and exponential. We measure the performance of $(a, b)$ pairs, where for each pair of $(a, b)$, we conduct five independent trials. For Gaussian and exponential distribution, we choose $-3 \le a < b \le 3$ and for uniform distribution, we choose $-1.5 \le a < b \le 1.5$. Here we set the dimension $d = 10$ and sample size $N = 1000000$. For the parameters, we choose $\alpha_1 = \alpha_3 = -0.1, \alpha_2 = -0.2$. See Figure 5.1 as the heatmap of $\sin \theta(u, \hat{u})$ given different pairs of $(a, b)$.

Although in Algorithm 1, we use extremely small values of the weight parameter $\alpha$, our experiments show that larger constant values also work empirically, leading to much smaller sample complexity. This coincides with our qualitative lemmas (Lemma 1, Lemma 2).

The algorithm performs well as seen in the results, except when $a$ and $b$ are both close to the edge, and thus there is almost no mass on one side of the band. Also, the uniform distribution is the easiest to learn, while the exponential is the hardest among these three distributions. As shown in all three plots, the algorithm performs the best when $a$ and $b$ are near symmetric with origin. In other words, contrastive covariance has better sample complexity than contrastive mean when we fix other hyperparameters. This coincides with our sample complexity bounds as in the proof of Theorem 1.

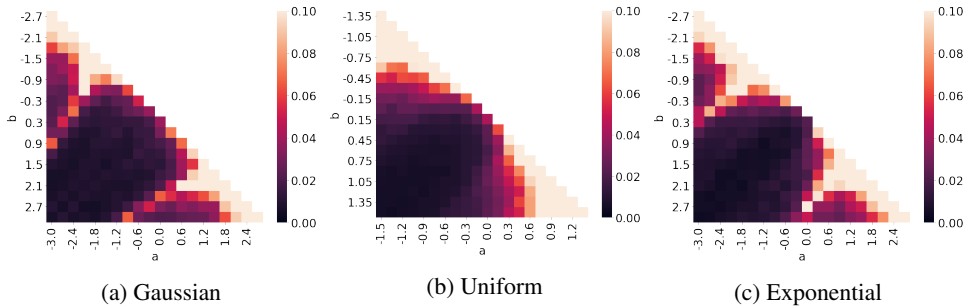

(a) Gaussian       (b) Uniform       (c) Exponential

Figure 5.1: The performance of Algorithm 1 with varying $(a, b)$.

**Performance of Contrastive Mean and Covariance.** In this experiment, we fix a negative $a$ as the left endpoint of the removed band, and measure the performance of both contrastive mean and contrastive covariance with respect to different margin right endpoint $b$. As shown in Figure 5.2, contrastive mean performs well except when $a + b$ is close to zero, while contrastive covariance performs well only when $a + b$ is close to zero. This coincides with our algorithm and analysis for the two cases. In addition, our algorithm chooses the best normal vector among candidates from both contrastive mean and covariance. So our algorithm achieves good performance (minimum of contrastive mean and covariance curves).

Specifically, we choose $a = -2, b \in [-1.9, 4]$ for Gaussian and exponential cases, and $a = -0.5, b \in [-0.4, 0.9]$ for uniform case. We choose the dimension $d = 10$, the sample size $N = 2000000$. We choose $\alpha_1 = \alpha_3 = -0.1, \alpha_2 = -0.2$. We average the result with 50 independent trials.

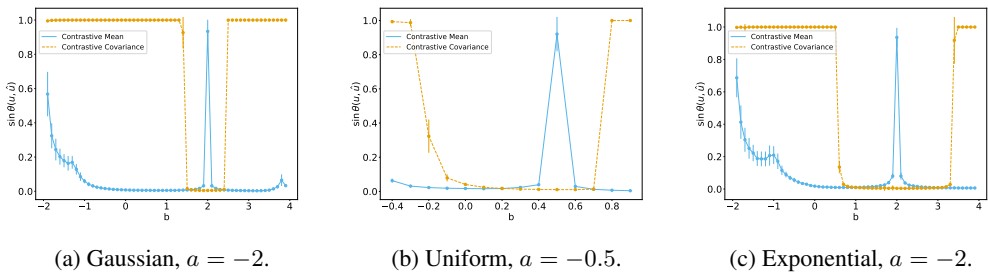

(a) Gaussian, $a = -2$.   (b) Uniform, $a = -0.5$.   (c) Exponential, $a = -2$.

Figure 5.2: The performance of Algorithm 1 for fixed $a$ and varying $b$. The yellow lines show the result computed using the top eigenvector of the contrastive covariance. The blue dotted lines show the better of the two contrastive means.

**Dimension Dependence.** In this experiment, we show the relationship between the input dimension $d$ and the sample complexity. For fixed number size $N = 1000000$, we measure the performance of our algorithm with different $d$. The result is averaged based on a grid search of $(a, b)$ pairs, where for each pair of $(a, b)$, we conduct five independent trials. For Gaussian and exponential distributions, we choose $-3 \leq a < b \leq 3$ and for uniform distribution, we choose $-0.8 \leq a < b \leq 0.8$.

As shown Figure 5.3, the performance scales linearly with growing dimension $d$, suggesting a linear relationship between the sample complexity and the input dimension.

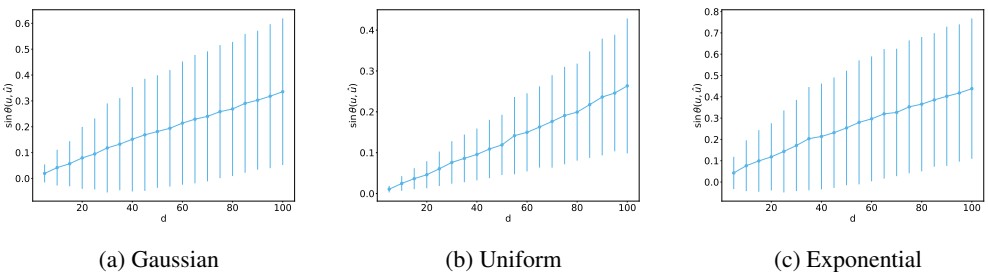

| (a) Gaussian | (b) Uniform | (c) Exponential |

Figure 5.3: The performance of Algorithm 1 for fixed sample size $N$ and varying dimension $d$.

$\epsilon$**-Dependence.** To further understand the dependence on the separation parameter $\epsilon$, we plot the performance versus $1/\epsilon$ in Figure 5.4. Here we calculate $1/\epsilon$ as $1/q([a, b])$, and the performance as the median $\sin \theta(u, \hat{u})$ for specific mass $q([a, b])$. As we can see the performance drops near linearly with respect to $1/\epsilon$, which indicates that the sample complexity is possibly linear in $1/\epsilon$ as well.

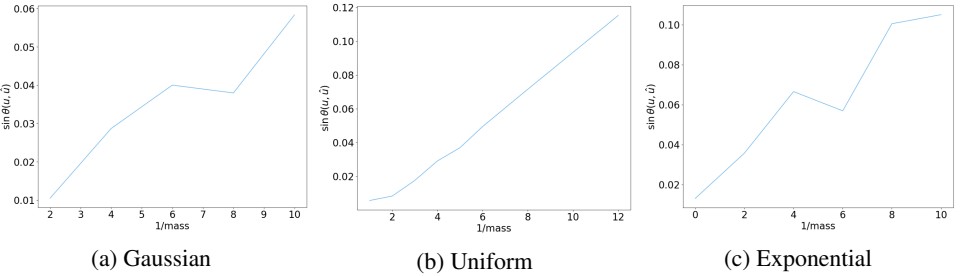

| (a) Gaussian | (b) Uniform | (c) Exponential |

Figure 5.4: The performance with respect to $1/\epsilon$.

# 6 Discussion

We have proposed and analyzed an efficient algorithm to learn the symmetric product logconcave distribution without labeled data. This is also connected to self-supervised learning. Specifically, contrastive learning without data augmentation is closely related to the contrastive covariance part in our algorithm.

The algorithm delivers more than the theoretical analysis. While our analysis focuses on specific values of $\alpha$, as demonstrated by the qualitative lemmas (Lemma 1, Lemma 2), any distinct pair of nonzero $\alpha$ values should work for contrastive mean, and any bounded small $\alpha$ should work for contrastive covariance. This flexibility ensures the applicability of the algorithm in various real-world scenarios. Furthermore, our experimental results align with this claim.

The experiments reveals a linear relationship with the input dimension $d$, raising an open question regarding the improvement of the sample complexity bound to be linear with respect to $d$ (as well as $1/\epsilon$). Additionally, it might be possible to extend the algorithm's application to more general distributions.

**Acknowledgements.** This work was supported in part by NSF awards CCF-2007443 and CCF-2134105 and an ARC fellowship.

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

# Appendix

## A    Preliminaries

### A.1    Logconcave Distributions

**Lemma 8** (Lemma 5.4, [LV07])**.** *Let $X$ be a random point drawn from a one-dimensional logconcave distribution. Then*

$$\mathbb{P}(X \geq \mathbb{E}X) \geq \frac{1}{e}.$$

**Lemma 9** (Lemma 5.5,[LV07])**.** *Let $p : \mathbb{R} \to \mathbb{R}_+$ be an isotropic logconcave density function. Then we have*

- *(a)  For all $x$, $g(x) \leq 1$.*

- *(b)  $g(0) \geq 1/8$.*

**Lemma 10** (Lemma 5.6,[LV07])**.** *Let $X$ be a random point drawn from a logconcave density function $p : \mathbb{R} \to \mathbb{R}_+$. Then for every $c \geq 0$,*

$$\mathbb{P}(p(X) \leq c) \leq \frac{c}{\max_x p(x)}$$

**Lemma 11** (Lemma 5.7, [LV07])**.** *Let $X$ be a random variable drawn from a logconcave distribution in $\mathbb{R}$. Assume that $\mathbb{E}X^2 \leq 1$. Then for any $t \geq 1$, we have*

$$\mathbb{P}(X \geq t) \leq e^{1-t}$$

**Corollary 1.** *Let $X$ be a random point drawn from an isotropic symmetric logconcave density function $p : \mathbb{R} \to \mathbb{R}_+$. Then we have for $t \geq 0$, we have*

$$\mathbb{P}(X \geq t) \leq 8p(t)$$

*Proof.* Since $p(x)$ is symmetric, we know $p(x)$ is monotonically decreasing for $x \geq 0$. Then we apply Lemma 10 with $c = p(x)$, and get

$$\mathbb{P}(x \geq t) \leq \mathbb{P}(p(X) \leq p(t)) \leq \frac{p(t)}{\max_x p(x)}$$

On other hand, by Lemma 9, we have $\max_x p(x) \geq p(0) \geq 1/8$. So we have

$$\mathbb{P}(X \geq t) \leq 8p(t).$$

$\square$

**Lemma 12** (Theorem 5.22, [LV07])**.** *For a random point $X$ drawn from a logconcave distribution in $\mathbb{R}^d$, then*

$$\mathbb{E}|X|^k \leq (2k)^k (\mathbb{E}(|X|))^k$$

**Lemma 13.** *Let $X$ be a random point drawn from an isotropic symmetric logconcave density function $p : \mathbb{R} \to \mathbb{R}_+$. Then for any $t \geq 3$, we have*

$$p(t) \leq p(0) \cdot 2^{-t/3}$$

*Proof.* First we claim that $p(3) < p(0)/2$. Otherwise,

$$\mathbb{E}X^2 \geq \int_0^3 x^2 p(x)\,dx \geq \frac{p(0)}{2}\frac{3^3}{3} \geq \frac{9}{16} > \frac{1}{2}$$

This leads to the contradiction. Then for any $t \geq 3$, from the logconcavity of $p$,

$$p(3) \geq p(0)^{1-\frac{3}{t}} p(t)^{\frac{3}{t}}$$

This implies that

$$p(t) \leq p(0) \cdot 2^{-t/3}$$

$\square$

## A.2 Descartes' Rule of Signs

Descartes' rule of signs is a well-known principle in algebra that offers a way to estimate the maximum number of real roots for any polynomial. This classical theorem can be stated as follows:

**Theorem 4** (Descartes' Rule of Signs). *For the generalized Dirichlet polynomial*

$$F(x) = \sum_{j=1}^{n} a_j e^{p_j x}, p_1 \geq p_2 \geq \cdots \geq p_n,$$

*the number of roots of $F(x) = 0$ is at most the number of sign changes in the series $\{a_1, a_2, \cdots, a_n\}$.*

In this section, we state and prove a variant of Descartes' rule of signs in the integral form, which we apply directly to prove Lemma 1. To begin with, we say a function $f$ has a root of order $k$ at point $x$ if

$$f(x) = f'(x) = \cdots = f^{(k-1)}(x) = 0 \text{ and } f^{(k)}(x) \neq 0.$$

We denote $Z(f)$ as the number of roots of $f$, counted with their orders. Then we can show that the number of roots of $f$ is upper bounded by one plus the number of roots of $f'$ in Lemma 14. We use Rolle's Theorem in the proof of the lemma.

**Theorem 5** (Rolle's Theorem). *Suppose that a function $f$ is differentiable at all points of interval $[a, b]$ and $f(a) = f(b)$. Then there is at least one point $x_0 \in (a, b)$ such that $f'(x_0) = 0$.*

**Lemma 14.** $Z(f) \leq Z(f') + 1$.

*Proof.* Suppose $f$ has a root of order $k_r$ as $x_r, 1 \leq r \leq n$. Then $f'$ has a root of order $k - 1$ at $x_r$. These add up to

$$\sum_{r=1}^{n} (k_r - 1) = Z(f) - n$$

By Rolle's Theorem, $f'(x)$ also has at least $n - 1$ roots in the gaps between the points $x_r$. Together, these two facts give

$$Z(f') \geq Z(f) - n + n - 1 = Z(f) - 1.$$

$\square$

**Theorem 3** (Descartes' Rule of Signs in the Integral Form). *Let $F(\alpha) = \int_0^\infty a(x)e^{\alpha x^2}\, dx$. Then the number of roots of $F(\alpha) = 0$ is at most the number of sign changes in $a(x)$ for $x \geq 0$.*

*Proof.* We prove the theorem with induction on the number of sign changes of $a(x)$. For the base case when the number of sign changes of $a(x)$ is zero, we assume wlog that $a(x) \geq 0$. Then $F(\alpha) > 0$ and thus $F(\alpha)$ has no root. Now we assume that the theorem holds when the number of sign changes is $t$ and we will show the $t + 1$ case.

Let one of the sign changes of $a(x)$ occurs at $x_0$. Define

$$F_0(\alpha) := \int_0^\infty a(x)e^{\alpha(x^2 - x_0^2)}\, dx,$$

which has the same roots as $F(\alpha)$. By taking derivative, we get

$$F_0'(\alpha) = \int_0^\infty a(x)(x^2 - x_0^2)e^{\alpha(x^2 - x_0^2)}\, dx.$$

Let $b(x) = a(x)(x^2 - x_0^2)$ be the new sequence. Then $b(x)$ has one less sign changes than $a(x)$. By induction hypothesis, the number of roots of $F_0'(\alpha)$ is upper bounded by the number of sign changes of $b(x)$. By Lemma 14, the number of roots of $F_0$ is upper bounded by the number of sign changes of $a(x)$, thus leading to the induction step.

$\square$

# B Warm-up: Isotropic Isoperimetric Distribution with $\epsilon$-Margin

As a warm-up, we consider the isotropic product distributions with $\epsilon$-margin. Notably, without applying an unknown transformation on data, we can extend the logconcave distributions to isoperimetric distributions. In this section, we will demonstrate how to retrieve the normal vector $u$ by calculating the empirical mean and top eigenvector of the empirical uncentered covariance matrix. This technique is similar to Principal Component Analysis (PCA), but instead of computing covariance matrix, we use the uncentered covariance matrix.

**Definition 2.** A distribution $p$ with support $\mathbb{R}$ is $\psi$-isoperimetric if there exists $\psi > 0$ such that for any $x \in \mathbb{R}$, we have $p(x) \geq \psi \min\{p([x, \infty)), p((-\infty, x])\}$.

**Definition 3** (Isotropic Isoperimetric Distribution with $\epsilon$-Margin). Let $q_1, \ldots, q_d$ be symmetric one-dimensional isotropic $\psi$-isoperimetric density functions bounded by $\tau$. Let $Q = q_1 \otimes \cdots \otimes q_d$. Let $\hat{q}$ be the density obtained after restricting $q_1$ to $\mathbb{R} \backslash [a, b]$ where $q_1((-\infty, a]) \geq \epsilon, q_1([a, b]) \geq \epsilon$ and $q_1([b, \infty)) \geq \epsilon$. Let $P$ be an arbitrary rotation of $\hat{q} \otimes q_2 \otimes \cdots \otimes q_d$. We refer $P$ as an *Isotropic Isoperimetric Distribution with $\epsilon$-Margin*. Let $u$ be the unit vector normal to the margin.

**Problem.** Given input parameters $\epsilon, \delta > 0$ and access to iid samples from $P$, an isotropic isoperimetric distribution with $\epsilon$-margin, the learning problem is to compute a unit vector $\tilde{u}$ that approximates $u$ to within TV distance $\delta$. That is, the TV distance between the corresponding $\tilde{P}$ and $P$ is at most $\delta$, where $\tilde{P}$ is the distribution with margin normal to $\tilde{u}$.

## B.1 Algorithm

Given data drawn from $P$, we compute the sample mean and the top eigenvector of the uncentered covariance matrix. Then we compare the max margin along these two candidate normal vectors. This gives an efficient algorithm for the problem with no re-weighting. We state the algorithm formally in Algorithm 2.

---

**Algorithm 2** Unsupervised Halfspace Learning from Isotropic Isoperimetric Data

---

**Input:** Unlabeled data $\{x^{(1)}, \cdots, x^{(N)}\} \subset \mathbb{R}^d$. $\epsilon, \delta > 0$.

- Compute the sample mean and uncentered covariance matrix:

$$\hat{\mu} = \frac{1}{N} \sum_{j=1}^{N} x^{(j)}, \hat{\Sigma} = \frac{1}{N} \sum_{j=1}^{N} x^{(j)} x^{(j)\top}$$

- Compute $\hat{\Sigma}$'s top eigenvector $v$.

- Calculate the max margin (i.e., maximum gap) of the one-dimensional projections of the data along the vectors $\hat{\mu}, v$. Let $\hat{u}$ be the vector among these two with a larger margin.

**return** *the vector $\hat{u}$.*

---

## B.2 Analysis

We demonstrate that Algorithm 2 operates within polynomial time and sample complexity. The details regarding sample complexity are presented in Theorem 2 (formal statement of Theorem 6). The time complexity is justified by the algorithm's process: it calculates the sample mean and the top eigenvector of the sample covariance matrix, both of which require polynomial time.

**Theorem 6** (Sample Complexity for Isotropic Isoperimetric Distribution). *Algorithm 2 with $N = \tilde{O}(d^2 \epsilon^{-6} \delta^{-2} \xi^{-1})$ samples learns the target isotropic isoperimetric distribution with $\epsilon$-margin to within TV distance $\delta$ with probability $1 - \xi$.*

The analytical approach is straightforward. Given that the component distributions are isotropic, the empirical mean will reveal the band if the removed band $[a, b]$ stays on one side of the origin. Otherwise, when $[a, b]$ spans across the origin, the variance along the component with the deleted band will increase. Consequently, this component emerges as the top principal component. Intriguing, this

property is "opposite" to the method used to identify low-dimensional convex concepts in [Vem10]. The latter relies on the Brascamp-Lieb inequality, where the variance of a restricted Gaussian is less than that of the original Gaussian.

To prove Theorem 6, we aim to quantify either the mean gap or the spectral gap (gap between the top two eigenvalues) of the uncentered covariance matrix. Specifically, Lemma 15 indicates that when $0 \leq a < b$, the mean along the direction $u$ significantly deviates from zero. Meanwhile, Lemma 16 demonstrates that when $a \leq 0 < b$, there's a gap between the first and second eigenvalues of the uncentered covariance matrix. Subsequently, we employ Lemma 6 [SV13] to determine the sample complexity, and utilize the Davis-Kahan Theorem [DK70] (Lemma 7) to leverage the eigenvalue gap in identifying the pertinent vector $u$. We leave the proof of the lemmas in Section B.3.

For any $x \in \mathbb{R}$, we denote $x_i$ as its $i$-th coordinate. We use $\|x\|$ to denote its $l_2$ norm. For a matrix $A \in \mathbb{R}^{m \times n}$, we denote its operator norm as $\|A\|_{\text{op}}$. We denote the standard basis of $\mathbb{R}^d$ by $\{e_1, \cdots, e_d\}$, and assume wlog that $e_1 = u$ is the (unknown) normal vector to the band. Denote $\Sigma = \mathbb{E}_{x \sim P} xx^\top$ as the uncentered covariance matrix of $P$, with eigenvalues $\lambda_1 \geq \lambda_2 \geq \cdots \geq \lambda_d$.

**Lemma 15** (Mean Gap). *For $0 \leq a < b$ and $b - a \geq c\epsilon$ for constant $c > 0$, we have*

$$\mathbb{E}_{x \sim \hat{q}} x < -\frac{\psi c^2 \epsilon^3}{2}, \operatorname*{var}_{x \sim \hat{q}} x \leq \frac{1}{2\epsilon}.$$

**Lemma 16** (Spectral Gap of Covariance). *If $a \leq 0 < b$ and $b > c\epsilon$ for constant $c > 0$, then the first and second eigenvalues of the uncentered covariance matrix $\Sigma$ have the following gap*

$$\lambda_1 - \lambda_2 > C\epsilon^3 \lambda_1 \text{ for constant } C > 0.$$

*Furthermore, the top eigenvector corresponds to $u$.*

Now we are ready to prove Theorem 6.

*Proof of Theorem 6.* We can proceed with the assumption that $|b| > |a|$. If this condition is not met, we can redefine our interval by setting $a' = -b$ and $b' = -a$. The proof can then be applied considering the distribution is restricted to $\{x \in \mathbb{R}^d : u^\top x \leq a' \text{ or } u^\top x \geq b'\}$. We will prove the theorem by considering two cases: $0 \leq a < b$ and $a \leq 0 < b$.

We first consider the case when $0 \leq a < b$. Given that $q_1$ is bounded by $\tau$, it follows that $b - a > \epsilon/\tau$. By Lemma 15, we know

$$\mathbb{E}_{x \sim P} x_1 < -\frac{\psi \epsilon^3}{2\tau^2}, \operatorname*{var}_{x \sim P} x_1 \leq \frac{1}{2\epsilon},$$

while for $i \geq 2$, we have

$$\mathbb{E}_{x \sim P} x_i = 0, \operatorname*{var}_{x \sim P} x_i = 1.$$

Given data $x^{(1)}, \cdots, x^{(N)}$, let $\hat{\mu} = \frac{1}{N} \sum_{j=1}^N x^{(j)}$ be the sample mean. Then by Chebyshev's Inequality,

$$\mathbb{P}(\hat{\mu}_1 > -\frac{\psi \epsilon^3}{4\tau^2}) \leq \frac{8\tau^4}{N\psi^2 \epsilon^7}, \quad \mathbb{P}(\hat{\mu}_i < -\frac{\psi \epsilon^3 \delta}{4\tau^2 \sqrt{d}}) \leq \frac{16\tau^4 d}{N\psi^2 \epsilon^6 \delta^2}, 2 \leq i \leq d$$

Let $0 < \xi < 1$. So we know with sample size $N_1 = \frac{16\tau^4 d^2}{\epsilon^7 \delta^2 \psi^2 \xi}$,

$$\mathbb{P}(\hat{\mu}_1 > -\frac{\psi \epsilon^3}{4\tau^2}) \leq \frac{\psi^2 \delta^2 \xi}{2d^2} < \frac{\xi}{d}, \quad \mathbb{P}(\hat{\mu}_i < -\frac{\psi \epsilon^3 \delta}{4\tau^2 \sqrt{d}}) \leq \frac{\epsilon \xi}{d} < \frac{\xi}{d}$$

Then we have

$$\begin{aligned}
\mathbb{P}(\sin \theta(\hat{\mu}, e_1) \leq \delta) = & \mathbb{P}(\frac{\hat{\mu}_1^2}{\sum_{i=1}^d \hat{\mu}_i^2} \geq 1 - \delta^2) \\
\geq & \mathbb{P}(\hat{\mu}_1 < -\frac{\psi \epsilon^3}{4\tau^2}, \hat{\mu}_i > -\frac{\psi \epsilon^3 \delta}{4\tau^2 \sqrt{d}}, 2 \leq i \leq d) \\
\geq & 1 - \xi
\end{aligned}$$

Secondly, we consider the case where $a \leq 0 < b$. Given that $b - a > \epsilon/\tau$ and $|b| > |a|$, it results in $b > \epsilon/(2\tau)$. By Lemma 16, the top two eigenvalues of $\Sigma$, denoted as $\lambda_1$ and $\lambda_2$ satisfies

$$\lambda_1 - \lambda_2 \geq C\epsilon^3 \lambda_1 \quad \text{for some constant } C > 0$$

By Lemma 6, with sample size $N_2 = \tilde{O}(d\epsilon_1^{-2})$, with probability at least $1 - \xi$,

$$\|\Sigma - \hat{\Sigma}\|_{\text{op}} \leq \epsilon_1 \|\Sigma\|_{\text{op}}$$

By Lemma 7, we know for the top eigenvector $v$ of $\hat{\Sigma}$ satisfies

$$\sin \theta(e_1, v) \leq \frac{2\|\Sigma - \hat{\Sigma}\|_{\text{op}}}{C\epsilon^3 \lambda_1} \leq \frac{2\epsilon_1 \lambda_1}{C\epsilon^3 \lambda_1} = \frac{2\epsilon_1}{C\epsilon^3}$$

Choose $\epsilon_1 = C\epsilon^3 \delta/2$, and we will get $\sin \theta(e_1, v) \leq \delta$. The sample size we need is $N_2 = \tilde{O}(d\epsilon^{-6}\delta^{-2})$. So with sample size $N = \max(N_1, N_2) = \tilde{O}(d^2\epsilon^{-6}\delta^{-2}\xi^{-1})$, Algorithm 2 can recover $e_1$ within TV distance $\delta$ with probability $1 - \xi$.

$\square$

## B.3 Proofs

**Lemma 15** (Mean Gap). *For $0 \leq a < b$ and $b - a \geq c\epsilon$ for constant $c > 0$, we have*

$$\mathbb{E}_{x \sim \hat{q}} x < -\frac{\psi c^2 \epsilon^3}{2}, \operatorname*{var}_{x \sim \hat{q}} x \leq \frac{1}{2\epsilon}.$$

*Proof.* Since $q_1$ is $\psi$-isoperimetric, $\forall t \in [a, b]$, $q_1(t) \geq \psi \int_t^\infty q_1(x)\,dx \geq \psi\epsilon$. Then we have

$$\int_a^b x q_1(x)\,dx > \psi\epsilon \int_a^b x\,dx > \frac{c^2 \psi \epsilon^3}{2}$$

By the definition of expectation, we have

$$\left| \mathbb{E}_{x \sim \hat{q}} x \right| = \frac{|\int_{\mathbb{R}\setminus[a,b]} x q_1(x)\,dx|}{\int_{\mathbb{R}\setminus[a,b]} q_1(x)\,dx} = \frac{|\int_a^b x q_1(x)\,dx|}{\int_{\mathbb{R}\setminus[a,b]} q_1(x)\,dx} > \frac{c^2 \epsilon^3}{2}$$

On the other hand, we can calculate the variance as follows.

$$\operatorname*{var}_{x \sim \hat{q}} x \leq \mathbb{E}_{x \sim \hat{q}} x^2 = \frac{\int_{\mathbb{R}\setminus[a,b]} x^2 q_1(x)\,dx}{\int_{-\infty}^a q_1(x)\,dx + \int_b^\infty q_1(x)\,dx} \leq \frac{\int_{\mathbb{R}} x^2 q_1(x)\,dx}{2\epsilon} = \frac{1}{2\epsilon}$$

$\square$

**Lemma 17** (Second Moment). *For $a, b$ satisfying $a \leq 0 < b$, $b > c\epsilon$ for constant $c > 0$, we have*

$$\mathbb{E}_{x \sim \hat{q}} x^2 > 1 + C\epsilon^2 \text{ for constant } C > 0.$$

*Proof.* By definition of $\hat{q}$, we know its density on the support $x \in \mathbb{R}\setminus[a,b]$ is

$$\hat{q}(x) = \frac{q_1(x)}{\int_{-\infty}^a q_1(x)\,dx + \int_b^\infty q_1(x)\,dx}$$

Then we calculate its second moment as follows.

$$\mathbb{E}_{x \sim \hat{q}} x^2 = \frac{\int_{-a}^\infty x^2 q_1(x)\,dx + \int_b^\infty x^2 q_1(x)\,dx}{\int_{-a}^\infty q_1(x)\,dx + \int_b^\infty q_1(x)\,dx}$$

Define $g(x) := \int_x^\infty (t^2 - 1)q_1(t)\,dt, x \geq 0$. Its derivative is $g'(x) = (1 - x^2)q_1(x)$. So we know $g(x)$ is monotonically increasing when $x \in [0, 1]$, and decreasing when $x \geq 1$. Since $P_1$ is symmetric and isotropic, we know $\int_0^\infty q_1(x)\,dx = \int_0^\infty x^2 q_1(x)\,dx = 1/2$. So we have $g(0) = 0$. This derives that $g(x) \geq 0, \forall x \geq 0$. In other words, $\int_{-a}^\infty x^2 q_1(x)\,dx \geq \int_{-a}^\infty q_1(x)\,dx$.

For any $x \in [c\epsilon, M]$, we have $g(x) \geq \min(g(c\epsilon), g(M))$. Here we let $M > 0$ such that $\int_M^\infty p(x)\,dx = \epsilon$. Then we can lower bound $g(c\epsilon)$ as follows.

$$g(c\epsilon) = g(0) + \int_0^{c\epsilon} g'(x)\,dx = \int_0^{c\epsilon} (1 - x^2)q_1(x)\,dx > c\epsilon(1 - c^2\epsilon^2)q_1(c\epsilon) > \psi c\epsilon^2(1 - c^2\epsilon^2)$$

If $M \leq 1$, we know $|b| < M \leq 1$. Then we have $g(b) \geq g(c\epsilon)$.

If $M \geq 1 + \epsilon$, we can lower bound $g(M)$ as

$$g(M) > (M^2 - 1) \int_M^\infty q_1(x)\,dx > ((1+\epsilon)^2 - 1)\epsilon = \epsilon^3 + 2\epsilon^2$$

Similarly we will get $g(b) > \min(\psi c\epsilon^2(1 - c^2\epsilon^2), \epsilon^3 + 2\epsilon^2)$. Finally if $1 < M < 1 + \epsilon$, there exists $M' > 0$ such that $\int_M^{M'} q_1(x)\,dx = \epsilon/2$. Here $M' - M > \epsilon/2$. Then we have

$$g(M') > (M'^2 - 1) \int_{M'}^\infty q_1(x)\,dx > ((1 + \epsilon/2)^2 - 1)\epsilon/2 = \epsilon^3/8 + \epsilon^2/2$$

In this case, we have $g(b) > \min(g(\epsilon), g(M')) > \min(\psi c\epsilon^2(1 - c^2\epsilon^2), \epsilon^3/8 + \epsilon^2/2)$. Therefore, we can lower bound the second moment of $\hat{q}$ as follows.

$$\begin{aligned}
\mathop{\mathbb{E}}_{x \sim \hat{q}} x^2 &> \frac{\int_{-a}^\infty q_1(x)\,dx + \int_b^\infty q_1(x)\,dx + g(b)}{\int_{-a}^\infty q_1(x)\,dx + \int_b^\infty q_1(x)\,dx} \\
&> 1 + \frac{\min(\psi c\epsilon^2(1 - c^2\epsilon^2), \epsilon^3/8 + \epsilon^2/2)}{\int_{-a}^\infty q_1(x)\,dx + \int_b^\infty q_1(x)\,dx} \\
&> 1 + \min(\psi c\epsilon^2(1 - c^2\epsilon^2), \epsilon^3/8 + \epsilon^2/2) \\
&> 1 + C\epsilon^3 \text{ where } C = \min(\psi c/2, 1/8)
\end{aligned}$$

$\square$

**Lemma 16** (Spectral Gap of Covariance). *If $a \leq 0 < b$ and $b > c\epsilon$ for constant $c > 0$, then the first and second eigenvalues of the uncentered covariance matrix $\Sigma$ have the following gap*

$$\lambda_1 - \lambda_2 > C\epsilon^3 \lambda_1 \text{ for constant } C > 0.$$

*Furthermore, the top eigenvector corresponds to $u$.*

*Proof.* We assume wlog that $e_1 = u$. Then the marginal distribution of $P$ in $e_1$ is $\hat{q}$, while the marginal distribution in $e_i$ is $q_i$ for any $2 \leq i \leq d$. Since $q_i$ is isotropic, for any $2 \leq i \leq d$, $\mathop{\mathbb{E}}_{x \sim P} x_i^2 = 1$. By Lemma 17, we have $\mathop{\mathbb{E}}_{x \sim P} x_1^2 > 1 + C'\epsilon^3$ for constant $C' > 0$. Let $g(v) := \mathop{\mathbb{E}}_{x \sim P} \frac{v^\top xx^\top v}{v^\top v}$. Then we have $g(e_1) > 1 + C'\epsilon^3$, while $g(e_i) = 1, \forall 2 \leq i \leq d$. Then for any unit vector $v = \sum_{i=1}^d \beta_i e_i$ satisfying $\sum_{i=1}^d \beta_i^2 = 1$, we have

$$g(v) = \sum_{i=1}^d \beta_i^2 g(e_i) \leq g(e_1)$$

Then we know the top eigenvalue of $\Sigma$ is $\lambda_1 > 1 + C\epsilon^3$. Furthermore, the top eigenvector corresponds to $e_1$. Similarly, the second eigenvalue of $\Sigma$ is $\lambda_2 = \max_{v:v \perp e_1} g(v) = g(e_i) = 1, 2 \leq i \leq d$. This implies that $\lambda_1 - \lambda_2 > C\epsilon^3 \lambda_1$ for constant $C > 0$.

$\square$

# C  General Case: Affine Product Distribution with $\epsilon$-Margin

In this section, we first prove the main theorem in Section C.1 using the quantitative lemmas, and then prove all the lemmas. We begin with proving the two qualitative lemmas (Lemma 1 and Lemma 2) in Section C.2, and then prove the quantitative lemmas (Lemma 4 and Lemma 5) in the remaining section. For the quantitative part, we first consider the asymmetric case where $|a + b| \geq \epsilon^5$. In this case, contrastive mean leads to recovering $u$, as elaborated in Section C.3. Secondly, we consider the symmetric case characterized by $a + b = 0$, addressed in Section C.4. We show that we can recover $u$ by calculating the top eigenvector of the re-weighted covariance matrix. Finally we extend the result to near-symmetric case where $|a + b| < \epsilon^5$ in Section C.5.

Recall that we are given data $x^{(1)}, \cdots, x^{(N)}$ drawn from the affine product distribution with $\epsilon$-margin $\widehat{P}$. Algorithm 1 first makes the data isotropic. Denote $y^{(1)}, \cdots, y^{(N)}$ as the corresponding isotropicized data. Then each $y^{(j)}$ is an independent and identically distributed variable drawn from $P = \hat{q} \otimes q \otimes \cdots q$. Since we compute the re-weighted moments on $y^{(j)}$ in the algorithm, we analyze the moments of $P$ directly.

Recall in Definition 1 that $q$ is the symmetric one-dimensional isotropic logconcave density function, and $\tilde{q}$ is the density obtained by restricting $q$ to $\mathbb{R}\backslash[a, b]$ for some unknown $a < b$. Denote $\mu_1, \sigma_1^2$ as the mean and variance of $\tilde{q}$. $\hat{q}$ is the density obtained after making $\tilde{q}$ isotropic, with support $\mathbb{R}\backslash[a', b']$, where $a' = \frac{a-\mu_1}{\sigma_1}, b' = \frac{b-\mu_1}{\sigma_1}$. The density $\hat{q}$ on its support is

$$\hat{q}(x) = \frac{\sigma_1 q(x\sigma_1 + \mu_1)}{\int_{-\infty}^{a} q(x)\,dx + \int_{b}^{\infty} q(x)\,dx}$$

We denote the standard basis of $\mathbb{R}^d$ by $\{e_1, \cdots, e_d\}$, and assume wlog that $e_1 = u$ is the (unknown) normal vector to the band. We write $x_i := \langle x, e_i \rangle$ as $x$'s $i$-th coordinate. We assume in our proof that $|b| > |a|$. If this condition is not met, we can redefine our interval by setting $a' = -b$ and $b' = -a$. The proof can then be applied considering the distribution is restricted to $\{x \in \mathbb{R}^d : u^\top x \leq a' \text{ or } u^\top x \geq b'\}$. For a vector $x \in \mathbb{R}^d$, we use $\|x\|$ to denote its $l_2$ norm. For a matrix $A \in \mathbb{R}^{m \times n}$, we denote its operator norm as $\|A\|_{\text{op}}$.

## C.1  Proof of Theorem 1

Armed with two quantitative lemmas and the Davis-Kahan Theorem, we are now prepared to prove the main theorem. We state the formal version of Theorem 1 as follows.

**Theorem 7** (Sample Complexity for Affine Product Distribution with $\epsilon$-Margin). *Algorithm 1 with $N = \tilde{O}(d^6 \epsilon^{-318} \delta^{-2} \xi^{-1})$ samples learns the target affine product distribution with $\epsilon$-margin to within TV distance $\delta$ with probability $1 - \xi$.*

*Proof.* Firstly we consider the case when $|a + b| \geq \epsilon^5$. Denote $\alpha^* = \arg\max_\alpha \{|(\mu_{\alpha_1})_1|, |(\mu_{\alpha_2})_1|\}$, and $\mu_\alpha = \mu_{\alpha^*}$. By Lemma 4, $|(\mu_\alpha)_1| \geq C_1 \epsilon^{159}/d^2$. Since for any negative $\alpha$, for any $1 \leq i \leq d$,

$$\text{var}(\mu_\alpha)_i \leq \mathop{\mathbb{E}}_{y \sim P} e^{2\alpha^* \|y\|^2} y^2 \leq \mathop{\mathbb{E}}_{y \sim P} y^2 = 1$$

By Chebyshev's Inequality, the re-weighted sample mean $\tilde{\mu} = \frac{1}{N} \sum_{j=1}^{N} e^{\alpha^* \|y^{(j)}\|^2} y^{(j)}$ satisfies

$$\mathbb{P}(|\tilde{\mu}_1| \leq \frac{C_1 \epsilon^{159}}{2d^2}) \leq \frac{4d^4}{NC_1^2 \epsilon^{318}}, \quad \mathbb{P}(|\tilde{\mu}_i| \geq \frac{C_1 \epsilon^{159} \delta}{2d^2 \sqrt{d}}) \leq \frac{4d^5}{NC_1^2 \epsilon^{318} \delta^2}, 2 \leq i \leq d.$$

Let the sample size $N_1 = \frac{4d^6}{C_1^2 \epsilon^{318} \delta^2 \xi} = O(Cd^6 \epsilon^{-318} \delta^{-2} \xi^{-1})$, and we have

$$\mathbb{P}(|\tilde{\mu}_1| > \frac{C_1 \epsilon^{159}}{2d^2}) > 1 - \frac{\xi}{d}, \quad \mathbb{P}(|\tilde{\mu}_i| < \frac{C_1 \epsilon^{159} \delta}{2d^2 \sqrt{d}}) > 1 - \frac{\xi}{d}, 2 \leq i \leq d.$$

So we have

$$\mathbb{P}(\sin \theta(\tilde{\mu}, e_1) \leq \delta) = \mathbb{P}(\frac{\tilde{\mu}_1^2}{\sum_{i=1}^{d} \tilde{\mu}_i^2} \geq 1 - \delta^2)$$

$$\geq \mathbb{P}(|\tilde{\mu}_1| \geq \frac{C_1 \epsilon^{159}}{2d^2}, |\tilde{\mu}_i| \leq \frac{C_1 \epsilon^{159} \delta}{2d^2 \sqrt{d}}, 2 \leq i \leq d)$$

$$\geq 1 - \xi$$

This indicates that with probability $1 - \xi$, the re-weighted mean can output the vector $\tilde{\mu}$ that is within angle $\delta$ to the vector $e_1$.

Secondly, for the case when $a$ and $b$ are near-symmetric. Denote $\Sigma$ as the re-weighted covariance matrix with eigenvalues $\lambda_i$ and $\tilde{\Sigma}$ as the empirical re-weighted covariance matrix with eigenvector $v$. By Lemma 5, $\lambda_1 - \lambda_2 > C_2 \epsilon^3 \lambda_1$. By Lemma 6, with sample size $N_2 = \tilde{O}(d\epsilon^{-6} \delta^{-2} \xi^{-1})$, with probability $1 - \xi$,

$$\|\Sigma - \tilde{\Sigma}\|_{\text{op}} \leq C_2 \epsilon^3 \delta \|\Sigma\|_{\text{op}} / 2$$

By Lemma 7,

$$\sin \theta(e_1, v) \leq \frac{2\|\Sigma - \tilde{\Sigma}\|_{\text{op}}}{C_2 \epsilon^3 \lambda_1} \leq \frac{C_2 \epsilon^3 \delta \lambda_1}{C_2 \epsilon^3 \lambda_1} = \delta$$

So given $N = \max(N_1, N_2) = \tilde{O}(d^6 \epsilon^{318} \delta^{-2} \xi^{-1})$, the algorithm learns the distribution $P$ with probability $1 - \xi$.

$\square$

## C.2 Proofs of Qualitative Bounds

We present proofs of two qualitative lemmas: the contrastive mean (Lemma 1) and the contrastive covariance (Lemma 2). Their quantitative counterparts can be found in Section C.3 and Section C.5. To establish the contrastive mean, we invoke Descartes' Rule of Signs. For the proof concerning contrastive covariance, we introduce a novel monotonicity property on the moment ratio, as described in Lemma 3. We include the complete proof within this section.

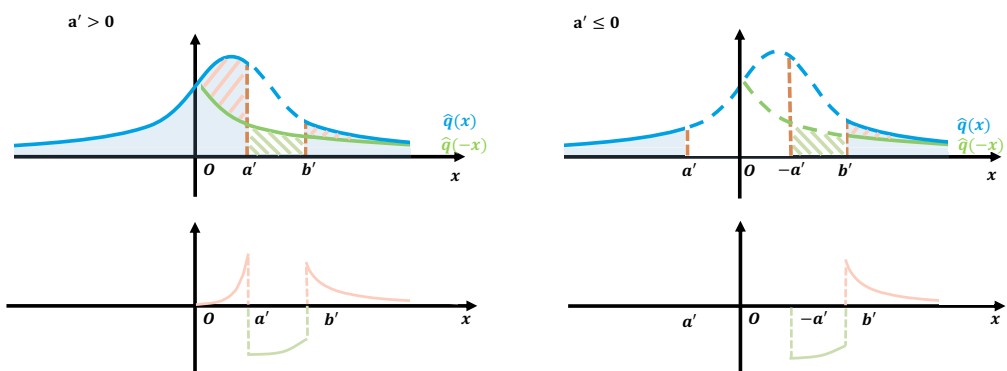

Figure C.1: Coefficients of $F(\alpha)$ ahead of $e^{\alpha x^2}$ term in Lemma 1's proof. By combining $e^{\alpha x^2}$ terms, we flip $\hat{q}(x)$ horizontally. For $a' > 0$, the coefficient is negative when $x \in (a', b')$ and non-negative outside the interval. For $a' \leq 0$, the it is negative when $x \in (-a', b')$ and positive when $x > b'$.

**Contrastive Mean.** We can write the contrastive mean as a linear combination of exponential functions of $\alpha$. By Descartes' rule of signs, the number of zeros of this function is at most two. Since $\alpha = 0$ is one root and corresponds to mean zero, there is at most one nonzero root. And thus we have that for any two distinct nonzero $\alpha$'s, at least one of them achieves nonzero contrastive mean.

**Lemma 1** (Contrastive Mean). *If $|a + b| > 0$, then for any two distinct nonzero $\alpha_1, \alpha_2 \in \mathbb{R}$, at least one of the corresponding re-weighted means is nonzero, i.e.,*

$$\max \left( \left| \mathbb{E}_{x \sim P} e^{\alpha_1 \|x\|^2} u^\top x \right|, \left| \mathbb{E}_{x \sim P} e^{\alpha_2 \|x\|^2} u^\top x \right| \right) > 0.$$

*Proof.* $|b| > |a|$ implies that $\mu_1 < 0$. For any $x \geq 0$, we have

$$\hat{q}(x) = \frac{\sigma_1 q(x\sigma_1 + \mu_1)}{1 - \int_a^b q(x) \, dx} \geq \frac{\sigma_1 q(-x\sigma_1 + \mu_1)}{1 - \int_a^b q(x) \, dx} = \hat{q}(-x).$$

Since $P$ is a product distribution, we have

$$\mathop{\mathbb{E}}_{x \sim P} e^{\alpha \|x\|^2} x_1 = \mathop{\mathbb{E}}_{x_1 \sim \hat{q}} e^{\alpha x_1^2} x_1 \cdot \prod_{i=2}^{d} \mathop{\mathbb{E}}_{x_i \sim q} e^{\alpha x_i^2}$$

We denote

$$F(\alpha) = \mathop{\mathbb{E}}_{x \sim \hat{q}} e^{\alpha x_1^2} x_1 \tag{C.1}$$

By calculation, we have

$$F(\alpha) = \int_{\mathbb{R} \setminus [a', b']} e^{\alpha x^2} x \hat{q}(x) \, dx$$

Then we rearrange $F(\alpha)$ by combining $e^{\alpha x^2}$ as in Figure C.1.

If $a' \leq 0$, we rewrite $F(\alpha)$ as

$$F(\alpha) = -\int_{-a'}^{b'} x \hat{q}(-x) e^{\alpha x^2} \, dx + \int_{b'}^{\infty} x(\hat{q}(x) - \hat{q}(-x)) e^{\alpha x^2} \, dx$$

We treat $F(\alpha)$ as the integral of $a(x)e^{\alpha x^2}$ for $x \geq -a'$. Since $\hat{q}(x) - \hat{q}(-x) > 0$ for $x > b'$, we have $a(x) > 0$ for $x \in [-a', b')$ and $a(x) < 0$ for $x > b'$. In other words, for increasing $x$, the sign of $a(x)$ only changes once. By Theorem 3, $F(\alpha) = 0$ has at most one root.

If $a' > 0$, we arrange $F(\alpha)$ in the same way and get

$$F(\alpha) = \int_0^{a'} x(\hat{q}(x) - \hat{q}(-x)) e^{\alpha x^2} \, dx - \int_{a'}^{b'} x \hat{q}(-x) e^{\alpha x^2} \, dx + \int_{b'}^{\infty} x(\hat{q}(x) - \hat{q}(-x)) e^{\alpha x^2} \, dx$$

Similarly, we treat $F(\alpha)$ as the integral of $a(x)e^{\alpha x^2}$ for $x \geq 0$. For increasing $x$, the sign of $a(x)$ changes twice. By Descartes' rule of signs, $F(\alpha) = 0$ has at most two roots. In addition, we know $F(0) = \mathop{\mathbb{E}}_P x_1 = 0$ by definition of $P$. So $\alpha = 0$ is one root of $F(\alpha) = 0$. So there is at most one nonzero root of $F(\alpha) = 0$. In other words, for any two distinct nonzero $\alpha_1, \alpha_2$, at least one of $F(\alpha_1), F(\alpha_2)$ is nonzero. This implies that

$$\max\left( \left| \mathop{\mathbb{E}}_{x \sim P} e^{\alpha_1 \|x\|^2} x_1 \right|, \left| \mathop{\mathbb{E}}_{x \sim P} e^{\alpha_2 \|x\|^2} x_1 \right| \right) > 0.$$

$\square$

**Moment Ratio.** To prove Lemma 2, we develop a new monotonicity property of the moment ratio of logconcave distributions. Moment ratio is specifically defined as the ratio of the fourth moment to the square of the second moment of truncated versions of the distribution. This measurement essentially reflects the uncentered kurtosis of the distribution. The formal definition is detailed in Definition 5.

**Definition 4** (One-side $t$-restriction distribution). Let $q$ be a distribution in one dimension with nonnegative support. For any $t \geq 0$, define $q_t$ as the one-side $t$-restriction distribution on $q$ obtained by restricting $q$ to $[t, \infty)$.

**Definition 5** (Moment Ratio). Let $q$ be a distribution in one dimension with nonnegative support. For any $t \geq 0$, define $q$'s moment ratio as a function of $t$, given by

$$\mathrm{mr}_q(t) := \frac{\mathrm{var}_{q_t}(X^2)}{(\mathbb{E}_{q_t} X^2)^2}, \quad \text{where } q_t \text{ is the one-side } t\text{-restriction distribution on } q.$$

We will prove the monotonicity of the moment ratio (Lemma 3) by reducing general logconcave distributions to exponential distributions. The monotonicity of the moment ratio for exponential distribution is detailed in Lemma 18.

**Lemma 18** (Monotonicity of Moment Ratio of Exponential Distribution). *Define* $h(x) = \beta e^{-\gamma x}, x \geq 0, \beta, \gamma > 0$. *Denote* $N_k(t) = \int_t^{\infty} x^k h(x) \, dx$. *Then for any* $t \geq 0$, *we have*

$$t^4 N_0(t) N_2(t) + N_2(t) N_4(t) - 2t^2 N_0(t) N_4(t) > 0.$$

*Proof.* By calculation, we have

$$N_0(t) = \int_t^\infty \beta e^{-\gamma x} \, dx = \frac{\beta}{\gamma} e^{-\gamma t}$$

$$N_1(t) = \int_t^\infty \beta x e^{-\gamma x} \, dx = \frac{\beta}{\gamma} t e^{-\gamma t} + \frac{1}{\gamma} N_0(t) = \left(t + \frac{1}{\gamma}\right) \frac{\beta}{\gamma} e^{-\gamma t}$$

$$N_2(t) = \int_t^\infty \beta x^2 e^{-\gamma x} \, dx = \frac{\beta}{\gamma} t^2 e^{-\gamma t} + \frac{2}{\gamma} N_1(t) = \left(t^2 + \frac{2}{\gamma} t + \frac{2}{\gamma^2}\right) \frac{\beta}{\gamma} e^{-\gamma t}$$

$$N_3(t) = \int_t^\infty \beta x^3 e^{-\gamma x} \, dx = \frac{\beta}{\gamma} t^3 e^{-\gamma t} + \frac{3}{\gamma} N_2(t) = \left(t^3 + \frac{3}{\gamma} t^2 + \frac{6}{\gamma^2} t + \frac{6}{\gamma^3}\right) \frac{\beta}{\gamma} e^{-\gamma t}$$

$$N_4(t) = \int_t^\infty \beta x^4 e^{-\gamma x} \, dx = \frac{\beta}{\gamma} t^4 e^{-\gamma t} + \frac{4}{\gamma} N_3(t) = \left(t^4 + \frac{4}{\gamma} t^3 + \frac{12}{\gamma^2} t^2 + \frac{24}{\gamma^3} t + \frac{24}{\gamma^4}\right) \frac{\beta}{\gamma} e^{-\gamma t}$$

Then we can plug them and get

$$t^4 N_0(t) N_2(t) + N_2(t) N_4(t) - 2t^2 N_0(t) N_4(t)$$
$$= \frac{\beta^2}{\gamma^2} e^{-2\gamma t} \left(t^4 \left(t^2 + \frac{2}{\gamma} t + \frac{2}{\gamma^2}\right) + \left(-t^2 + \frac{2}{\gamma} t + \frac{2}{\gamma^2}\right) \cdot \left(t^4 + \frac{4}{\gamma} t^3 + \frac{12}{\gamma^2} t^2 + \frac{24}{\gamma^3} t + \frac{24}{\gamma^4}\right)\right)$$
$$= \frac{8\beta^2}{\gamma^2} e^{-2\gamma t} \left(\frac{t^3}{\gamma^3} + \frac{6t^2}{\gamma^4} + \frac{12t}{\gamma^5} + \frac{6}{\gamma^6}\right) > 0$$

$\square$

Next, we will prove the monotonicity of moment ratio for logconcave distributions.

**Lemma 3** (Monotonicity of Moment Ratio). *Let $q$ be a logconcave distribution in one dimension with nonnegative support. For any $t \geq 0$, let $q_t$ be the distribution obtained by restricting $q$ to $[t, \infty)$. Then the moment ratio of $q$, defined as $mr_q(t) := \frac{\mathrm{var}_{q_t}(X^2)}{(\mathbb{E}_{q_t} X^2)^2}$, is strictly decreasing with $t$.*

*Proof.* Denote $M_k(t) = \int_t^\infty x^k q(x) \, dx$. By Definition 5,

$$mr_q(t) = \frac{\mathrm{var}_{q_t}(X^2)}{(\mathbb{E}_{q_t} X^2)^2} = \frac{\frac{\int_t^\infty x^4 q(x) \, dx}{\int_t^\infty q(x) \, dx}}{\left(\frac{\int_t^\infty x^2 q(x) \, dx}{\int_t^\infty q(x) \, dx}\right)^2} - 1 = \frac{M_0(t) M_4(t)}{M_2(t)^2} - 1$$

Next we will show that $mr'(t) < 0, \forall t \geq 0$. By taking the derivative,

$$mr'(t) = \frac{-q(t)}{M_2(t)} (t^4 M_0(t) M_2(t) + M_4(t) M_2(t) - 2t^2 M_0(t) M_4(t))$$

Define $H(t) = t^4 M_0(t) M_2(t) + M_4(t) M_2(t) - 2t^2 M_0(t) M_4(t)$. We will show that $H(t) > 0, \forall t \geq 0$. Clearly $H(0) = M_4(0) M_2(0) > 0$, So we only consider $t > 0$ in the following proof.

Let $h(x) = \beta e^{-\gamma x}$ be an exponential function ($\beta, \gamma > 0$) such that

$$M_0(t) = N_0(t), M_2(t) = N_2(t), \text{ where } N_k(t) = \int_t^\infty x^k h(x) \, dx, k \in \mathbb{N}.$$

Then we have

$$\int_t^\infty (h(x) - q(x)) \, dx = 0, \int_t^\infty x^2 (h(x) - q(x)) = 0$$

By the logconcavity of $q$, the graph of $h$ intersects with the graph of $q$ at exactly two points $u' < v$, where $v > 0$. Also we have $h(x) \leq q(x)$ at the interval $[u', v]$ and $h(x) > q(x)$ outside the interval.

Let $u = \max\{0, u'\}$. So for $x \geq 0$, $(x - u)(x - v)$ has the same sign as $h(x) - q(x)$. Since $t \geq 0$, we have

$$\int_t^\infty (x^2 - u^2)(x^2 - v^2)(h(x) - q(x)) \geq 0$$

Expanding and we get

$$\int_t^\infty x^4(h(x) - q(x)) \geq (u^2 + v^2) \int_t^\infty x^2(h(x) - q(x))\, dx - u^2 v^2 \int_t^\infty (h(x) - q(x))\, dx = 0$$

This shows that $N_4(t) \geq M_4(t)$. To show that $H(t) > 0$, we consider two cases.

Firstly if $M_2(t) - 2t^2 \geq 0$, we have

$$H(t) = t^4 M_0(t) M_2(t) + M_4(t)(M_2(t) - 2t^2) > 0.$$

Secondly if $M_2(t) - 2t^2 < 0$, by calculation of the exponential function's moments (Lemma 18), we have

$$t^4 N_0(t) N_2(t) > -N_4(t)(N_2(t) - 2t^2)$$

Then we have

$$\begin{aligned}
H(t) =& t^4 M_0(t) M_2(t) + M_4(t)(M_2(t) - 2t^2) \\
=& t^4 N_0(t) N_2(t) + M_4(t)(M_2(t) - 2t^2) \\
\geq& -N_4(t)(N_2(t) - 2t^2) + M_4(t)(M_2(t) - 2t^2) \\
=& (M_2(t) - 2t^2)(M_4(t) - N_4(t)) \\
\geq& 0
\end{aligned}$$

The equality holds if and only if $M_4(t) = N_4(t)$. This implies that

$$H(t) = t^4 N_0(t) N_2(t) + N_4(t)(N_2(t) - 2t^2) > 0.$$

Combining both cases, $\mathrm{mr}'(t) < 0, \forall t \geq 0$, which implies that the moment ratio of $q$ is strictly decreasing with respect to $t$. $\qquad\square$

**Contrastive Covariance.** View the spectral gap of the re-weighted covariance, denoted as $\lambda_1(\tilde{\Sigma}) - \lambda_2(\tilde{\Sigma})$, as a function of $\alpha$, which we denote as $S(\alpha)$. By calculation, $S(0) = 0$ and $S'(0)$ is proportional to $\mathrm{mr}(b) - \mathrm{mr}(0)$, which is negative by the monotonicity property of moment ratio. Then we can prove Lemma 2 using Taylor expansion.

**Lemma 2** (Contrastive Covariance). *If $a + b = 0$, then there exists an $\alpha < 0$, such that (1) there is a positive gap between the top two eigenvalues of the re-weighted uncentered covariance matrix $\tilde{\Sigma} = \mathbb{E}_{x \sim P}\, e^{\alpha \|x\|^2}(xx^\top)$. That is, $\lambda_1(\tilde{\Sigma}) > \lambda_2(\tilde{\Sigma})$. (2) The top eigenvector of $\tilde{\Sigma}$ is $u$.*

*Proof.* Denote $M_k(t) = \int_t^\infty x^k q(x)\, dx$. The variance $\sigma_1^2$ of $q$ restricted to $\mathbb{R} \backslash [-b, b]$ is

$$\sigma_1^2 = \frac{\int_b^\infty x^2 p(x)\, dx}{\int_b^\infty p(x)\, dx} = \frac{M_2(b)}{M_0(b)}$$

Since $\hat{q}$ is isotropic, the density on the support $\mathbb{R} \backslash [-b/\sigma_1, b/\sigma_1]$ is

$$\mathbb{P}_{\hat{q}}(x) = \frac{\sigma_1 q(x\sigma_1)}{2 \int_b^\infty q(x)\, dx}$$

Let

$$S(\alpha) := \mathbb{E}_{x \sim \hat{q}} e^{\alpha x^2} x^2 \, \mathbb{E}_{x \sim q} e^{\alpha x^2} - \mathbb{E}_{x \sim q} e^{\alpha x^2} x^2 \, \mathbb{E}_{x \sim \hat{q}} e^{\alpha x^2}$$

Since $q$ and $\hat{q}$ are both isotropic, $S(0) = \mathbb{E}_{x \sim \hat{q}} x^2 - \mathbb{E}_{x \sim q} x^2 = 0$. Then,

$$S'(0) = \frac{M_4(b) M_0(b)}{M_2^2(b)} - \frac{M_4(0) M_0(0)}{M_2^2(0)}$$

The last step is because the distribution $q$ is isotropic. By Lemma 3, we know $S'(0) = \mathrm{mr}'(0) < 0$. On the other hand, $\forall \alpha \leq 0$, $S''(\alpha)$ can be bounded.

$$S''(\alpha) \leq \frac{M_6(b)}{M_0(b)M_2^3(b)} + \frac{M_4(b)}{M_0(b)M_2^2(b)} < \mathrm{poly}(1/\epsilon)$$

By Taylor expansion, we know there exists $\alpha < 0$ such that

$$S(\alpha) = S(0) + \alpha S'(0) + \frac{\alpha^2}{2} S''(\alpha') > 0, \text{ where } \alpha' \in [\alpha, 0]$$

Then we have for $2 \leq j \leq d$,

$$\mathbb{E}_{x \sim P} e^{\alpha\|x\|^2} x_1^2 - \mathbb{E}_{x \sim P} e^{\alpha\|x\|^2} x_j^2 = S(\alpha) \left( \mathbb{E}_{x \sim q} e^{\alpha x^2} \right)^{d-2} > 0$$

For any $v \in \mathbb{R}^d$, define $\phi(v)$ as

$$\phi(v) := \mathbb{E}_{x \sim P} \frac{e^{\alpha\|x\|^2} v^\top x x^\top v}{v^\top v}$$

Substituting $v$ with $e_1$ and $e_j$, $2 \leq j \leq d$, we have

$$\phi(e_1) = \mathbb{E}_{x \sim P} e^{\alpha\|x\|^2} x_1^2, \phi(e_j) = \mathbb{E}_{x \sim P} e^{\alpha\|x\|^2} x_j^2$$

This implies that for $2 \leq j \leq d$,

$$\phi(e_1) - \phi(e_j) = \mathbb{E}_{x \sim P} e^{\alpha\|x\|^2} x_1^2 - \mathbb{E}_{x \sim P} e^{\alpha\|x\|^2} x_j^2 > 0$$

For any vector $v = \sum_{i=1}^d \gamma_i e_i$, we have

$$\phi(v) = \frac{1}{\sum_{i=1}^d \gamma_i^2} \mathbb{E} \, e^{\alpha\|x\|^2} (\sum_{i=1}^d \gamma_i x_i)^2 = \frac{\sum_{i=1}^d \gamma_i^2 \phi(e_i)}{\sum_{i=1}^d \gamma_i^2} \leq \phi(e_1)$$

This shows that the top eigenvalue of $\tilde{\Sigma}$ is $\lambda_1(\tilde{\Sigma}) = \max_v \phi(v) = \phi(e_1)$. Similarly, $\lambda_2(\tilde{\Sigma}) = \max_{v:v \perp e_1} \phi(v) = \phi(e_j), 2 \leq j \leq d$. Therefore, $\lambda_1(\tilde{\Sigma}) > \lambda_2(\tilde{\Sigma})$ and the top eigenvector is $e_1$, which is essentially $u$. $\square$

### C.3 Quantitative Bounds for Contrastive Mean

We will prove Lemma 4 in this section. Here we consider the case when $|a + b| \geq \epsilon^5$. We compute the contrastive mean of $P$ given $\alpha < 0$ as $\mathbb{E}_{x \sim P} e^{\alpha x^2} x$ using two different $\alpha$'s.

**Definition 6.** We define $F(\alpha)$ as re-weighted mean for the one-dimensional distribution $\hat{q}$.

$$F(\alpha) = \mathbb{E}_{x \sim \hat{q}} e^{\alpha x^2} x = \int_{\mathbb{R} \setminus [a', b']} e^{\alpha x^2} x \hat{q}(x) \, dx \tag{C.2}$$

Since $P$ is isotropic, $F(0) = \mathbb{E}_{x \sim \hat{q}} x = 0$.

To prove Lemma 4, we need to show that for given $\alpha_1, \alpha_2$, the maximum of $|F(\alpha_1)|, |F(\alpha_2)|$ exceeds a certain positive threshold. We follow the same idea of bounding the number of roots of $F(\alpha)$ as in the qualitative lemma (Lemma 1). By taking the derivative of $F(\alpha)$, we can show that either $F'(0) \neq 0$ or $F''(0) \neq 0$. Then by Taylor expansion, we can choose two distinct $\alpha$'s (near zero) so that one of the corresponding contrastive means is bounded away from zero.

In the process of proving the quantitative bounds, similar to our approach with qualitative bounds, we must consider two distinct scenarios based on the sign of $a'$, as illustrated in Figure 4.1.

- In the case where $a'$ is negative, Lemma 20 asserts that the first derivative of $F$ at zero, $F'(0)$, is always positive.

- Conversely, when $a'$ is nonnegative, Lemma 23 reveals an essential characteristic of the function $F(\alpha)$: it's not possible for both $F'(0)$ and $F''(0)$ to be zero at the same time.

- Lemma 24 provides upper bounds for the derivatives of $F(\alpha)$. These upper bounds are crucial as they help in managing the extra terms that emerge during the Taylor expansion of $F(\alpha)$.

- The section concludes with the proof of Lemma 4, which is the quantitative lemma for the contrastive mean.

We start with Lemma 19 showing that $|\mu_1|$ is away from zero provided that $|a + b|$ is also different from zero.

**Lemma 19** (Lower Bound of $|\mu_1|$). *If $|a + b| \geq \epsilon^s$ for $s \geq 2$, then $|\mu_1| \geq \epsilon^s/2e$.*

*Proof.* Firstly, let's consider the case when $a \leq 0$. By Lemma 9, we know $q(x)$ is upper bounded by 1. Since $|a + b| \geq \epsilon^s$ and $q$ is logconcave, by Lemma 8, the mean of the density restricted $q$ in $[-a, b]$ satisfies $\mu_{[-a,b]} \geq 1/e$. Then

$$|\mu_1| = \frac{\mu_{[-a,b]} \int_{-a}^{b} q(x)\,dx}{1 - \int_{a}^{b} q(x)\,dx} \geq \frac{1}{e} \frac{\int_{-a}^{b} q(x)\,dx}{\int_{-a}^{b} q(x)\,dx + 2\int_{b}^{\infty} q(x)\,dx}$$

By Lemma 11,

$$q(b) \geq 2 \int_{b}^{\infty} q(x)\,dx$$

Since $|a| < |b|$, we have

$$\int_{-a}^{b} q(x)\,dx \geq (b + a)q(b) \geq 2(b + a) \int_{b}^{\infty} q(x)\,dx$$

So we have

$$|\mu_1| \geq \frac{1}{e} \frac{2(b + a)}{2(b + a) + 2} \geq \frac{1}{e} \frac{\epsilon^s}{1 + \epsilon^s} > \frac{\epsilon^s}{2e}$$

Secondly, when $a > 0$, since $b - a > \epsilon$,

$$|\mu_1| = \frac{\mu_{[a,b]} \int_{a}^{b} q(x)\,dx}{1 - \int_{a}^{b} q(x)\,dx} > (a + \frac{\epsilon}{e}) \frac{\epsilon}{1 - \epsilon} > \frac{\epsilon^2}{e}$$

$\square$

**Lemma 20** (Derivative of $F(0)$ when $a' < 0$). *If $|a + b| \geq \epsilon^s$ for $s \geq 2$ and $a' < 0$, then $F'(0) > \epsilon^{3s+3.5}/2$.*

*Proof.* We rearrange $F(\alpha)$ by combining terms with same $e^{\alpha x^2}$ as in Figure 4.1, and get

$$F(\alpha) = -\int_{-a'}^{b'} x\hat{q}(-x)e^{\alpha x^2}\,dx + \int_{b'}^{\infty} x(\hat{q}(x) - \hat{q}(-x))e^{\alpha x^2}\,dx$$

Define $r(x) = \begin{cases} -\hat{q}(-x) & x \in [-a', b'] \\ \hat{q}(x) - \hat{q}(-x) & x \in [b', \infty) \end{cases}$. Then we have

$$F(\alpha) = \int_{-a'}^{\infty} xr(x)e^{\alpha x^2}\,dx$$

By calculating the derivative of $F(\alpha)$, we have

$$F'(\alpha) = \int_{-a'}^{\infty} x^3 r(x)e^{\alpha x^2}\,dx$$

$$= \int_{-a'}^{\infty} x(x^2 - b'^2)r(x)e^{\alpha x^2}\,dx + b'^2 \int_{-a'}^{\infty} xr(x)e^{\alpha x^2}\,dx$$

$$= \int_{-a'}^{\infty} x(x^2 - b'^2)r(x)e^{\alpha x^2}\, dx + b'^2 F(\alpha)$$

Since $r(x)$ is nonnegative for $x \geq b'$ and negative otherwise, then for any $x \geq -a'$, we have

$$x(x^2 - b'^2)r(x)e^{\alpha x^2} \geq 0$$

Since $F(0) = 0$, we have

$$F'(0) = \int_{-a'}^{\infty} x(x^2 - b'^2)r(x)\, dx \geq \int_{-a'}^{b'} x(b'^2 - x^2)\hat{q}(-x)\, dx$$

By calculation,

$$F'(0) \geq \frac{1}{1 - \int_a^b q(x)\, dx} \int_{-\frac{a-\mu_1}{\sigma_1}}^{\frac{b-\mu_1}{\sigma_1}} x\left(\left(\frac{b-\mu_1}{\sigma_1}\right)^2 - x^2\right) \sigma_1 q(-x\sigma_1 + \mu_1)\, dx$$

$$\geq \frac{1}{\sigma_1^3(1 - \int_a^b q(x)\, dx)} \int_{-a}^{b-2\mu_1} (x + \mu_1)\left((b-\mu_1)^2 - (x+\mu_1)^2\right) q(x)\, dx$$

$$\geq \frac{1}{\sigma_1^3(1 - \int_a^b q(x)\, dx)} \int_{-a}^{b} (x + \mu_1)(x + b)(b - x - 2\mu_1)q(x)\, dx$$

$$\geq \frac{1}{\sigma_1^3(1 - \int_a^b q(x)\, dx)}(b - a)2|\mu_1| \int_{-a}^{b} (x + \mu_1)q(x)\, dx$$

Choose $t_0 \in [-a, b]$ such that $\int_{-a}^{t_0} q(x)\, dx = \int_{t_0}^{b} q(x)\, dx$. Since $q(x)$ is bounded by 1 by Lemma 9, we have

$$t_0 + a \geq \frac{\int_{-a}^{b} q(x)\, dx}{2}$$

On the other hand, similar to the proof of Lemma 19, we have

$$\int_{-a}^{b} q(x)\, dx \geq 2(b + a) \int_b^{\infty} q(x)\, dx \geq 2\epsilon^{s+1}$$

So we have

$$\int_{-a}^{b} (x + \mu_1)q(x)\, dx \geq \int_{t_0}^{b} (x + a)q(x)\, dx \geq (t_0 + a)\frac{1}{2}\int_{-a}^{b} q(x)\, dx$$

$$\geq \frac{1}{4}\left(\int_{-a}^{b} q(x)\, dx\right)^2 \geq \epsilon^{2s+2}$$

By definition, we have

$$\sigma_1^2 \leq \frac{\int_{\mathbb{R}\setminus[a,b]} x^2 q(x)\, dx}{1 - \int_a^b q(x)\, dx} \leq \frac{\mathbb{E}_{x\sim q} x^2}{1 - \int_a^b q(x)\, dx} \leq \frac{1}{1 - \int_a^b q(x)\, dx}$$

Applying Lemma 19, we know $|\mu_1| > \epsilon^s/2e$. Using these results to estimate $F'(0)$, we get

$$F'(0) \geq (1 - \int_a^b q(x)\, dx)^{0.5} \cdot \epsilon \cdot \frac{\epsilon^s}{e}\epsilon^{2s+2} > \frac{\epsilon^{3s+3.5}}{2}$$

$\square$

**Lemma 21.** *If $a \geq \mu_1, \mu_1 \leq 0$, then we have $\int_{a-2\mu_1}^{b} xq(x)\, dx \geq |\mu_1| \int_b^{\infty} q(x)\, dx$.*

*Proof.* Firstly we will show that $\int_{2\mu_1-a}^{a} (x - \mu_1)q(x)\, dx \geq 0$. For $x \in [\mu_1, a]$, we have $(2\mu_1 - x)^2 - x^2 = 4\mu_1(\mu_1 - x) \geq 0$. Since $q(x)$ is symmetric and uni-modal, we have $q(2\mu_1 - x) \leq q(x), \forall x \in [\mu_1, a]$.

$$\int_{2\mu_1-a}^{a} (x - \mu_1)q(x)\, dx = \int_{2\mu_1-a}^{\mu} (x - \mu_1)q(x)\, dx + \int_{\mu_1}^{a} (x - \mu_1)q(x)\, dx$$

$$= \int_{\mu_1}^{a} (\mu_1 - x)q(2\mu_1 - x)\, dx + \int_{\mu_1}^{a} (x - \mu_1)q(x)\, dx$$

$$= \int_{\mu_1}^{a} (x - \mu_1)(q(x) - q(2\mu_1 - x))\, dx$$

$$\geq 0$$

And then, we have

$$\int_{(-\infty, 2\mu_1 - b] \cup [b - 2\mu_1, \infty)} (x - \mu_1)q(x)\, dx = -\mu_1 \int_{(-\infty, 2\mu_1 - b] \cup [b - 2\mu_1, \infty)} q(x)\, dx$$

$$= -2\mu_1 \int_{b - 2\mu_1}^{\infty} q(x)\, dx$$

Since $\mu_1$ is the mean of the distribution $\tilde{q}$. We have

$$\int_{\mathbb{R} \setminus [a,b]} (x - \mu_1)q(x)\, dx = 0$$

Then we know

$$\int_{2\mu_1 - b}^{2\mu_1 - a} (x - \mu_1)q(x)\, dx + \int_{b}^{b - 2\mu_1} (x - \mu_1)q(x)\, dx \leq 2\mu_1 \int_{b - 2\mu_1}^{\infty} q(x)\, dx$$

Then we have

$$0 \geq \int_{2\mu_1 - b}^{2\mu_1 - a} (x - \mu_1)q(x)\, dx + \int_{b}^{b - 2\mu_1} (x - \mu_1)q(x)\, dx - 2\mu_1 \int_{b - 2\mu_1}^{\infty} q(x)\, dx$$

$$= - \int_{a - 2\mu_1}^{b - 2\mu_1} xq(x)\, dx - \mu_1 \int_{a - 2\mu_1}^{b - 2\mu_1} q(x)\, dx + \int_{b}^{b - 2\mu_1} xq(x)\, dx - \mu_1 \int_{b}^{b - 2\mu_1} q(x)\, dx$$

$$- 2\mu_1 \int_{b - 2\mu_1}^{\infty} q(x)\, dx$$

$$= - \int_{a - 2\mu_1}^{b - 2\mu_1} xq(x)\, dx - \mu_1 \int_{a - 2\mu_1}^{\infty} q(x)\, dx + \int_{b}^{b - 2\mu_1} xq(x)\, dx - \mu_1 \int_{b}^{\infty} q(x)\, dx$$

This derives that

$$\int_{a - 2\mu_1}^{b} xq(x)\, dx \geq -\mu_1 \int_{a - 2\mu_1}^{\infty} q(x)\, dx - \mu_1 \int_{b}^{\infty} q(x)\, dx$$

$$\geq -\mu_1 \int_{b}^{\infty} q(x)\, dx$$

$\square$

**Lemma 22** (Second Derivative of $F(0)$ when $a' \geq 0$). *If $a' \geq 0$, we define the following functions,*

$$r(x) = \begin{cases} \hat{q}(x) - \hat{q}(-x) & x \in [0, a'] \cup [b', \infty) \\ -\hat{q}(-x) & x \in (a', b') \end{cases}, H(\alpha) = \int_{0}^{\infty} x(x^2 - a'^2)(x^2 - b'^2)r(x)e^{\alpha x^2}\, dx.$$

*Then we have*

$$F''(\alpha) = H(\alpha) + (a'^2 + b'^2)F'(\alpha) + a'^2 b'^2 F(\alpha).$$

*Proof.* We rearrange $F(\alpha)$ by combining terms with same $e^{\alpha x^2}$ as in Figure 4.1, and get

$$F(\alpha) = \int_{0}^{a'} x(\hat{q}(x) - \hat{q}(-x))e^{\alpha x^2}\, dx - \int_{a'}^{b'} x\hat{q}(-x)e^{\alpha x^2}\, dx + \int_{b'}^{\infty} x(\hat{q}(x) - \hat{q}(-x))e^{\alpha x^2}\, dx$$

By the definition of $r(x)$, we naturally have

$$F(\alpha) = \int_{0}^{\infty} xr(x)e^{\alpha x^2}\, dx$$

Then we can calculate its first and second derivative as follows.

$$F'(\alpha) = \int_0^\infty x^3 r(x) e^{\alpha x^2} \, dx, \quad F''(\alpha) = \int_0^\infty x^5 r(x) e^{\alpha x^2} \, dx$$

By the definition of $H(\alpha)$, we have

$$H(\alpha) = \int_0^\infty x^5 r(x) e^{\alpha x^2} \, dx - (a'^2 + b'^2) \int_0^\infty x^3 r(x) e^{\alpha x^2} \, dx + a'^2 b'^2 \int_0^\infty x r(x) e^{\alpha x^2} \, dx$$

$$= F''(\alpha) - (a'^2 + b'^2) F'(\alpha) + a'^2 b'^2 F(\alpha)$$

□

**Lemma 23** (First and Second Derivatives of $F(0)$ when $a' \geq 0$). *If $|a + b| \geq \epsilon^s$ for $s \geq 2$ and $a' \geq 0$, then we have either $F'(0) < -\frac{C_2 \epsilon^{6s+6.5}}{\log^4(1/\epsilon)}$ or $F''(0) > \frac{C_3 \epsilon^{6s+4.5}}{\log^4(1/\epsilon)}$ for constants $C_2, C_3 > 0$.*

*Proof.* We prove the lemma by showing that $H(0) > \frac{C_1 \epsilon^{6s+4.5}}{\log^5(1/\epsilon)}$. We calculate $H(0)$ as follows.

$$H(0) = \int_0^\infty x(x^2 - a'^2)(x^2 - b'^2) r(x) \, dx$$

$$\geq \int_{a'}^{b'} x(x^2 - a'^2)(b'^2 - x^2) \hat{q}(-x) \, dx$$

$$= \int_{a'}^{b'} x(x^2 - a'^2)(b'^2 - x^2) \frac{\sigma_1 q(x\sigma_1 - \mu_1)}{1 - \int_a^b q(x) \, dx} \, dx$$

$$= \frac{\int_{a-2\mu_1}^{b-2\mu_1} (x + \mu_1)\left((x + \mu_1)^2 - (a - \mu_1)^2\right)\left((b - \mu_1)^2 - (x + \mu_1)^2\right) q(x) \, dx}{\sigma_1^3 \left(1 - \int_a^b q(x) \, dx\right)}$$

$$= \frac{\int_{a-2\mu_1}^{b-2\mu_1} (x + \mu_1)(x + a)(x - a + 2\mu_1)(b + x)(b - x - 2\mu_1) q(x) \, dx}{\sigma_1^3 \left(1 - \int_a^b q(x) \, dx\right)}$$

Denote $\rho := \int_{a-2\mu_1}^b q(x) \, dx$, by Lemma 21 and the bound of $b$, we know

$$\rho = \int_{a-2\mu_1}^b q(x) \, dx \geq \frac{\epsilon^s/2e \cdot \epsilon}{1 + \ln(1/\epsilon)} \geq \frac{\epsilon^{s+1}}{6 \ln(1/\epsilon)}$$

Choose $t_1 < t_2$ such that $\int_{a-2\mu_1}^{t_1} q(x) \, dx = \int_{t_1}^{t_2} q(x) \, dx = \int_{t_2}^b q(x) \, dx$. Since $q(x)$ is upper bounded by 1 by Lemma 9, we have

$$t_1 - (a - 2\mu_1) \geq \frac{\rho}{3}, b - t_2 \geq \frac{\rho}{3}.$$

Using this, we can bound

$$\int_{t_1}^{t_2} (x + \mu_1)(x + a)(x - a + 2\mu_1)(b + x)(b - x - 2\mu_1) q(x) \, dx$$

$$\geq |\mu_1|^2 \frac{\rho}{3} \frac{\rho}{3} \frac{\rho}{3} \frac{\rho}{3} \geq \frac{C_1 \epsilon^{6s+4}}{\ln^4(1/\epsilon)} \text{ for some constant } C_1 > 0$$

We have shown that $\sigma_1 \leq 1/\sqrt{1 - \int_a^b q(x) \, dx}$. Combining all results, we can compute $H(0)$ as

$$H(0) > \sqrt{1 - \int_a^b q(x) \, dx} \frac{C_1 \epsilon^{6s+4}}{\log^4(1/\epsilon)} > \frac{C_1 \epsilon^{6s+4.5}}{\log^4(1/\epsilon)}$$

Given $F(0) = 0$, by Lemma 22,

$$F''(0) = H(0) + (a'^2 + b'^2) F'(0)$$

Since the distribution is symmetric, we know

$$|\mu_1| = \frac{|\int_{|a|}^{b} xq(x)\,dx|}{1 - \int_a^b q(x)\,dx} \leq \frac{\mathbb{E}_{x\sim q}|x|}{\epsilon} \leq \frac{\sqrt{\mathbb{E}_{x\sim q} x^2}}{\epsilon} = \frac{1}{\epsilon}$$

So we know $0 \leq a' < b' = b - 2\mu_1 < 2.5/\epsilon$, thus $(a'^2 + b'^2)^2 < 25/(2\epsilon^2)$. Thus we have either $F'(0) < -\frac{C_2\epsilon^{6s+6.5}}{\log^4(1/\epsilon)}$ or $F''(0) > \frac{C_3\epsilon^{6s+4.5}}{\log^4(1/\epsilon)}$ for constants $C_2, C_3 > 0$.

$\square$

**Lemma 24** (Upper Bound of $F$'s derivatives). *For $\alpha \leq 0$, the derivatives of $F(\alpha)$ are bounded as*

$$|F'(\alpha)| \leq C_4/\epsilon^3, |F''(\alpha)| \leq C_5/\epsilon^5, |F'''(\alpha)| \leq C_6/\epsilon^7 \quad \text{for constants } C_4, C_5, C_6 > 0.$$

*Proof.* Define $M_k = \mathbb{E}_{x\sim q}|x|^k$. By Cauchy-Schwarz Inequality, $M_1 \leq \sqrt{M_2 M_0} = 1$. By Lemma 12,

$$M_k \leq (2k)^k (M_1)^k \leq (2k)^k$$

Also we have proved that $|\mu_1| \leq 1/\epsilon$. By definition of $F(\alpha)$, we calculate its first derivative as follows.

$$\begin{aligned}
|F'(\alpha)| &\leq \int_{-\infty}^{\infty} q(x)|x - \mu_1|^3 e^{\alpha(x-\mu)^2}\,dx \\
&\leq \int_{-\infty}^{\infty} q(x)|x - \mu_1|^3\,dx \\
&\leq \int_{-\infty}^{\infty} q(x)(|x|^3 - 3\mu_1 x^2 + 3\mu_1^2|x| - \mu_1^3)\,dx \\
&= M_3 - 3\mu_1 M_2 + 3\mu_1^2 M_1 - \mu_1^3 \\
&\leq C_4/\epsilon^3 \quad \text{for some constant } C_4 > 0
\end{aligned}$$

Similarly, we calculate its second and third derivatives as follows.

$$\begin{aligned}
|F''(\alpha)| &\leq \int_{-\infty}^{\infty} q(x)|x - \mu_1|^5\,dx \\
&\leq M_5 - 5\mu_1 M_4 + 10\mu_1^2 M_3 - 10\mu_1^3 M_2 + 5\mu_1^4 M_1 - \mu^5 \\
&\leq C_5/\epsilon^5 \quad \text{for some constant } C_5 > 0 \\
|F'''(\alpha)| &\leq \int_{-\infty}^{\infty} q(x)|x - \mu_1|^7\,dx \\
&= M_7 - 7\mu_1 M_6 + 21\mu_1^2 M_5 - 35\mu_1^3 M_4 + 35\mu_1^4 M_3 - 21\mu_1^5 M_2 + 7\mu_1^6 M_1 - \mu_1^7 \\
&\leq C_6/\epsilon^7 \quad \text{for some constant } C_6 > 0
\end{aligned}$$

$\square$

Now we are ready to prove Lemma 4.

**Lemma 4** (Quantitative Gap of Contrastive Mean). *Suppose that $|a + b| \geq \epsilon^5$. Then, for $\alpha_1 = -c_1\epsilon^{82}/d, \alpha_2 = -c_2\epsilon^{42}/d$, the re-weighted mean of $P$, denoted as $\mu_{\alpha_1}$ and $\mu_{\alpha_2}$, satisfies*

$$\max\left(\left|u^\top \mu_{\alpha_1}\right|, \left|u^\top \mu_{\alpha_2}\right|\right) > \frac{C\epsilon^{159}}{d^2} \text{ for some constant } C > 0,$$

$$\forall v \perp u, \quad v^\top \mu_{\alpha_1} = v^\top \mu_{\alpha_2} = 0.$$

*Proof of Lemma 4.* For any $2 \leq k \leq d$, for any $\alpha$, by symmetry of $q$, the contrastive mean is

$$\mathbb{E}_{x\sim P} e^{\alpha\|x\|^2} x_k = \mathbb{E}_{x\sim q} e^{\alpha x^2} x \cdot \mathbb{E}_{x\sim \hat{q}} e^{\alpha x^2} \cdot \left(\mathbb{E}_{x\sim q} e^{\alpha x^2}\right)^{d-2} = 0$$

Next we will consider $\mathbb{E}_{x \sim P} e^{\alpha \|x\|^2} x_1$. For any $x \geq 0$, we have

$$\hat{q}(x) = \frac{\sigma_1 q(x\sigma_1 + \mu_1)}{1 - \int_a^b q(x)\,dx} \geq \frac{\sigma_1 q(-x\sigma_1 + \mu_1)}{1 - \int_a^b q(x)\,dx} = \hat{q}(-x)$$

Since $P$ is a product distribution, we have

$$\mathbb{E}_{x \sim P} e^{\alpha \|x\|^2} x_1 = \mathbb{E}_{x_1 \sim \hat{q}} e^{\alpha x_1^2} x_1 \cdot \prod_{i=2}^d \mathbb{E}_{x_i \sim q} e^{\alpha x_i^2} = F(\alpha) \left( \mathbb{E}_{x \sim q} e^{\alpha x^2} \right)^{d-1}$$

We will consider two cases depending on whether $a' \geq 0$. See Figure 4.1.

Firstly, if $a' \leq 0$. We use $\alpha_2$ in this case. By Lemma 20, $F'(0) > \epsilon^{18.5}/2$. By Taylor expansion, there exists $\alpha_2 < \eta_0 < 0$ such that

$$F(\alpha_2) = F(0) + \alpha_2 F'(0) + \frac{\alpha_2^2}{2} F''(\eta_0).$$

By Lemma 24, we know $F''(\eta_0) \leq C_5/\epsilon^5$. Since $F(0) = 0$, we know for $\alpha_2 = -c_2\epsilon^{42}/d$,

$$\begin{aligned}
F(\alpha_2) &= \alpha_2 F'(0) + \frac{\alpha_2^2}{2} F''(\eta_0) \\
&\leq \frac{c_2 \epsilon^{42}}{d} \left( -\frac{\epsilon^{18.5}}{2} + \frac{c_2 \epsilon^{42}}{2d} \frac{C_5}{\epsilon^5} \right) \\
&< -\frac{C_7 \epsilon^{61}}{d} \text{ for some constant } C_7 > 0
\end{aligned}$$

Then we consider the case when $a' > 0$. By Lemma 23, we have either $F'(0) < -\frac{C_2 \epsilon^{36.5}}{\log^4(1/\epsilon)}$ or $F''(0) > \frac{C_3 \epsilon^{34.5}}{\log^4(1/\epsilon)}$. Here we consider three cases with respect to $F'(0)$.

**Case 1**: $F'(0) < -\frac{C_2 \epsilon^{36.5}}{\ln^4(1/\epsilon)}$. We use $\alpha_2$ in this case. By Taylor expansion, there exists $\eta_1$ such that $\alpha_2 < \eta_1 < 0$ and

$$F(\alpha_2) = F(0) + \alpha_2 F'(0) + \frac{\alpha_2^2}{2} F''(\eta_1)$$

By Lemma 24, we know $|F''(\eta_1)| \leq C_5/\epsilon^5$. By choosing $\alpha_2 = -c_2\epsilon^{42}/d$,

$$\begin{aligned}
F(\alpha_1) &= \alpha_2 F'(0) + \frac{\alpha_2^2}{2} F''(\eta_1) \\
&\geq \frac{c_2 \epsilon^{42}}{d} \left( \frac{C_2 \epsilon^{36.5}}{\ln^4(1/\epsilon)} - \frac{c_2 \epsilon^{42}}{2d} \frac{C_5}{\epsilon^5} \right) \\
&> \frac{C_8 \epsilon^{79}}{d} \text{ for some constant } C_8 > 0
\end{aligned}$$

**Case 2**: $-\frac{C_2 \epsilon^{36.5}}{\ln^4(1/\epsilon)} \leq F'(0) \leq \frac{c_s \epsilon^{77}}{d}$ with some constant $c_s > 0$. We use $\alpha_2$ in this case. By Lemma 23, we know $F''(0) > C_3 \epsilon^{34.5}/\ln^4(1/\epsilon)$. Then there exists $\eta_2$ satisfying $\alpha_2 < \eta_2 < 0$ and

$$F(\alpha_2) = F(0) + \alpha_2 F'(0) + \frac{\alpha^2}{2} F''(0) + \frac{\alpha_2^3}{6} F'''(\eta_2).$$

By Lemma 24, we know $|F'''(\eta_2)| \leq C_6/\epsilon^7$. Thus by choosing $\alpha_2 = -c_2\epsilon^{42}/d$,

$$\begin{aligned}
F(\alpha_2) &= \alpha_2 F'(0) + \frac{\alpha_2^2}{2} F''(0) + \frac{\alpha_2^3}{6} F'''(\eta_2) \\
&> \frac{c_2 \epsilon^{42}}{d} \left( -\frac{c_s \epsilon^{77}}{d} + \frac{c_2 \epsilon^{42}}{2d} \frac{C_3 \epsilon^{34.5}}{\ln^4(1/\epsilon)} - \frac{c_2^2 \epsilon^{84}}{6d^2} \frac{C_6}{\epsilon^7} \right) \\
&\geq \frac{C_9 \epsilon^{119}}{d^2} \text{ for some constant } C_9 > 0
\end{aligned}$$

**Case 3**: $F'(0) > \frac{c_s \epsilon^{77}}{d}$. We use $\alpha_1$ in this case. Then there exists $\eta_3$ satisfying $\alpha_1 < \eta_3 < 0$ and

$$F(\alpha_1) = F(0) + \alpha_1 F'(0) + \frac{\alpha_1^2}{2} F''(\eta_3).$$

By Lemma 24, we know $|F''(\eta_3)| \le C_5/\epsilon^5$. For $\alpha_1 = -c_1 \epsilon^{82}/d$, we have

$$\begin{aligned}
F(\alpha_1) &= \alpha_1 F'(0) + \frac{\alpha_1^2}{2} F''(\eta_3) \\
&\le \frac{c_1 \epsilon^{82}}{d} \left( -\frac{c_s \epsilon^{77}}{d} + \frac{c_1 \epsilon^{82}}{2d} \frac{C_5}{\epsilon^5} \right) \\
&\le -\frac{C_{10} \epsilon^{159}}{d^2} \text{ for some constant } C_{10} > 0
\end{aligned}$$

Then we know for all cases, there exists a constant $C' = \min(C_7, C_8, C_9, C_{10})$ such that

$$\max(|F(\alpha_1)|, |F(\alpha_2)|) \ge \frac{C' \epsilon^{159}}{d^2}$$

Finally we will lower bound $\left( \underset{x \sim q}{\mathbb{E}} e^{\alpha x^2} \right)^{d-1}$ as follows.

$$\left( \underset{x \sim q}{\mathbb{E}} e^{\alpha x^2} \right)^{d-1} \ge \left( \underset{x \sim q}{\mathbb{E}} (1 + \alpha x^2) \right)^{d-1} = (1+\alpha)^{d-1} \ge \left( 1 - \frac{1}{d} \right)^{d-1} \ge 1/e$$

Let $C = C'/e$, and we will get

$$\max(|u^\top \mu_{\alpha_1}|, |u^\top \mu_{\alpha_2}|) > \frac{C \epsilon^{159}}{d^2}.$$

$\square$

## C.4 Quantitative Bounds for Contrastive Covariance: Symmetric Case

Before addressing Lemma 5 which is applicable in the scenario where $|a + b| < \epsilon^5$, we first demonstrate that contrastive covariance works for the case where the removed band $[a, b]$ is symmetric around the origin. That is, $a = -b$. In such cases, we aim to establish that there's a noticeable difference between the top two eigenvalues ($\lambda_1$ and $\lambda_2$) of the contrastive covariance matrix $\tilde{\Sigma}$, stated in Lemma 25. We will then extend the lemma to the near-symmetric scenario in Section C.5.

**Lemma 25** (Quantitative Spectral Gap of Contrastive Covariance - Symmetric Case). *Suppose $a + b = 0$. Choose $\alpha_3 = -c_3 \epsilon^2$ for some constant $c_3 > 0$. Then, for an absolute constant $C$, the top two eigenvalues $\lambda_1 \ge \lambda_2$ of the corresponding re-weighted covariance of $P$ satisfy*

$$\lambda_1 - \lambda_2 \ge C \epsilon^3 \lambda_1.$$

We recall the definition of moment ratio as in Definition 5.

$$\mathrm{mr}_q(t) = \frac{\mathrm{var}_{q_t}(X^2)}{(\mathbb{E}_{q_a} X^2)^2} = \frac{M_0(t) M_4(t)}{M_2^2(t)} - 1, \text{ where } M_k(t) = \int_t^\infty x^k q(x)\, dx.$$

For simplicity, in the remaining section, we'll use $\mathrm{mr}(t)$ as a shorthand notation for this moment ratio. Just as in the proof of the qualitative bound in Section C.2, the difference between the first and second eigenvalues (the spectral gap) of the re-weighted covariance matrix, denoted as $\lambda_1 - \lambda_2$, is proportional to the difference in the moment ratio of the distribution $q$ at $0$ and $b$, denoted as $\mathrm{mr}(0) - \mathrm{mr}(b)$. To prove the quantitative result Lemma 25, our proof strategy involves several steps.

- Proving monotonicity of Moment Ratio $\mathrm{mr}(t)$. This property is stated in Lemma 3 and its proof is presented in Section C.2. The proof involves reducing the case of general logconcave distributions to that of exponential distributions.

- Establishing a positive spectral gap for small $b$. With Lemma 26, we focus on illustrating that for values of $b$ which are relatively small (less than a certain constant), there is a guaranteed positive gap $\mathrm{mr}(0) - \mathrm{mr}(b)$.

- Generalizing to any $b$ satisfying $\int_{-b}^{b} q(x)\,dx \geq \epsilon$ in Lemma 29. The lemma will directly imply the quantitative result lemma for the symmetric case (Lemma 25).

Having demonstrated the monotonicity of the moment ratio in Section C.2, we now begin by illustrating the positive gap, denoted as $\mathrm{mr}(0) - \mathrm{mr}(b)$, for small $b$ in Lemma 26. The proof relies on the properties of moments derived by the unimodality of the distribution (Lemma 27, Lemma 28).

**Lemma 26** (Gap for small $t$). *For $t > 0$ such that $\int_0^t q(x)\,dx \leq 0.4$, we have the following gap*

$$mr(0) - mr(t) \geq \frac{\int_0^t q(x)\,dx}{100}.$$

*Proof.* Denote $\int_0^t q(x)\,dx = \nu$, $\int_0^t x^2 q(x)\,dx = \omega$. Then we have

$$
\begin{aligned}
\mathrm{mr}(0) - \mathrm{mr}(t) &= \frac{M_0(0)M_4(0)}{M_2^2(0)} - \frac{M_0(t)M_4(t)}{M_2^2(t)} \\
&= 2\int_0^t x^4 q(x)\,dx + \frac{M_4(t)}{2(\omega + M_2(t))^2} - \frac{(\frac{1}{2} - \nu)M_4(t)}{M_2^2(t)} \\
&= 2\int_0^t x^4 q(x)\,dx + \frac{M_4(t)}{2M_2^2(t)}\left(\frac{1}{\left(1 + \frac{\omega}{M_2(t)}\right)^2} - 1 + 2\nu\right)
\end{aligned}
$$

Fix $q(t)$ and $\nu$, we apply Lemma 27 and have

$$M_2(t) \geq \left(\frac{1}{2} - \nu\right) t^2 \left(1 + \frac{\frac{1}{2} - \nu}{q(t)t}\right)$$

Since $q(x)$ is monotonically decreasing, for any $x \leq t, q(x) \geq q(t)$. So we have the constraint that

$$\nu = \int_0^t q(x)\,dx \geq tq(t)$$

Plug into the previous inequality and we get

$$M_2(t) \geq \left(\frac{1}{2} - \nu\right) t^2 \left(1 + \frac{\frac{1}{2} - \nu}{q(t)t}\right) \geq \left(\frac{1}{2} - \nu\right) t^2 \left(1 + \frac{\frac{1}{2} - \nu}{\nu}\right) = \left(\frac{1}{2} - \nu\right) \frac{t^2}{2\nu}$$

In addition, by fixing $v$ and $t$, we apply Lemma 28 and get

$$\omega \leq \frac{\nu t^2}{3}$$

So we know

$$\frac{\omega}{M_2(t)} \leq \frac{\frac{\nu t^2}{3}}{\left(\frac{1}{2} - \nu\right) \frac{t^2}{2\nu}} = \frac{2\nu^2}{3\left(\frac{1}{2} - \nu\right)}$$

By calculation, we will get

$$
\begin{aligned}
\frac{1}{\left(1 + \frac{\omega}{M_2(t)}\right)^2} - 1 + 2\nu &\geq \frac{9\left(\frac{1}{2} - \nu\right)^2 - 2\left(\frac{1}{2} - \nu\right)\left(\frac{3}{2} - 3\nu + 2\nu^2\right)^2}{\left(\frac{3}{2} - 3\nu + 2\nu^2\right)^2} \\
&\geq \frac{4}{9}\left(\frac{1}{2} - \nu\right)\left(9\left(\frac{1}{2} - \nu\right) - 2\left(\frac{3}{2} - 3\nu + 2\nu^2\right)^2\right) \\
&= \frac{4}{9}\nu\left(\frac{1}{2} - \nu\right)\left(-8\nu^3 + 24\nu^2 - 30\nu + 9\right)
\end{aligned}
$$

Let $T(\nu) = -8\nu^3 + 24\nu^2 - 30\nu + 9$, then we know its derivative is

$$T'(\nu) = -24\nu^2 + 48\nu - 30 = -24(\nu - 1)^2 - 6 < 0$$

So $T(\nu)$ is monotonically decreasing. Since $\nu < 0.4$, we know

$$T(\nu) \geq T(0.4) > 0.3, \frac{1}{2} - \nu > 0.1$$

Plugging in and we will get

$$\frac{1}{\left(1 + \frac{\omega}{M_2(t)}\right)^2} - 1 + 2\nu \geq \frac{4}{9}\nu \cdot 0.1 \cdot 0.3 > 0.01\nu$$

Finally by Cauchy-Schwarz Inequality, we have $M_0(t)M_4(t) \geq M_2^2(t)$.

$$\mathrm{mr}(0) - \mathrm{mr}(t) \geq \frac{M_4(t)}{2M_2^2(t)}0.01\nu \geq \frac{\nu}{200M_0(t)} \geq \frac{\nu}{100}$$

$\square$

**Lemma 27.** *Let* $t, r \geq 0, 0 < s \leq 1$. *Define* $\mathcal{P} = \{p(x) : [t, \infty) \to [0, 1)$, *logconcave* , $p'(x) \leq 0, \int_t^\infty p(x)\, dx = s, p(t) = r\}$. *Then we have*

$$\min_{p \in \mathcal{P}} \int_t^\infty x^2 p(x)\, dx \geq st^2\left(1 + \frac{s}{rt}\right).$$

*Proof.* For any $p \in \mathcal{P}$, we denote $M_2(p) = \int_t^\infty x^2 p(x)\, dx$. Define

$$u(x) = r \cdot \mathbb{1}_{x \in [t, t+s/r]}$$

Clearly $u \in \mathcal{P}$. We will show that $u(x) = \mathrm{argmin}_{p \in P} M_2(p)$. For any $p \in \mathcal{P}$, we have $\int_t^\infty u(x)\, dx = \int_t^\infty p(x)\, dx$, $u(t) = p(t)$ and $p'(x) \leq 0$. So we know the graph of $u$ and $p$ intersects at points $t$ and $t + s/r$, where $u(x) \geq p(x)$ in the interval $[t, t + s/r]$ and $u(x) < p(x)$ outside the interval. So we know

$$\int_t^{t+s/r} u(x) - p(x)\, dx = \int_{t+s/r}^\infty p(x) - u(x)\, dx$$

Since for any $x \in [t, t + s/r]$, any $y \in [t + s/r, \infty)$, we have $x \leq y$. So we have

$$M_2(u) - M_2(p) = \int_t^{t+s/r} (u(x) - p(x))x^2\, dx - \int_{t+s/r}^\infty (p(x) - u(x))x^2\, dx \leq 0$$

This shows that

$$\min_{p \in \mathcal{P}} M_2(p) = M_2(u).$$

By calculating $M_2(u)$, we have

$$\min_{p \in \mathcal{P}} M_2(p) = M_2(u) = \int_t^{t+s/r} rx^2\, dx = r(t^2\frac{s}{r} + t\frac{s^2}{r^2} + \frac{s^3}{3r^3}) \geq st^2\left(1 + \frac{s}{rt}\right)$$

$\square$

**Lemma 28.** *Let* $t \geq 0, 0 < s \leq 1$. *Define* $\mathcal{P} = \{p(x) : [0, t] \to [0, 1)$, *logconcave,* $p'(x) \leq 0, \int_0^t p(x)\, dx = s\}$. *Then we have*

$$\max_{p \in \mathcal{P}} \int_0^t x^2 p(x)\, dx = \frac{st^2}{3}.$$

*Proof.* For any $p \in \mathcal{P}$, we denote $M_2(p) = \int_0^t x^2 p(x)\, dx$. Define $u(x) = \frac{s}{t} \cdot \mathbb{1}_{x \in [0, t]}$. Clearly, $u \in \mathcal{P}$. Then for any $p \in \mathcal{P}$, because it is monotonically decreasing and $\int_0^t p(x)\, dx = \int_0^t u(x)\, dx$, the graphs of $p$ and $u$ intersect at point $l \in [0, t]$. Also $p(x) \geq u(x)$ for $x \in [0, l]$ and $p(x) \leq u(x)$ for $x \in [l, t]$. Since for any $x \in [0, l]$ and any $y \in [l, t]$, we have $x \leq y$. So we know

$$M_2(p) - M_2(u) = \int_0^l x^2(p(x) - u(x))\, dx - \int_l^t x^2(u(x) - p(x)) \leq 0$$

By calculating $M_2(u)$, we have

$$\max_{p \in \mathcal{P}} M_2(p) = M_2(u) = \int_0^t \frac{s}{t} x^2 \, dx = \frac{s}{t} \frac{1}{3} t^3 = \frac{st^2}{3}$$

$\square$

Using Lemma 3 and Lemma 26, we will show the positive gap $\mathrm{mr}(0) - \mathrm{mr}(b)$ for any $b$ satisfying $\int_{-b}^b q(x) \, dx \geq \epsilon$ as in Lemma 29.

**Lemma 29** (Gap for Log-concave Distribution). *Let $0 < \epsilon < 0.1$, let $b > 0$ satisfying $\int_0^b q(x) \, dx \geq \epsilon/2$. Then we have for $\mathrm{mr}(0) - \mathrm{mr}(b) \geq \epsilon/200$.*

*Proof.* Let $b_0$ such that $\int_0^{b_0} q(x) \, dx = \epsilon/2$. By Lemma 26, we know

$$\mathrm{mr}(0) - \mathrm{mr}(b_0) \geq \frac{\int_0^{b_0} q(x) \, dx}{100} \geq \frac{\epsilon}{200}$$

By Lemma 3, we know $\mathrm{mr}'(t) \leq 0, \forall t \geq 0$. So for any $b > 0$ such that $\int_0^b q(x) \, dx \geq \epsilon/2$, $\mathrm{mr}(b) \leq \mathrm{mr}(a_0)$. Therefore,

$$\mathrm{mr}(0) - \mathrm{mr}(b) \geq \mathrm{mr}(0) - \mathrm{mr}(b_0) \geq \frac{\epsilon}{200}$$

$\square$

Before moving on to the proof of the quantitative lemma, we will first present a helper lemma that can be directly applied.

**Lemma 30.** *Define*

$$S(\alpha) := \mathop{\mathbb{E}}_{x \sim \hat{q}} e^{\alpha x^2} x^2 \mathop{\mathbb{E}}_{x \sim q} e^{\alpha x^2} - \mathop{\mathbb{E}}_{x \sim q} e^{\alpha x^2} x^2 \mathop{\mathbb{E}}_{x \sim \hat{q}} e^{\alpha x^2} \tag{C.3}$$

*For a given $\alpha = -c\epsilon^s$ with $s \geq 2$ and a certain positive constant $c$, it can be established that*

$$S(\alpha) > C\epsilon^{s+1}, \quad \text{where } C \text{ is a positive constant.}$$

*Proof.* We show the lower bound of $S(\alpha)$ using Taylor expansion.

Firstly, since $q$ and $\hat{q}$ are both isotropic,

$$S(0) = \mathop{\mathbb{E}}_{x \sim \hat{q}} x^2 - \mathop{\mathbb{E}}_{x \sim q} x^2 = 0$$

Secondly, we will lower bound $|S'(0)|$ using the monotonicity of moment ratio. The variance $\sigma_1^2$ of $q$ restricted to $\mathbb{R} \backslash [-b, b]$ is

$$\sigma_1^2 = \frac{\int_b^\infty x^2 p(x) \, dx}{\int_b^\infty p(x) \, dx} = \frac{M_2(b)}{M_0(b)}$$

Since $\hat{q}$ is isotropic, the density on the support $\mathbb{R} \backslash [-b/\sigma_1, b/\sigma_1]$ is

$$\mathbb{P}_{\hat{q}}(x) = \frac{\sigma_1 q(x \sigma_1)}{2 \int_b^\infty q(x) \, dx}$$

By calculation, we have

$$\begin{aligned}
S(\alpha) =& \frac{2 \int_{b/\sigma_1}^\infty e^{\alpha x^2} x^2 \sigma_1 p(x \sigma_1) \, dx}{2 \int_b^\infty q(x) \, dx} \cdot 2 \int_0^\infty e^{\alpha x^2} q(x) \, dx - \frac{2 \int_{b/\sigma_1}^\infty e^{\alpha x^2} \sigma_1 p(x \sigma_1) \, dx}{2 \int_b^\infty q(x) \, dx} \cdot 2 \int_0^\infty e^{\alpha x^2} x^2 q(x) \, dx \\
=& \frac{2 \int_b^\infty e^{\alpha y^2 / \sigma_1^2} y^2 / \sigma_1^2 q(y) \, dy}{\int_b^\infty q(x) \, dx} \cdot \int_0^\infty e^{\alpha x^2} q(x) \, dx - \frac{2 \int_b^\infty e^{\alpha y^2 / \sigma_1^2} q(y) \, dy}{\int_b^\infty q(x) \, dx} \cdot \int_0^\infty e^{\alpha x^2} x^2 q(x) \, dx
\end{aligned}$$

$$= \frac{\frac{2}{\sigma_1^2} \int_b^\infty e^{\alpha y^2/\sigma_1^2} y^2 q(y)\, dy \cdot \int_0^\infty e^{\alpha x^2} q(x)\, dx - 2\int_b^\infty e^{\alpha y^2/\sigma_1^2} q(y)\, dy \cdot \int_0^\infty e^{\alpha x^2} x^2 q(x)\, dx}{\int_b^\infty q(x)\, dx}$$

Then we can compute $S'(0)$ as

$$
\begin{aligned}
S'(0) =& \frac{2}{\sigma_1^2 \int_b^\infty q(x)\, dx} \left( \frac{1}{\sigma_1^2} \int_b^\infty y^4 q(y)\, dy \cdot \int_0^\infty q(x)\, dx + \int_b^\infty y^2 q(y)\, dy \cdot \int_0^\infty x^2 q(x)\, dx \right) \\
& - \frac{2}{\int_b^\infty q(x)\, dx} \left( \frac{1}{\sigma_1^2} \int_b^\infty y^2 q(y)\, dy \cdot \int_0^\infty x^2 q(x)\, dx + \int_b^\infty q(y)\, dy \cdot \int_0^\infty x^4 q(x)\, dx \right) \\
=& \frac{2 M_0(b) M_4(b) M_0(0)}{M_2^2(b)} - 2 M_4(0) \\
=& \frac{M_4(b) M_0(b)}{M_2^2(b)} - \frac{M_4(0) M_0(0)}{M_2^2(0)} \\
=& \mathrm{mr}(b) - \mathrm{mr}(0)
\end{aligned}
$$

The last step is because the $q$ is isotropic. By Lemma 29, $\mathrm{mr}(0) - \mathrm{mr}(b) \geq \epsilon/200$. This indicates that

$$S'(0) \leq -\epsilon/200$$

Next, we can upper bound $S''(\alpha)$ for any $\alpha \leq 0$ as

$$
\begin{aligned}
S''(\alpha) =& \frac{2}{\int_b^\infty p(x)\, dx} \left( \frac{1}{\sigma_1^6} \int_b^\infty e^{\alpha y^2/\sigma_1^2} y^6 q(y)\, dy \cdot \int_0^\infty e^{\alpha x^2} q(x)\, dx \right. \\
& + \frac{1}{\sigma_1^4} \int_b^\infty e^{\alpha y^2/\sigma_1^2} y^4 q(y)\, dy \cdot \int_0^\infty e^{\alpha x^2} x^2 q(x)\, dx \\
& - \frac{1}{\sigma_1^2} \int_b^\infty e^{\alpha y^2/\sigma_1^2} y^2 q(y)\, dy \cdot \int_0^\infty e^{\alpha x^2} x^4 q(x)\, dx \\
& \left. - \int_b^\infty e^{\alpha y^2/\sigma_1^2} q(y)\, dy \cdot \int_0^\infty e^{\alpha x^2} x^6 q(x)\, dx \right) \\
\geq& -\frac{2}{M_0(b)} \left( \frac{M_2(b) M_4(0)}{M_2(0)} + M_0(b) M_6(0) \right) \\
\geq& -\frac{2}{M_0(b)} \left( M_4(0) + M_0(0) M_6(0) \right)
\end{aligned}
$$

By Cauchy-Schwarz Inequality, $M_1(0) \leq \sqrt{M_2(0) M_0(0)} = 1/2$. By Lemma 12,

$$M_k(0) \leq (2k)^k (2 M_1(0))^k / 2 \leq (2k)^k / 2$$

Since $M_0(b) \geq \epsilon$, for some positive constant $c_{sec}$,

$$S''(\alpha) \geq -\frac{c_{sec}}{\epsilon}$$

By Taylor expansion, we know for $\alpha_3 = -c\epsilon^s, s \geq 2, c = 1/(101 c_{sec})$, there exists $\alpha' \in [\alpha, 0]$ such that for some constant $C > 0$,

$$S(\alpha) = S(0) + \alpha S'(0) + \frac{\alpha^2}{2} S''(\alpha') \geq 0 + c\epsilon^s \frac{\epsilon}{200} - \frac{c^2 \epsilon^{2s}}{2} \frac{c_{sec}}{\epsilon} > C\epsilon^{s+1}$$

$\square$

Now we are ready to prove the contrastive covariance lemma (Lemma 25).

*Proof of Lemma 25.* Define

$$S(\alpha) := \mathop{\mathbb{E}}_{x \sim \hat{q}} e^{\alpha x^2} x^2 \mathop{\mathbb{E}}_{x \sim q} e^{\alpha x^2} - \mathop{\mathbb{E}}_{x \sim q} e^{\alpha x^2} x^2 \mathop{\mathbb{E}}_{x \sim \hat{q}} e^{\alpha x^2}$$

Then for $2 \leq j \leq d$,

$$\mathop{\mathbb{E}}_{x \sim P} e^{\alpha \|x\|^2} x_1^2 - \mathop{\mathbb{E}}_{x \sim P} e^{\alpha \|x\|^2} x_j^2$$

$$= \left( \mathop{\mathbb{E}}_{x_1 \sim \hat{q}} e^{\alpha x_1^2} x_1^2 \mathop{\mathbb{E}}_{x_2 \sim q} e^{\alpha x_2^2} - \mathop{\mathbb{E}}_{x_2 \sim \hat{q}} e^{\alpha x_2^2} x_1^2 \mathop{\mathbb{E}}_{x_1 \sim q} e^{\alpha x_1^2} \right) \left( \mathop{\mathbb{E}}_{x \sim q} e^{\alpha x^2} \right)^{d-2}$$

$$= S(\alpha) \frac{\mathop{\mathbb{E}}_{x \sim P} e^{\alpha \|x\|^2} x_1^2}{\mathop{\mathbb{E}}_{x_1 \sim \hat{q}} e^{\alpha x_1^2} x_1^2 \mathop{\mathbb{E}}_{x_2 \sim q} e^{\alpha x_2^2}}$$

Since $\alpha_3 < 0$, we have

$$\mathop{\mathbb{E}}_{x_1 \sim \hat{q}} e^{\alpha_3 x_1^2} x_1^2 \leq \mathop{\mathbb{E}}_{x_1 \sim \hat{q}} x_1^2 = 1, \quad \mathop{\mathbb{E}}_{x_2 \sim q} e^{\alpha_3 x_2^2} \leq 1$$

By Lemma 30, $\alpha_3 = c_3 \epsilon^2$ implies that $S(\alpha_3) > C\epsilon^3$. So we have

$$\mathop{\mathbb{E}}_{x \sim P} e^{\alpha \|x\|^2} x_1^2 - \mathop{\mathbb{E}}_{x \sim P} e^{\alpha \|x\|^2} x_j^2 \geq C\epsilon^3 \mathop{\mathbb{E}}_{x \sim P} e^{\alpha \|x\|^2} x_1^2$$

Finally we will show that the first eigenvector corresponds to $e_1$. For any $v \in \mathbb{R}$, define $\phi(v)$ as

$$\phi(v) := \mathop{\mathbb{E}}_{x \sim P} \frac{e^{\alpha_3 \|x\|^2} v^\top x x^\top v}{v^\top v}$$

Then we know for $2 \leq j \leq d$,

$$\phi(e_1) - \phi(e_j) > C\epsilon^3 \phi(e_1)$$

For any vector $v = \sum_{i=1}^{d} \gamma_i e_i$, we have

$$\phi(v) = \frac{1}{\sum_{i=1}^{d} \gamma_i^2} \mathbb{E} e^{\alpha_3 \|x\|^2} \left( \sum_{i=1}^{d} \gamma_i x_i \right)^2$$

$$= \frac{1}{\sum_{i=1}^{d} \gamma_i^2} \left( \sum_{i=1}^{d} \gamma_i^2 \phi(e_i) + 2 \mathbb{E} e^{\alpha \|x\|^2} \sum_{i \neq j} \gamma_i \gamma_j x_i x_j \right)$$

$$= \frac{1}{\sum_{i=1}^{d} \gamma_i^2} \left( \sum_{i=1}^{d} \gamma_i^2 \phi(e_i) + 2 \sum_{i \neq j} \gamma_i \gamma_j \mathbb{E} e^{\alpha_3 \sum_{k \neq i,j} x_k^2} \mathbb{E} e^{\alpha_3 \langle x, e_i \rangle^2} x_1 \mathbb{E} e^{\alpha \langle x, e_j \rangle^2} x_j \right)$$

$$= \frac{\sum_{i=1}^{d} \gamma_i^2 \phi(e_i)}{\sum_{i=1}^{d} \gamma_i^2}$$

$$\leq \phi(e_1)$$

This shows that the top eigenvalue of $\tilde{\Sigma}$ is $\lambda_1 = \max_v \phi(v) = \phi(e_1)$. In other word, the top eigenvector is $e_1$, which is essentially $u$. Similarly the second eigenvalue of $\tilde{\Sigma}$ is $\lambda_2 = \max_{v:v \perp e_1} \phi(v) = \phi(e_j), 2 \leq j \leq d$. So we get

$$\lambda_1 - \lambda_2 > C\epsilon^3 \lambda_1$$

$\square$

## C.5 Quantitative Bounds for Contrastive Covariance: Near-Symmetric Case

We have shown the result for symmetric case $a + b = 0$ in Section C.4. Here we will show that we can extend the contrastive covariance lemma (Lemma 25) to the near-symmetric case, where $|a + b| < \epsilon^5$. In this section, we will present the proof of Lemma 5, which addresses the nearly symmetric case quantitatively. The proof idea is to approximate the re-weighted covariance of the distribution with margin $[a, b]$, by comparing it to the same distribution truncated with the symmetric interval $[-b, b]$. This enables us to generalize the result from the symmetric scenario to the near-symmetric scenario.

Recall that $\tilde{q}$ is the distribution obtained by restricting $q$ to the set $\mathbb{R}\backslash[a, b]$. We denote $\tilde{r}$ as the distribution that is obtained by restricting $q$ to the set $\mathbb{R}\backslash[-b, b]$, and $\hat{r}$ as the isotropized distribution of $\tilde{r}$. Let $\sigma_2^2$ be the variance of $\tilde{r}$.

To approximate the characteristics of $\hat{q}$ using those of $\hat{r}$, we undertake the subsequent steps.

- Assessing the mean. We illustrate that the mean of $\tilde{q}$ is adequately small in Lemma 31.
- Variance approximation. We approximate the variance of $\hat{q}$ by using the variance of $\hat{r}$, as elaborated in Lemma 32.
- Re-weighted second moment. We use the re-weighted second moment of $\hat{r}$ to approximate the corresponding moment in $\hat{q}$. The details are provided in Lemma 34.
- Re-weighted zeroth moment. We use the re-weighted zero moment of $\hat{r}$ to approximate the corresponding moment in $\hat{q}$, which is shown in Lemma 35.

**Lemma 31.** *For an integer $s \geq 1$, if $|a + b| < \epsilon^{s+1}$, then*

*(1) the mean of $\tilde{q}$, $|\mu_1| < \epsilon^s \ln(1/\epsilon)$;*

*(2) $a < 0$.*

*Proof.* We first consider the case when $a < 0$. By Lemma 8, we have

$$|\mu_1| = \frac{\int_{|a|}^b xq(x)\,dx}{1 - \int_a^b q(x)\,dx} \leq \frac{b\int_{|a|}^b q(x)\,dx}{\int_{-\infty}^a q(x)\,dx + \int_b^\infty q(x)\,dx} \leq \frac{b(b+a)}{2\epsilon}$$

By the tail bound of logconcave distributions (Lemma 11),

$$|b| < 1 + \ln\frac{1}{\epsilon}$$

Then we have

$$|\mu_1| < \frac{(1 + \ln\frac{1}{\epsilon})\epsilon^{s+1}}{2\epsilon} < \epsilon^s \ln(\frac{1}{\epsilon})$$

Next for $a \geq 0$, we can bound $b$ by $\epsilon^{s+1}/2$ because $|b| \geq |a|$. This implies that $b - a \leq \epsilon^{s+1}/2$, which leads to a contradiction that $\int_a^b q(x)\,dx \geq \epsilon$. Therefore, $a$ can only be negative in this scenario.

$\square$

**Lemma 32.** *For $|\mu_1| \leq r$ with $r < 1/6$, we can bound the variance as follows.*

$$\frac{\sigma_2^2}{1 + 2er} \leq \sigma_1^2 \leq \sigma_2^2$$

*Proof.* By Lemma 31, we know $a < 0$. We can calculate the variance as

$$\sigma_2^2 = \frac{\int_b^\infty x^2 q(x)\,dx}{\int_b^\infty q(x)\,dx}, \quad \sigma_1^2 = \frac{\int_b^\infty x^2 q(x)\,dx + \frac{1}{2}\int_{-a}^b x^2 q(x)\,dx}{\int_b^\infty q(x)\,dx + \frac{1}{2}\int_{-a}^b q(x)\,dx}$$

On one hand,

$$\frac{\int_{-a}^b x^2 q(x)\,dx}{\int_b^\infty x^2 q(x)\,dx} \leq \frac{b^2 \int_{-a}^b q(x)\,dx}{b^2 \int_b^\infty q(x)\,dx} = \frac{\int_{-a}^b q(x)\,dx}{\int_b^\infty q(x)\,dx}$$

So we have

$$\frac{\sigma_1^2}{\sigma_2^2} = \frac{1 + \frac{\int_{-a}^b x^2 q(x)\,dx}{2\int_b^\infty x^2 q(x)\,dx}}{1 + \frac{\int_{-a}^b q(x)\,dx}{2\int_b^\infty q(x)\,dx}} \leq 1$$

On the other hand, since $|\mu_1| \leq r$, by Lemma 8,

$$r \geq |\mu_1| = \frac{\int_{-a}^b xq(x)\,dx}{\int_{-a}^b q(x)\,dx + 2\int_b^\infty q(x)\,dx} > \frac{\frac{1}{e}\int_{-a}^b q(x)\,dx}{\int_{-a}^b q(x)\,dx + 2\int_b^\infty q(x)\,dx}$$

So we have

$$\frac{\sigma_2^2}{\sigma_1^2} = \frac{1 + \frac{\int_{-a}^b q(x)\,dx}{2\int_b^\infty q(x)\,dx}}{1 + \frac{\int_{-a}^b x^2 q(x)\,dx}{2\int_b^\infty x^2 q(x)\,dx}} \leq 1 + \frac{\int_{-a}^b q(x)\,dx}{2\int_b^\infty q(x)\,dx} \leq 1 + \frac{er}{1 - er} < 1 + 2er$$

$\square$

**Lemma 33.** *The variance $\sigma_2^2$ is monotonically increasing with respect to b. Furthermore, $\sigma_2^2 \geq 1$.*

*Proof.* By taking the derivative,

$$\frac{\partial \sigma_2^2}{\partial b} = -\frac{q(b)}{(\int_b^\infty q(x)\,dx)^2}\left(b^2 \int_b^\infty q(x)\,dx - \int_b^\infty x^2 q(x)\,dx\right) > 0$$

So for $b > 0$,

$$\sigma_2^2 \geq \frac{\int_0^\infty x^2 q(x)\,dx}{\int_0^\infty q(x)\,dx} = 1$$

$\square$

**Lemma 34** (Approximation for Re-weighted Second Moment). *For $|a + b| \leq \epsilon^5$, by choosing $\alpha = -c\epsilon^2$, then for some constant $C' > 0$, we have the following inequalities.*

$$\int_b^\infty e^{\alpha y^2/\sigma_2^2} y^2 q(y)\,dy - C'\epsilon^5 \leq \int_b^\infty e^{\alpha(y-\mu_1)^2/\sigma_1^2} y^2 q(y)\,dy \leq \int_b^\infty e^{\alpha y^2/\sigma_2^2} y^2 q(y)\,dy \quad \text{(C.4)}$$

$$\int_b^\infty e^{\alpha y^2/\sigma_2^2} y^2 q(y)\,dy - C'\epsilon^5 \leq \int_{-a}^\infty e^{\alpha(y+\mu_1)^2/\sigma_1^2} y^2 q(y)\,dy \leq \int_b^\infty e^{\alpha y^2/\sigma_2^2} y^2 q(y)\,dy + C'\epsilon^5$$
$$\text{(C.5)}$$

*Proof.* By Lemma 31, we can bound $|\mu_1|$ as

$$|\mu_1| < \epsilon^4 \ln\frac{1}{\epsilon} < \epsilon^3$$

We begin by showing that $\int_b^\infty e^{\alpha y^2/\sigma_2^2} y^2 q(y)\,dy$ is close to $\int_b^\infty e^{\alpha y^2/\sigma_1^2} y^2 q(y)\,dy$. By Lemma 32,

$$\frac{\sigma_2^2}{1 + 2e\epsilon^3} \leq \sigma_1^2 \leq \sigma_2^2$$

Since $\alpha < 0$, $\alpha y^2/\sigma_1^2 < \alpha y^2/\sigma_2^2$ for $y > 0$. This implies

$$\int_b^\infty e^{\alpha y^2/\sigma_1^2} y^2 q(y)\,dy \leq \int_b^\infty e^{\alpha y^2/\sigma_2^2} y^2 q(y)\,dy.$$

On the other hand,

$$\int_b^\infty e^{\alpha y^2/\sigma_2^2} y^2 q(y)\,dy - \int_b^\infty e^{\alpha y^2/\sigma_1^2} y^2 q(y)\,dy$$

$$\leq \int_b^\infty e^{\alpha y^2/\sigma_2^2} y^2 q(y)\,dy - \int_b^\infty e^{\alpha y^2(1+2e\epsilon^3)/\sigma_2^2} y^2 q(y)\,dy$$

$$= \int_b^\infty \left(1 - e^{2e\alpha y^2 \epsilon^3/\sigma_2^2}\right) e^{\alpha y^2/\sigma_2^2} y^2 q(y)\,dy$$

$$\leq \int_b^\infty \frac{2e|\alpha|y^2 \epsilon^3}{\sigma_2^2} e^{\alpha y^2/\sigma_2^2} y^2 q(y)\,dy$$

$$= \frac{2e|\alpha|\epsilon^3}{\sigma_2^2} \int_b^\infty e^{\alpha y^2/\sigma_2^2} y^4 q(y)\,dy$$

$$\leq \frac{2e|\alpha|\epsilon^3}{\sigma_2^2} \int_0^\infty y^4 q(y)\,dy$$

$$\leq \frac{2e|\alpha|\epsilon^3}{\sigma_2^2} * 8^4/2$$

$$= \frac{8^4 e c \epsilon^5}{\sigma_2^2}$$

The last inequality is implied by Lemma 12. Furthermore, by Lemma 33, $\sigma_2^2 \geq 1$. So there exists a constant $c_1 > 0$ such that

$$\int_b^\infty e^{\alpha y^2/\sigma_2^2} y^2 q(y)\, dy - c_1 \epsilon^5 \leq \int_b^\infty e^{\alpha y^2/\sigma_1^2} y^2 q(y)\, dy \leq \int_b^\infty e^{\alpha y^2/\sigma_2^2} y^2 q(y)\, dy. \qquad \text{(C.6)}$$

This also applies for the integral from $-a$ to $\infty$. To be specific,

$$\int_{-a}^\infty e^{\alpha y^2/\sigma_2^2} y^2 q(y)\, dy - c_1 \epsilon^5 \leq \int_{-a}^\infty e^{\alpha y^2/\sigma_1^2} y^2 q(y)\, dy \leq \int_{-a}^\infty e^{\alpha y^2/\sigma_2^2} y^2 q(y)\, dy. \qquad \text{(C.7)}$$

Next we will show that $\int_b^\infty e^{\alpha y^2/\sigma_1^2} y^2 q(y)\, dy$ and $\int_b^\infty e^{\alpha(y-\mu_1)^2/\sigma_1^2} y^2 q(y)\, dy$ are close to each other. Since $\mu_1 < 0, \alpha < 0$, we derive that $\alpha y^2 > \alpha(y-\mu_1)^2$. This implies

$$\int_b^\infty e^{\alpha y^2/\sigma_1^2} y^2 q(y)\, dy > \int_b^\infty e^{\alpha(y-\mu_1)^2/\sigma_1^2} y^2 q(y)\, dy$$

On the other hand,

$$\int_b^\infty e^{\alpha y^2/\sigma_1^2} y^2 q(y)\, dy - \int_b^\infty e^{\alpha(y-\mu_1)^2/\sigma_1^2} y^2 q(y)\, dy$$

$$\leq \int_b^\infty \left( 1 - e^{-(|\alpha|\mu_1^2 + |\alpha\mu_1|y)/\sigma_1^2} \right) e^{\alpha y^2/\sigma_1^2} y^2 q(y)\, dy$$

$$\leq \frac{|\alpha\mu_1|}{\sigma_1^2} \int_b^\infty e^{\alpha y^2/\sigma_1^2} (y + |\mu_1|) y^2 q(y)\, dy$$

$$\leq \frac{|\alpha\mu_1|}{\sigma_1^2} \int_0^\infty (y + |\mu_1|) y^2 q(y)\, dy$$

$$\leq c_2 \epsilon^5 \quad \text{for some constant } c_2 > 0$$

The last inequality is implied by Lemma 12 and Lemma 33 . Combining two inequalities, we get

$$\int_b^\infty e^{\alpha y^2/\sigma_1^2} y^2 q(y)\, dy - c_2 \epsilon^5 \leq \int_b^\infty e^{\alpha(y-\mu_1)^2/\sigma_1^2} y^2 q(y)\, dy \leq \int_b^\infty e^{\alpha y^2/\sigma_1^2} y^2 q(y)\, dy \qquad \text{(C.8)}$$

Similarly, we will show the approximation inequality for $\int_{-a}^\infty e^{\alpha(y+\mu_1)^2/\sigma_1^2} y^2 q(y)\, dy$. We will decompose the integral by the summation of the integral on $[-a, 6\ln(1/\epsilon)]$ and $(6\ln(1/\epsilon), \infty)$ respectively. For the first part of the integral,

$$\left| \int_{-a}^{6\ln(1/\epsilon)} e^{\alpha(y+\mu_1)^2/\sigma_1^2} y^2 q(y)\, dy - \int_{-a}^{6\ln(1/\epsilon)} e^{\alpha y^2/\sigma_1^2} y^2 q(y)\, dy \right|$$

$$= \left| \int_{-a}^{6\ln(1/\epsilon)} e^{\alpha y^2/\sigma_1^2} \left( e^{\frac{\alpha\mu_1}{\sigma_1^2}(2y+\mu_1)} - 1 \right) y^2 q(y)\, dy \right|$$

$$\leq \int_{-a}^{-\mu_1/2} e^{\alpha y^2/\sigma_1^2} \left( 1 - e^{\frac{\alpha\mu_1}{\sigma_1^2}(2y+\mu_1)} \right) y^2 q(y)\, dy + \int_{-\mu_1/2}^{6\ln(1/\epsilon)} e^{\alpha y^2/\sigma_1^2} \left( e^{\frac{\alpha\mu_1}{\sigma_1^2}(2y+\mu_1)} - 1 \right) y^2 q(y)\, dy$$

The first term can be bounded using $e^{-t} \geq 1 - t$.

$$\int_{-a}^{-\mu_1/2} e^{\alpha y^2/\sigma_1^2} \left( 1 - e^{\frac{\alpha\mu_1}{\sigma_1^2}(2y+\mu_1)} \right) y^2 q(y)\, dy \leq \int_{-a}^{-\mu_1/2} \frac{\alpha\mu_1}{\sigma_1^2}(-2y-\mu_1) y^2 q(y)\, dy$$

$$\leq \int_{-a}^{-\mu_1/2} |\alpha||\mu_1|^2 y^2 q(y)\, dy \leq \epsilon^8$$

The second term can be bounded using the upper limit of the integral. For $y \leq 6\ln(1/\epsilon)$,

$$e^{\frac{\alpha\mu_1}{\sigma_1^2}(2y+\mu_1)} - 1 \leq e^{24\epsilon^6\ln^2(1/\epsilon)} - 1 \leq 40\epsilon^5$$

Substituting into the second term, we get

$$\int_{-\mu_1/2}^{6\ln(1/\epsilon)} e^{\alpha y^2/\sigma_1^2}\left(e^{\frac{\alpha\mu_1}{\sigma_1^2}(2y+\mu_1)} - 1\right)y^2 q(y)\,dy \leq 40\epsilon^5\int_0^\infty y^2 q(y)\,dy = 20\epsilon^5$$

Combining both terms,

$$\left|\int_{-a}^{6\ln(1/\epsilon)} e^{\alpha(y+\mu_1)^2/\sigma_1^2}y^2 q(y)\,dy - \int_{-a}^{6\ln(1/\epsilon)} e^{\alpha y^2/\sigma_1^2}y^2 q(y)\,dy\right| \leq 21\epsilon^5 \qquad \text{(C.9)}$$

For the remaining part of the integral, we can bound using logconcave distribution's upper bound as in Lemma 13. For any $t \geq 3$,

$$q(t) \leq q(0) \cdot 2^{-t/3} < e^{-t/5}$$

Then the following holds with some constant $c_3' > 0$.

$$\int_{6\ln(1/\epsilon)}^\infty e^{\alpha y^2/\sigma_1^2}y^2 q(y)\,dy \leq \int_{6\ln(1/\epsilon)}^\infty y^2 e^{-y/5}\,dy \leq (1+6\ln(\frac{1}{\epsilon}))e^{-6\ln(\frac{1}{\epsilon})} \leq c_3'\epsilon^5 \qquad \text{(C.10)}$$

Similarly, we get

$$\int_{6\ln(1/\epsilon)}^\infty e^{\alpha(y+\mu_2)^2/\sigma_1^2}y^2 q(y)\,dy \leq c_3'\epsilon^5 \qquad \text{(C.11)}$$

With Equations (C.9), (C.10), (C.11), we get

$$\left|\int_{-a}^\infty e^{\alpha(y+\mu_1)^2/\sigma_1^2}y^2 q(y)\,dy - \int_{-a}^\infty e^{\alpha y^2/\sigma_1^2}y^2 q(y)\,dy\right| \leq c_3\epsilon^5 \text{ for constant } c_3 > 0 \qquad \text{(C.12)}$$

Combining Equations (C.6), (C.7), (C.8), (C.12) , we have

$$\int_b^\infty e^{\alpha y^2/\sigma_2^2}y^2 q(y)\,dy - (c_1+c_2)\epsilon^5 \leq \int_b^\infty e^{\alpha(y-\mu_1)^2/\sigma_1^2}y^2 q(y)\,dy \leq \int_b^\infty e^{\alpha y^2/\sigma_2^2}y^2 q(y)\,dy$$

$$\int_b^\infty e^{\alpha y^2/\sigma_2^2}y^2 q(y)\,dy - (c_1+c_3)\epsilon^5 \leq \int_{-a}^\infty e^{\alpha(y+\mu_1)^2/\sigma_1^2}y^2 q(y)\,dy \leq \int_b^\infty e^{\alpha y^2/\sigma_2^2}y^2 q(y)\,dy + c_3\epsilon^5$$

By choosing $C' = c_1 + c_2 + c_3$, we prove the lemma.

$\square$

**Lemma 35** (Approximated for Re-weighted Zeroth Moment). *By choosing $\alpha = -c\epsilon^2$, for some constant $C' > 0$, we have the following inequalities.*

$$\int_b^\infty e^{\alpha y^2/\sigma_2^2}q(y)\,dy - C'\epsilon^5 \leq \int_b^\infty e^{\alpha(y-\mu_1)^2/\sigma_1^2}q(y)\,dy \leq \int_b^\infty e^{\alpha y^2/\sigma_2^2}q(y)\,dy \qquad \text{(C.13)}$$

$$\int_b^\infty e^{\alpha y^2/\sigma_2^2}q(y)\,dy - C'\epsilon^5 \leq \int_{-a}^\infty e^{\alpha(y+\mu_1)^2/\sigma_1^2}q(y)\,dy \leq \int_b^\infty e^{\alpha y^2/\sigma_2^2}q(y)\,dy + C'\epsilon^5 \qquad \text{(C.14)}$$

The proof follows exactly from the proof of Lemma 34 by replacing $y^2$ with 1.

Now we are ready to prove Lemma 5.

**Lemma 5** (Quantitative Spectral Gap of Contrastive Covariance). *Suppose that $|a+b| < \epsilon^5$. Choose $\alpha_3 = -c_3\epsilon^2$ for some constant $c_3 > 0$. Then, for an absolute constant $C$, the top two eigenvalues $\lambda_1 \geq \lambda_2$ of the corresponding re-weighted covariance of $P$ satisfy*

$$\lambda_1 - \lambda_2 \geq C\epsilon^3\lambda_1.$$

*Proof.* By calculation, we have

$$\mathop{\mathbb{E}}_{x\sim\hat{r}} e^{\alpha x^2}x^2 = \frac{\int_{b/\sigma_2}^{\infty} e^{\alpha x^2}x^2\sigma_2 q(x\sigma_2)\,dx}{\int_b^{\infty} q(x)\,dx} = \frac{\int_b^{\infty} e^{\alpha y^2/\sigma_1^2}y^2 q(y)\,dy}{\sigma_2^2\int_b^{\infty} q(x)\,dx}$$

$$\mathop{\mathbb{E}}_{x\sim\hat{q}} e^{\alpha x^2}x^2 = \frac{\int_{-\infty}^{(a-\mu_1)/\sigma_1} e^{\alpha x^2}x^2\sigma_1 q(x\sigma_1+\mu_1)\,dx + \int_{(b-\mu_1)/\sigma_1}^{\infty} e^{\alpha x^2}x^2\sigma_1 q(x\sigma_1+\mu_1)\,dx}{\int_{-a}^{\infty} q(x)\,dx + \int_b^{\infty} q(x)\,dx}$$

$$= \frac{\int_{-\infty}^{a} e^{\alpha(y-\mu_1)^2/\sigma_1^2}(y-\mu_1)^2 q(y)\,dy + \int_b^{\infty} e^{\alpha(y-\mu_1)^2/\sigma_1^2}(y-\mu_1)^2 q(y)\,dy}{\sigma_1^2\left(\int_{-a}^{\infty} q(x)\,dx + \int_b^{\infty} q(x)\,dx\right)}$$

$$= \frac{\int_{-a}^{\infty} e^{\alpha(y+\mu_1)^2/\sigma_1^2}(y+\mu_1)^2 q(y)\,dy + \int_b^{\infty} e^{\alpha(y-\mu_1)^2/\sigma_1^2}(y-\mu_1)^2 q(y)\,dy}{\sigma_1^2\left(\int_{-a}^{\infty} q(x)\,dx + \int_b^{\infty} q(x)\,dx\right)}$$

$$= \frac{\int_{-a}^{\infty} e^{\alpha(y+\mu_1)^2/\sigma_1^2}y^2 q(y)\,dy + \int_b^{\infty} e^{\alpha(y-\mu_1)^2/\sigma_1^2}y^2 q(y)\,dy}{\sigma_1^2\left(\int_{-a}^{\infty} q(x)\,dx + \int_b^{\infty} q(x)\,dx\right)}$$

$$+ \frac{2\mu_1}{\sigma_1^2}\frac{\int_{-a}^{\infty} e^{\alpha(y+\mu_1)^2/\sigma_1^2}yq(y)\,dy - \int_b^{\infty} e^{\alpha(y-\mu_1)^2/\sigma_1^2}yq(y)\,dy}{\int_{-a}^{\infty} q(x)\,dx + \int_b^{\infty} q(x)\,dx}$$

$$+ \frac{\mu_1^2}{\sigma_1^2}\frac{\int_{-a}^{\infty} e^{\alpha(y+\mu_1)^2/\sigma_1^2}q(y)\,dy + \int_b^{\infty} e^{\alpha(y-\mu_1)^2/\sigma_1^2}q(y)\,dy}{\int_{-a}^{\infty} q(x)\,dx + \int_b^{\infty} q(x)\,dx}$$

The first term is close to $\mathop{\mathbb{E}}_{x\sim\hat{r}} e^{\alpha x^2}x^2$ while the second and third terms are close to zero. We first give the bound on the absolute values of last two terms. Since $\alpha < 0$,

$$\left|\int_{-a}^{\infty} e^{\alpha(y+\mu_1)^2/\sigma_1^2}yq(y)\,dy - \int_b^{\infty} e^{\alpha(y-\mu_1)^2/\sigma_1^2}yq(y)\,dy\right|$$

$$\leq \int_{-a}^{\infty} e^{\alpha(y+\mu_1)^2/\sigma_1^2}yq(y)\,dy + \int_b^{\infty} e^{\alpha(y-\mu_1)^2/\sigma_1^2}yq(y)\,dy$$

$$\leq \int_{-a}^{\infty} yq(y)\,dy + \int_b^{\infty} yq(y)\,dy < 1$$

Similarly,

$$\int_{-a}^{\infty} e^{\alpha(y+\mu_1)^2/\sigma_1^2}q(y)\,dy + \int_b^{\infty} e^{\alpha(y-\mu_1)^2/\sigma_1^2}q(y)\,dy < 1$$

By Lemma 32, Lemma 34 and Lemma 35, we have

$$\frac{\int_{-a}^{\infty} e^{\alpha(y+\mu_1)^2/\sigma_1^2}y^2 q(y)\,dy + \int_b^{\infty} e^{\alpha(y-\mu_1)^2/\sigma_1^2}y^2 q(y)\,dy}{\sigma_1^2\left(\int_{-a}^{\infty} q(x)\,dx + \int_b^{\infty} q(x)\,dx\right)}$$

$$\leq \frac{2\int_b^{\infty} e^{\alpha y^2/\sigma_2^2}y^2 q(y)\,dy + C'\epsilon^5}{2\int_b^{\infty} q(y)\,dy}\cdot\frac{1+2e\epsilon^3}{\sigma_2^2}$$

$$= \frac{\int_b^{\infty} e^{\alpha y^2/\sigma_2^2}y^2 q(y)\,dy}{\sigma_2^2\int_b^{\infty} q(y)\,dy} + \frac{C'\epsilon^5}{\sigma_2^2 2\int_b^{\infty} q(y)\,dy} + \frac{2e\epsilon^3}{\sigma_2^2}\frac{2\int_b^{\infty} e^{\alpha y^2/\sigma_2^2}y^2 q(y)\,dy + C'\epsilon^5}{2\int_b^{\infty} q(y)\,dy}$$

$$< \frac{\int_b^{\infty} e^{\alpha y^2/\sigma_2^2}y^2 q(y)\,dy}{\sigma_2^2\int_b^{\infty} q(y)\,dy} + c_1\epsilon^4 \quad \text{for constant } c_1 > 0$$

By combining with the second and third terms, we conclude that for constants $c_3, c_4 > 0$,

$$\mathop{\mathbb{E}}_{x\sim\hat{r}} e^{\alpha x^2}x^2 - c_3\epsilon^4 < \mathop{\mathbb{E}}_{x\sim\hat{q}} e^{\alpha x^2}x^2 < \mathop{\mathbb{E}}_{x\sim\hat{r}} e^{\alpha x^2}x^2 + c_4\epsilon^4 \tag{C.15}$$

Similarly, we have

$$\underset{x \sim \hat{r}}{\mathbb{E}} e^{\alpha x^2} - c_3 \epsilon^4 < \underset{x \sim \hat{q}}{\mathbb{E}} e^{\alpha x^2} < \underset{x \sim \hat{r}}{\mathbb{E}} e^{\alpha x^2} + c_4 \epsilon^4 \tag{C.16}$$

Then, we would like to compute the gap between first and second eigenvalues of the re-weighted second moment of $\hat{q}$. We denote $Q$ as the product of $\hat{q}$ and $d-1$ fold of $q$. For $2 \leq j \leq d$,

$$\underset{x \sim Q}{\mathbb{E}} e^{\alpha \|x\|^2} x_1^2 - \underset{x \sim Q}{\mathbb{E}} e^{\alpha \|x\|^2} x_j^2 = \left( \underset{x_1 \sim \hat{q}}{\mathbb{E}} e^{\alpha x_1^2} x_1^2 \underset{x_2 \sim q}{\mathbb{E}} e^{\alpha x_2^2} - \underset{x_2 \sim q}{\mathbb{E}} e^{\alpha x_2^2} x_2^2 \underset{x_1 \sim \hat{q}}{\mathbb{E}} e^{\alpha x_1^2} \right) \left( \underset{x \sim q}{\mathbb{E}} e^{\alpha x^2} \right)^{d-2}$$

We define $T(\alpha)$ as follows.

$$T(\alpha) = \underset{x_1 \sim \hat{q}}{\mathbb{E}} e^{\alpha x_1^2} x_1^2 \underset{x_2 \sim q}{\mathbb{E}} e^{\alpha x_2^2} - \underset{x_2 \sim q}{\mathbb{E}} e^{\alpha x_2^2} x_2^2 \underset{x_1 \sim \hat{q}}{\mathbb{E}} e^{\alpha x_1^2}$$

Recall that $S(\alpha)$ is defined in Lemma 30.

$$S(\alpha) = \underset{x_1 \sim \hat{r}}{\mathbb{E}} e^{\alpha x_1^2} x_1^2 \underset{x_2 \sim q}{\mathbb{E}} e^{\alpha x_2^2} - \underset{x_2 \sim q}{\mathbb{E}} e^{\alpha x_2^2} x_2^2 \underset{x_1 \sim \hat{r}}{\mathbb{E}} e^{\alpha x_1^2}$$

We calculate the difference between $T(\alpha)$ and $S(\alpha)$ using Equation (C.15) and Equation (C.16).

$$S(\alpha) - T(\alpha)$$
$$= \left( \underset{x_1 \sim \hat{r}}{\mathbb{E}} e^{\alpha x_1^2} x_1^2 - \underset{x_1 \sim \hat{q}}{\mathbb{E}} e^{\alpha x_1^2} x_1^2 \right) \underset{x_2 \sim q}{\mathbb{E}} e^{\alpha x_2^2} - \left( \underset{x_1 \sim \hat{r}}{\mathbb{E}} e^{\alpha x_1^2} - \underset{x_1 \sim \hat{q}}{\mathbb{E}} e^{\alpha x_1^2} \right) \underset{x_2 \sim q}{\mathbb{E}} e^{\alpha x_2^2} x_2^2$$
$$\leq c_3 \epsilon^4 + c_4 \epsilon^4$$

By Lemma 30, $S(\alpha) > C' \epsilon^3$ for constant $C' > 0$. So we know $T(\alpha) > C \epsilon^3$ for $C > 0$. Then all proof follows as same as the case when $a + b = 0$. We write out the proof for completeness.

For $2 \leq j \leq d$,

$$\underset{x \sim P}{\mathbb{E}} e^{\alpha \|x\|^2} x_1^2 - \underset{x \sim P}{\mathbb{E}} e^{\alpha \|x\|^2} x_j^2$$
$$= \left( \underset{x_1 \sim \hat{q}}{\mathbb{E}} e^{\alpha x_1^2} x_1^2 \underset{x_2 \sim q}{\mathbb{E}} e^{\alpha x_2^2} - \underset{x_2 \sim \hat{q}}{\mathbb{E}} e^{\alpha x_2^2} x_1^2 \underset{x_1 \sim q}{\mathbb{E}} e^{\alpha x_1^2} \right) \left( \underset{x \sim q}{\mathbb{E}} e^{\alpha x^2} \right)^{d-2}$$
$$= T(\alpha) \frac{\underset{x \sim P}{\mathbb{E}} e^{\alpha \|x\|^2} x_1^2}{\underset{x_1 \sim \hat{q}}{\mathbb{E}} e^{\alpha x_1^2} x_1^2 \underset{x_2 \sim q}{\mathbb{E}} e^{\alpha x_2^2}}$$

Since $\alpha_3 < 0$, we have

$$\underset{x_1 \sim \hat{q}}{\mathbb{E}} e^{\alpha_3 x_1^2} x_1^2 \leq \underset{x_1 \sim \hat{q}}{\mathbb{E}} x_1^2 = 1, \ \underset{x_2 \sim q}{\mathbb{E}} e^{\alpha_3 x_2^2} \leq 1.$$

Also we have shown that $T(\alpha_3) > C \epsilon^3$. So we have

$$\underset{x \sim P}{\mathbb{E}} e^{\alpha \|x\|^2} x_1^2 - \underset{x \sim P}{\mathbb{E}} e^{\alpha \|x\|^2} x_j^2 \geq C \epsilon^3 \underset{x \sim P}{\mathbb{E}} e^{\alpha \|x\|^2} x_1^2$$

Finally we will show that the first eigenvector corresponds to $e_1$. For any $v \in \mathbb{R}$, define $\phi(v)$ as

$$\phi(v) := \underset{x \sim P}{\mathbb{E}} \frac{e^{\alpha_3 \|x\|^2 v^\top x x^\top v}}{v^\top v}$$

Then we know for $2 \leq j \leq d$,

$$\phi(e_1) - \phi(e_j) > C \epsilon^3 \phi(e_1)$$

For any vector $v = \sum_{i=1}^d \gamma_i e_i$, we have

$$\phi(v) = \frac{1}{\sum_{i=1}^d \gamma_i^2} \mathbb{E} e^{\alpha_3 \|x\|^2} (\sum_{i=1}^d \gamma_i x_i)^2$$

$$= \frac{1}{\sum_{i=1}^{d} \gamma_i^2} \left( \sum_{i=1}^{d} \gamma_i^2 \phi(e_i) + 2 \, \mathbb{E} \, e^{\alpha \|x\|^2} \sum_{i \neq j} \gamma_i \gamma_j x_i x_j \right)$$

$$= \frac{1}{\sum_{i=1}^{d} \gamma_i^2} \left( \sum_{i=1}^{d} \gamma_i^2 \phi(e_i) + 2 \sum_{i \neq j} \gamma_i \gamma_j \, \mathbb{E} \, e^{\alpha_3 \sum_{k \neq i,j} x_k^2} \, \mathbb{E} \, e^{\alpha_3 \langle x, e_i \rangle^2} x_1 \, \mathbb{E} \, e^{\alpha \langle x, e_j \rangle^2} x_j \right)$$

$$= \frac{\sum_{i=1}^{d} \gamma_i^2 \phi(e_i)}{\sum_{i=1}^{d} \gamma_i^2}$$

$$\leq \phi(e_1)$$

This shows that the top eigenvalue of $\tilde{\Sigma}$ is $\lambda_1 = \max_v g(v) = g(e_1)$. In other word, the top eigenvector is $e_1$. Similarly the second eigenvalue of $\tilde{\Sigma}$ is $\lambda_2 = \max_{v:v \perp e_1} \phi(v) = \phi(e_j), 2 \leq j \leq d$. So we get

$$\lambda_1 - \lambda_2 > C\epsilon^3 \lambda_1$$

$\square$

