# OpenReview forum: "Contrastive Moments: Unsupervised Halfspace Learning in Polynomial Time"
_NeurIPS.cc/2023/Conference — NeurIPS 2023 poster_

### Official Review · Reviewer_rhCN · 2023-06-28

**Soundness:** 4 excellent
**Presentation:** 4 excellent
**Contribution:** 4 excellent
**Rating:** 8
**Confidence:** 3

**Summary:**

Assuming that the data is generated from a full-rank affine transformation of a $d$-fold product distribution, one of which is divided into two groups by a halfspace. The authors address the problem of learning the halfspace by weighting the isotropized sample so that either the corresponding sample mean or the top eigenvector of the covariance matrix shifts along the normal direction to the halfspace.

**Strengths:**

- Good presentation of theoretical results, with the proof of the main theorem summarized succinctly.
- Interesting idea of using Descartes'  Rule of Signs to single out a nonzero mean from weighted samples.
- Experimental results agree with the theory.
- In contrast to prior work which focuses on Gaussian data, the proposed method extends to data generated from logconcave distributions.

**Weaknesses:**

- Real-world applications is still unclear to me. In what situation do we expect the set-up in Definition 1 to arise? The authors list the methods that benefit from the halfspace learning in Line 40-49, but where do these methods apply?
- Related to the point above, the authors might want to test their method on real-world data.
- The paper seems to have been reorganized but some notations are left unchanged---see questions below.


**Questions:**

- The purpose of Lemma 1 and Lemma 2 is unclear. From a quick search in the supplement, they are not used to prove the main Theorem. And I personally think that Lemma 4 and Lemma 5 are more informative. Do the maximum of the two expectations shows up in the proof of Lemma 4?
- What are $\mu_{\alpha_1}$ and $\mu_{\alpha_2}$ in Lemma 4?
- What are $a'$ and $b'$ in the Proof of Lemma 1?
- Bottom of Page 6: "Since $\hat{q}(x) - \hat{q}(-x) \geq 0$ for $x\geq b'$." looks like an incomplete sentence.
- Proof Idea of Lemma 4: I do not understand the statement "$F'(0)$ has at most one root." and "$F"(0)$ has no root."
- Do the time and sample complexities also depend on the distribution's tail decay rate?

**Limitations:**

The authors address some challenges of improving the sample complexity bound to linear, and extending the algorithm to more general distributions.

---

> ### Author Rebuttal · Authors · 2023-08-10
>
> We thank the reviewer for the careful reading of the paper and the valuable suggestions. Here are specific responses:
>
> - **W1&W2**. Unsupervised halfspace learning is a new perspective on maximum margin clustering. On the practical side, unsupervised learning has a wide range of applications in computer vision, natural language processing and graph, where data might have structure. While the margin assumption we make is likely unrealistic, we expect our methods to extend to "soft margins" where the density in some interval is significantly lower, thereby allowing this direction to be identified. Extending this to the kernel setting, with non-linear features, could be very interesting.
>
>     One real world application that is close to our setting is to do sentimental analysis given text only (without labels). Suppose we have a powerful pre-trained model that maps the input data into embedding vectors, we can cluster the text into either positive or negative without knowing the data labels.
>
>     Our setting and method are also connected to self-supervised learning. As mentioned in line 297, contrastive learning without data augmentation is similar to the contrastive covariance part in our algorithm.
>
> - **W3**. We thank the reviewer for pointing out the issues. We will address them.
>
> - **Q1**. The purpose of Lemmas 1 and 2 is expository --- to give a qualitative bound of two key lemmas (contrastive mean and variance), with simpler proofs that convey the main intuition.  Their quantitative versions are Lemma 4 and Lemma 5, which are actually used in the proof of the main theorem. Though Lemma 4 and Lemma 5 are more informative, their proof contains many technical details.
>
>     In Lemma 1, the maximum of the two expectations being greater than zero indicates that using two different $\alpha$'s in our algorithm will lead to a positive gap in contrastive mean. However to have a polynomial gap (as in Lemma 4), we need to prove much more work, and we do this in the Appendix.
>
> - **Q2**. $\mu_{\alpha_i}$ is the re-weighted mean using $\alpha_i$, which corresponds to $\tilde{\mu}_i$ in Algorithm 1. We will make the notation consistent.
>
> - **Q3**. We denote the support of isotropic density $\hat{q}$ as $\mathbb{R}\backslash[a',b']$, where $a'=(a-\mu_1)/\sigma_1, b'=(b-\mu_1)/\sigma_1$. The notation is defined in Appendix C. We will add the necessary parts to the main body of the paper.
>
> - **Q4**. We apologize for the confusion. The bottom of Page 6 should be as follows.
>
>     We treat $F(\alpha)$ as the integral of $a(x)e^{\alpha x^2}$ for $x\geq -a'$. Since $\hat{q}(x)-\hat{q}(-x)>0 $ for $x> b'$, we have $a(x)> 0$ for $x\in[-a',b')$ and $a(x) < 0$ for $x> b'$. In other words, for increasing $x$, the sign of $a(x)$ only changes once.
>
> - **Q5**. We apologize for the confusion in the proof idea of Lemma 4. In fact, for both $F'(0)$ and $F''(0)$, there exists pairs $(a,b)$ that make one of them zero. However, we can show that $F'(0)$ and $F''(0)$ cannot both be zero at the same time (Lemma 24). The detailed proof is in the Appendix (starting from line 545).
> The proof idea of Lemma 4 starting from line 225 should be as follows.
>
>     By taking the derivatives of $F(\alpha)$, we can show that either $F'(0)\neq 0$ or $F''(0)\neq 0$. Then by Taylor expansion, we can choose two distinct $\alpha$'s (near zero) so that one of the corresponding contrastive means is bounded away from zero.
>
> - **Q6**. Yes, it is possible the time and sample complexity depend on the distribution's tail decay rate, or other properties of the distribution. Here we give a bound for general logconcave distributions. It would be interesting to see if natural assumptions such as decay rate lead to better complexity (and what the quantitative dependence would be). Figure 5.1 shows the difference between Gaussian, Uniform and Exponential, where Uniform has the best performance, and Gaussian has the worst.

---

> > ### Comment · Reviewer_rhCN · 2023-08-17
> > **Response**
> >
> > I thank the authors for thoroughly addressing my concerns. I think providing theoretical results for "hard-margin" problems is an important first step towards more harder and prevalent "soft-margin" problems, so I will keep the score as is.
> >
> > For **Q5**, are we on the same page that "$F'(0)$ has at most one root." and "$F''(0)$ has no root." are typos, since $F'(0)$ and $F''(0)$ are both numbers, and not functions.

---

> > > ### Author Response · Authors · 2023-08-17
> > > **Follow-up Response**
> > >
> > > Yes, sorry, we should have said explicitly, those are typos in our proof idea description. As you point out,  F'(0) and F''(0) are scalars. The corrected proof idea is summarized in our response to Q5.

---

### Official Review · Reviewer_YLvd · 2023-07-03

**Soundness:** 3 good
**Presentation:** 2 fair
**Contribution:** 2 fair
**Rating:** 5
**Confidence:** 3

**Summary:**

This paper proposes a non-standard approach to the problem of learning halfspaces, where instead of labelled samples, the learner only receives unlabelled ones as well as strong prior information about the unknown distribution. In particular, the marginal distribution is assumed to correspond to some affinely skewed product distribution, with identical components except from one which is truncated outside some interval. The unsupervised distribution learning problem corresponds to a supervised learning problem when the labels are determined by a halfspace in the direction of the truncation.

The main result shows that when the truncation region is wide enough but not too wide and either symmetric or bounded away from being symmetric, then the direction of truncation can be discovered efficiently from unlabelled samples. The algorithm exploits standard ideas from distribution learning (data normalization, PCA and sample re-weighing), but its proof involves careful application of appropriate tools.

**Strengths:**

The paper proposes and studies an interesting connection between unsupervised and supervised learning. Moreover, I appreciated the technical work required to prove guarantees about the main algorithm, even though it is based on simple and clean algorithmic ideas.

**Weaknesses:**

There are several concerns I have regarding the setup, the impact and the results. In particular:

1. While the model demonstrates an interesting connection between supervised and unsupervised learning, if we focus on either of these aspects, it seems less relevant. For supervised learning, the distributional assumptions are too strong (usually one either makes strong assumptions on the marginal distribution or assumes the existence of margin, but not both). For unsupervised learning, the component we want to discover is too specific (instead of a general property, like matching some Gaussian moments, it is assumed that the component is just a truncation of the distribution of the other components).

2. On the algorithmic level, the impact seems to be limited. Variants of the algorithm presented have largely appeared in the literature for similar problems.

3. The main result (Theorem 1) does not seem to be completely accurate. More specifically, there has been no account for the case where the truncation region $[a,b]$ has $|a+b|>0$ but $|a+b|\approx 0$ (e.g., $|a+b| = 2^{-d}$). While Lemma 4 accounts for the case when $|a+b|$ is large enough, Lemma 5 only accounts for the case $|a+b|=0$. It is not clear what happens in between. In Figure 5.2, it seems that Lemma 5 could be generalized to capture cases when $|a+b|$ is positive but bounded. However, this has not been proven.

Overall, the paper seems to have important unresolved technical issues and also it could highly benefit by relaxing the distributional assumptions. I would encourage the authors to continue working on the paper, but in its current form, it is below the acceptance threshold.

-- The authors fixed an important technical issue in their proof (item 3 above) and I have increased my score (from 3 to 5) to reflect this.

**Questions:**

I would be interested to hear any comments that the authors have regarding the points (1,2 and 3) I made in the weaknesses section. Moreover, I have the following suggestions regarding the presentation.

1. The problem definition (page 2) needs to be clarified further to avoid confusion. In particular, it is not clear what is known and what is unknown. For example, does the learner know $q,a$ and/or $b$?

2. How does the paper [DK'22] compare to your results? On a more general note, I think it would be beneficial to present in section 1.2 some result on NGCA (e.g. [DK'22]) more concretely, as well as some result on supervised learning and emphasize the trade-offs when considering the problem proposed in this paper.

[DK'22]: Diakonikolas, Ilias, and Daniel Kane. "Non-gaussian component analysis via lattice basis reduction." Conference on Learning Theory. PMLR, 2022.

3. Some typos:
- Line 105: double "the".
- Line 106: "follows uses" $\to$ "uses".

**Limitations:**

Yes

---

> ### Author Rebuttal · Authors · 2023-08-10
>
> We thank the reviewer for their comments. Here are specific responses:
>
> - **W1**. We are solving an unsupervised learning problem --- there are no labels given. This inherently requires assuming structure on the underlying distribution to be identifiable. Halfspaces *with labels* can be learned with no distributional assumptions. However, margins, and special distributions are both considered in the literature either to improve sample complexity or in more involved models such as active learning, agnostic learning and multi-task/lifelong learning. For learning without labels, prior work is centered around ICA and NGCA; in both cases there is a product distribution to be identified and this is done with moments. In the latter, NGCA, even with the Gaussian assumption, it is nontrivial to identify the non-Gaussian component, and methods such as the ones we analyzed have been conjectured, but previously without proof. Moreover, applying moment-matching methods directly gives much worse (super-polynomial) bounds, as the exponent will depend on the moment number, which (as we mention in the paper), will need to be super-constant to identify a direction with a margin. Taking all this into account, we feel that the progress made in this paper, establishing a polytime algorithm for learning margin halfspaces without labels, for Gaussian and more general logconcave product distributions (and their affine transformations) is an exciting result, suggesting there is more to be gained in this direction of study.
>
>     Finally, there are at least two directions that can be explored to further relax the distribution assumption. The first one is to generalize the logconcave distributions to isoperimetric distributions. The second one is to consider a soft margin, wherein the density within the margin is significantly modified without being made zero.
>
> - **W2**. Variants of algorithm have been presented in literature as is shown in the last paragraph of related work. For example, [VX11] used higher order re-weighted moment. [TV18] uses only re-weighted second moment while requiring another copy of data with the same distribution. They conjecture that new ways of re-weighted moment, as we do, could possibly work and this is what we manage to prove --- our algorithm works, using only re-weighted first and second moments.
>
> - **W3**. We apologize for not properly taking care of the "almost symmetric" case and we thank the review for pointing it out. We generalize Lemma 5 to the case when $|a+b|>c\epsilon$. In fact we can show the following lemma.
>
>     **Lemma 5\***. Let $\mu$ be the mean of the distribution restricting $q$ to $\mathbb{R}\backslash [a,b]$. If $|\mu| \leq c\epsilon^{10}$ for some constant $c>0$, then for an absolute constant $C$, the top two eigenvalues $\lambda_1\geq \lambda_2$ of the re-weighted covariance of $P$ satisfy
>     $$
>     \lambda_1-\lambda_2 \geq C\epsilon^{9}.
>     $$
>
>     In the meantime, we will revise Lemma 4 accordingly as follows.
>
>     **Lemma 4**. If $|\mu|>c\epsilon^{10}$ for some constant $c>0$. Then for $\alpha_1=-c_1 \epsilon^{75}/d,\alpha_2=-c_2\epsilon^{42}/d$, the re-weighted mean of $P$ satisfies
>
>     $$
>     \max(u^\top \mu_{\alpha_1},u^\top \mu_{\alpha_2}) > \frac{C \epsilon^{127}}{d^2}, \text{for some constant } C>0.
>     $$
>     $$
>     \forall v\bot u, v^\top \mu_{\alpha_1}=v^\top \mu_{\alpha_2}=0.
>     $$
>
>     The proof of the contrastive covariance lemma is built upon the original proof of Lemma 5 while bounding the error caused the asymmetry of $a$ and $b$. We append the proof in the following link:
>     <https://anonymous.4open.science/r/Submission-7B72/Proof_appended.pdf>
>
> - **Q1**. In our problem, we are given only data sampled from distribution $\hat{P}$. In other words, the learner does not know $q,a,b$.
>
> - **Q2**. We thank the reviewer for providing this reference. We will expand our discussion of related work on NGCA. Briefly,
> [DK22] considers the case when the underlying distribution along with the hidden direction is discrete or nearly discrete. For example, they consider the distribution supported on $k$ intervals, each of length at most $\eta$. However, they require $\eta<2^{-\Omega(dk^2)}B^{-\Omega(k)}$ for constant $B$ to make the algorithm works. Instead, we consider the distribution lies on two intervals contained in $(-\infty, a],[b,\infty)$. "Almost-discreteness" is essential in [DK22], and e.g., their algorithm cannot handle the case of a single Gaussian with a margin (note also that moment matching to find the margin direction could require a superconstant moment).
>
> References:
>
> [DK22] Diakonikolas, Ilias, and Daniel Kane. "Non-gaussian component analysis via lattice basis reduction." Conference on Learning Theory. PMLR, 2022.
>
> [VX11] Vempala, Santosh S., and Ying Xiao. "Structure from local optima: Learning subspace juntas via higher order PCA." arXiv preprint arXiv:1108.3329 (2011).
>
> [YV18] Tan, Yan Shuo, and Roman Vershynin. "Polynomial time and sample complexity for non-gaussian component analysis: Spectral methods." Conference On Learning Theory. PMLR, 2018.

---

> > ### Comment · Reviewer_YLvd · 2023-08-11
> >
> > I thank the authors for responding to my concerns. I have increased my score to reflect that I now believe the paper to be technically sound.

---

### Official Review · Reviewer_Ua7t · 2023-07-08

**Soundness:** 4 excellent
**Presentation:** 3 good
**Contribution:** 3 good
**Rating:** 7
**Confidence:** 4

**Summary:**

This work addresses learning a halfspace with margin from unlabeled samples (which is possible due the combination of margin and distributional assumptions). They impose that the ambient distribution is an affine transformation of the d-fold product of some 1D symmetric log-concave distribution, conditioned on one coordinate lying outside of an interval with mass at least $\varepsilon$. They provide an algorithm for recovering this distribution up to TV error $\delta$ with time and sample complexity $poly(1/\varepsilon,1/\delta,d)$, based on analyzing the first two moments of a certain exponential reweighting of the empirical distribution.

**Strengths:**

- The paper was generally well-written and organized. The proposed algorithm is easy to implement, and its analysis requires some clever new techniques.
- I appreciate the initial simplified lemmas; they allow the reader to understand main ideas at a high-level without getting bogged down with the full details.
- Approaches for this problem based on previous work would seem to require Gaussianity and fail to provide as strong recovery guarantees in polynomial time
- Experiments validate their algorithm on Gaussian and non-Gaussian data

**Weaknesses:**

- The notion of margin seems a bit non-standard. Can the authors relate this to the standard notion (or refer to an existing example of this usage)?
- The proven time/sample complexity, while polynomial, is quite slow
- While the authors suspect that improved guarantees are possible for the same algorithm, it would be preferable if they could support this conjecture with experiments; the current sample sizes are quite large

Minor Nits:
- Some inconsistency in Sec 3 for whether $\hat{q}$ is isotropized or not (it mostly is, but not in the warm up)
- Inconsistency between single bars and double bars for the Euclidean norm (e.g. Lemma 1 uses both)
- Inconsistency between $\mu_i$ and $\mu_{\alpha_i}$ in Sec 3
- Line 214, clause beginning with "Since" looks like it should be combined with following sentence
- Line 217, should be for $x \geq 0$?
- Perhaps the rule of signs should be stated in the integral form for which is applied? Also, it would improve readability to mention the u-substitution used to eliminate the square in the exponential
- Line 225, $F'(0)$ is a scalar, doesn't have a root, same with $F''(0)$ (presumably should just be $\alpha$)
- Line 226, doesn't $F''(0) \neq 0$ always hold?
- Equation after line 240, would match previous notation to use $\hat{q}(x)$?

**Questions:**

- The results are stated for non-singular affine transformations. What happens as the transformation approaches singularity? Given that there appears to be no breakdown in the run time bound in this regime, the non-singularity requirement was surprising to me

**Limitations:**

Yes, authors are clear about assumptions and potential impacts.

---

> ### Author Rebuttal · Authors · 2023-08-10
>
> We thank the reviewer for the careful reading of the paper and the valuable suggestions. Here are specific responses:
> - **W1**. In the literature, halfspace margin is defined as the distance between the halfspace and the data points closest to the halfspace (support vectors). In our setting (Definition 1), the margin is $(b-a)/2$.
> Here we require the density $q([a,b]) \geq \epsilon$.
> From the the property that the max density of logconcave distribution is upper bounded by a constant, we derive that
> the margin in our setting is at least $\epsilon/2$.
>
> - **W2&W3**. The time and sample complexity in our analysis, though polynomial in $d,1/\epsilon$, has a high order of dependence on $1/\epsilon$. We conjecture that the dependence can be improved.
> Experiments in the supplementary material on three representative logconcave distributions --- uniform, gaussian and exponential distributions ---  indicate that the sample complexity is possibly linear in $1/\epsilon$ (Appendix Figure D.2).
>
> - **M1**. We apologize for the confusion caused by the inconsistency between the warm-up case and general case. We will have a separate warm-up subsection to avoid inconsistency.
> In the warm-up case (isotropic isoperimetric distributions), $\hat{q}$ is not isotropized. The algorithm used here is PCA, which
> is different from Algorithm 1 and we do not have the isotropize step.
> For general distributions, the notation is consistent with Definition 1.
>
> - **M2**. We use single bars to denote the absolute value of a scalar, and double bars to denote the $l_2$ norm of a vector (line 197). In Lemma 1, the correct formula should be
>
>     $\max(|E_{x\sim P} e^{\alpha_1||x||^2}u^\top x|,  |E_{x\sim P} e^{\alpha_2 ||x||^2} u^\top x|)> 0.$
>
> - **M3**. We thank the reviewers for pointing out. We will consistently write $\mu_{\alpha_i}$ to indicate the re-weighted mean by choosing $\alpha_i$.
>
> - **M4**. Line 214, the whole sentence should be as follows.
>
>     We treat $F(\alpha)$ as the integral of $a(x)e^{\alpha x^2}$ for $x\geq -a'$. Since $\hat{q}(x)-\hat{q}(-x)>0 $ for $x> b'$, we have $a(x)> 0$ for $x\in[-a',b')$ and $a(x) < 0$ for $x> b'$. In other words, for increasing $x$, the sign of $a(x)$ only changes once.
>
> - **M5**. Yes, it should be $x\geq 0$.
>
> - **M6**. We thank the reviewer for the suggestion.
> We will complement with the integral formula of the Descartes Rule of Signs that we can apply to. To be specific, define
> $F(\alpha) = \int_0^\infty a(x) e^{\alpha x^2}dx.$
> Then the number of roots of $F(\alpha)=0$ is at most the number of sign changes in $a(x),x\geq 0$.
>
>     We sketch the proof here.
>     The theorem can be proved with induction on the number of sign changes. Let one of the sign changes occur at $x_0$. Let $F_0(\alpha) = \int_0^\infty a(x) e^{\alpha (x^2-x_{0}^2)}dx$, which has the same zeros as $F(\alpha)$. By taking a derivative, we get $F_0'(\alpha) = \int_0^\infty a(x)(x^2-x_0^2)e^{\alpha (x^2-x_{0}^2)}dx$. The new sequence $a(x)(x^2-x_{0}^2)$ has one less sign change than $a(x)$. By a lemma stating that the zeros of a function is at most the zeros of its derivative plus one, we can complete the inductive step.
>
> - **M7&M8**. We apologize for the confusion in the proof idea of Lemma 4. In fact, for both $F'(0)$ and $F''(0)$, there exists pairs of $(a,b)$ that makes one of them to be zero. However, we can show that $F'(0)$ and $F''(0)$ cannot be zero at the same time (Lemma 24). The detailed proof is in the Appendix (starting from line 545).
> The proof idea of Lemma 4 starting from line 225 should be as follows.
>
>     By taking the derivatives of $F(\alpha)$, we can show that either $F'(0)\neq 0$ or $F''(0)\neq 0$. Then by Taylor exansion, we can choose two distinct $\alpha$'s (near zero) so that one of the corresponding contrastive means is bounded away from zero.
>
> - **M9**. Yes, we use $\hat{q}(x)$ to denote the isotropized density  after restricting $q$ to $\mathbb{R}\backslash[a,b]$ (Definition 1) in the whole paper except in the warm-up part.
>
> - **Q**. We thank the reviewer for this insightful question.
> We require the non-singularity of the affine transformation mainly because we would like to preserve the margin after the affine transformation. What we really need is that the affine transformation does not collapse the margin direction.

---

> > ### Comment · Reviewer_Ua7t · 2023-08-10
> >
> > Thanks for these clarifications. I find the answers to my questions satisfactory and maintain my accept decision.

---

### Official Review · Reviewer_6stc · 2023-07-12

**Soundness:** 3 good
**Presentation:** 3 good
**Contribution:** 3 good
**Rating:** 6
**Confidence:** 4

**Summary:**

In this work, the authors study the problem of unsupervised learning a halfspace. In their setting they have a continuous distribution (isotropic-log-concave) and they remove $\geq \epsilon$ mass from an interval $u\cdot x\in[a,b]$, and their goal is to find the half space that separates the two regions. Their idea is basically to reweight the samples and use the first two moments to find the unknown vector.

**Strengths:**

Well written and easy to follow. The authors provide experiments a proof of concept. This is a nice problem and it was not studied before. Even if the method of attacking this problem is what someone expect, still I found the results intersting.

**Weaknesses:**

1. The distributional assumptions are strong. They consider a product of the same log-concave distribution and also they assume that this distribution is symmetric.
2. Some steps that are trivial are over-simplified where some other steps that are not easy to follow are not explained, especially in proof of lemma 4.
3. The results are not tight as is needed a high polynomial on $\epsilon$.



**Questions:**

545-546: Is this part of the proof ok? The distribution is no longer symmetric after removing the mass.
I think this problem is closed related to learning with truncated samples. Maybe the authors should comment on this/add related work.

typos:
550: I guess you mean that the directions are uncorrelated as after removing the mass and rotating this is not a product distribution but it is true that the directions are uncorrelated.
529: there is a missing number.

**Limitations:**

no limitations.

---

> ### Author Rebuttal · Authors · 2023-08-10
>
> We thank the reviewer for the feedback. Regarding specific points:
>
> - **W1**. Our data distribution applies an unknown affine transformation on the product logconcave distribution. So the input data itself is neither isotropic nor a product distribution. We note that the two prominent special cases are product distributions (ICA) and Gaussians with margin. As far as we know these are the first results even under these restrictions (some form of which is necessary to guarantee uniqueness of the learning problem).
>
> - **W2**.  See responses to Q's below.
>
> - **W3**. The results are not tight in $\epsilon$ and our experiments (Figure D.2) suggest a linear dependence on $\epsilon$. Our primary goal for this paper was to prove that the sample and time complexity of the problem are polynomial. We agree that getting a tight bound is an important future goal.
>
> - **Q1**. In any direction orthogonal to the halfspace normal vector, the contrastive mean along the direction is zero by symmetry. Specifically since the underlying distribution $P$ is a product distribution, all orthogonal directions are independent. Removing slices in one direction orthogonal to all others does not affect the symmetry of orthogonal directions.
>
> - **Q2**.  Lemma 4 considers distribution $P$ (product distribution, before affine transformation, see Definition 1 or Figure 1.1(b)), instead of $\hat{P}$, obtained by an unknown affine transformation of $P$. In fact, the Isotropize step in Algorithm 1 turns distribution $\hat{P}$ to $P$.
>
> - **Q3**. We thank the reviewer for pointing this out. There is a missing number in line 529. It should be
> $$
> H(0)> \frac{C_1 \epsilon^{17.5}}{\log^5(1/\epsilon)},\text{  for constant }C_1>0.
> $$

---

### Official Review · Reviewer_efHe · 2023-07-13

**Soundness:** 2 fair
**Presentation:** 2 fair
**Contribution:** 2 fair
**Rating:** 6
**Confidence:** 1

**Summary:**

This paper studies the unsupervised learning for high-dimensional halfspace under affine transformation. The goal is to estimate the normal vector of the halfspace such that the learned distribution is close to the underlying truth. The paper provides an algorithm using so-called ''contrastive moments'' with sample and time complexity guarantee.

**Strengths:**

- The algorithm provided achieves polynomial dependence in $d, 1/\epsilon, 1/\delta$ for both time and sample complexity.
- Existing literature relies on supervised learning, or Gaussianity of the distribution, or NGCA and ICA. While this paper only requires log-concavity.
- Techniques introduced, e.g. contrastive moments, might be of independent interest.

**Weaknesses:**

- The product and isotropic distribution with affine transformation assumption seems to be restrictive, can the analysis or algorithm go beyond that?
- The presentation of the problem is hard to follow, might need more explicit mathematical definitions. For example, where is $\tilde{P}$ in **Problem** defined.

**Questions:**

- Are coordinates of $P(Q)$ independent with each other before the affine transformation?
- Can author(s) provide an real data application example for the problem setup in Definition 1, and specify the the roles of $q,Q,d, P,u$?

**Limitations:**

No limitation is discussed.

---

> ### Author Rebuttal · Authors · 2023-08-10
>
> We thank the reviewer for the feedback. Regarding specific points:
> - **W1**. We clarify that our algorithm does not assume that the data is isotropic. We assume that samples are drawn from a distribution obtained after an (arbitrary) affine transformation of a product logconcave distribution. Both the underlying distribution and the affine transformation are unknown. This family is significantly more general than the Gaussian setting considered in prior work. Since no labels are given, there needs to be a (strong) distributional assumption to uniquely identify the underlying halfspace.
> We hope that future work might further generalize these results and techniques.
> -  **W2**. We will try to clarify the presentation as much as possible.
> $\tilde{P}$ is defined as the distribution induced by the normal vector $\tilde{u}$ output by the algorithm along with a margin. We will make this explicit.
> - **Q1**. Yes, the coordinates are independent of each other before the affine transformation. But since an unknown affine transformation is applied, they are not independent in the given representation.
> - **Q2**. Describing the setting in Definition 1 using real-world data applications is challenging due to the disparity between theoretical constructs and practical realities. We illustrate with an example where $q$ represents a standard Gaussian density, where $q(x)=\frac{1}{\sqrt{2\pi}}e^{-x^2/2}$. Let $d$ be the dimension.
> Then $Q$ refers to the density of $d$-dimensional standard Gaussian $N(0,I_d)$. Given $a<b$ (endpoints of a margin in one dimension),
> $\hat{q}$ is obtained by isotropizing a truncated standard Gaussian density. Given a unit vector $u\in\mathbb{R}^d$, let $P$ be the product distribution with density $\hat{q}$ along with $u$ direction and $q$ in all directions that are orthogonal to $u$. Then we apply an unknown affine transformation to $P$, and get $\hat{P}$, which is a general Gaussian distribution with a margin normal to an unknown direction $\hat{u}$.

---

> > ### Comment · Reviewer_efHe · 2023-08-14
> >
> > I thank the authors for the response. I have increased the score.

---

### Decision · Program_Chairs · 2023-09-21

**Decision:**

Accept (poster)

**Comment:**

All reviewers found this paper well written and the theoretical results interesting, so agreed the paper should be accepted. The authors cleared up some doubts in the discussion period, and we hope they both improve the completeness of the proof in the final version, and expand discussion of prior work mentioned by the reviewers.